# PROVABLE BENEFITS OF SINUSOIDAL ACTIVATION FOR MODULAR ADDITION

## ABSTRACT

This paper studies the role of activation functions in learning modular addition with two-layer neural networks. We first establish a sharp expressivity gap: sine MLPs admit width-2 exact realizations for any fixed length $m$ and, with bias, width-2 exact realizations uniformly over all lengths. In contrast, the width of ReLU networks must scale linearly with $m$ to interpolate, and they cannot simultaneously fit two lengths with different residues modulo $p$. We then provide a novel Natarajan-dimension generalization bound for sine networks, yielding nearly optimal sample complexity $\widetilde{\mathcal{O}}(p)$ for ERM over constant-width sine networks. We also derive width-independent, margin-based generalization for sine networks in the overparametrized regime and validate it. Empirically, sine networks generalize consistently better than ReLU networks across regimes and exhibit strong length extrapolation.

## 1 INTRODUCTION

Most modern neural networks use *nonperiodic* activations such as ReLU or GELU, a choice that is highly effective on vision and language benchmarks. When the target has inherently periodic structure, however, this choice can be statistically and computationally mismatched: approximating periodic functions with nonperiodic networks may require substantially larger width or depth than architectures that encode periodic features or activations (Rahaman et al., 2019; Rahimi & Recht, 2007; Tancik et al., 2020). Beyond in-distribution generalization, models can degrade under distribution shift, especially at sequence lengths longer than training. Compositional tests (SCAN, CFQ) and long-context suites (LRA) expose brittleness and sensitivity to positional encoding (Lake & Baroni, 2018; Keysers et al., 2020; Tay et al., 2021).

We study this mismatch through a standard testbed in deep learning: *modular addition*. Given $m$ input tokens in $\{0, \ldots, p-1\}$, the label is their sum modulo $p$. This task generalizes $k$-parity and is widely used to probe how networks represent and discover algorithms, as well as to study *grokking*—delayed generalization after a long memorization phase (Power et al., 2022). Mechanistic analyses report *Fourier-like* internal circuits for models that solve modular addition, where tokens are embedded as phases and addition is implemented as rotation on the unit circle. Distinct learning procedures ("clock" vs. "pizza") emerge under different hyperparameters and architectures (Nanda et al., 2023; Zhong et al., 2023). These observations suggest a simple design principle: when the task is periodic, an *explicit periodic inductive bias* should help.

Periodic representations already play a central role across machine learning. Sinusoidal positional encodings are historically canonical in Transformers (Vaswani et al., 2017); ROPE encodes positions as complex rotations, mapping offsets to phase differences and imposing a periodic bias preserving attention geometry (Su et al., 2021). Fourier and random features mitigate spectral bias and improve high-frequency fidelity (Rahimi & Recht, 2007; Tancik et al., 2020; Rahaman et al., 2019); sinusoidal activations (SIREN) enable compact implicit neural representations for images, audio, and PDEs (Sitzmann et al., 2020). In 3D view synthesis (NeRF), Fourier positional encodings are key to recovering fine detail from coordinates (Mildenhall et al., 2020), and spectral parameterizations power operator-learning methods for PDEs (Li et al., 2021b). These observations motivate the following hypothesis:

*On periodic tasks, periodic bias increases expressivity and makes learning provably easier.*

We formalize and test this hypothesis in a minimal yet nontrivial setting: two-layer MLPs trained on a modular addition task with one-hot inputs and a shared, position-independent embedding. While the underlying principle implies broader utility, we restrict our theoretical validation to this testbed to derive sharp separation results. We compare ReLU and sinusoidal activations and analyze the multiclass 0–1 loss in underparameterized, overparameterized, and out-of-distribution regimes.

We summarize our contributions below:

1. **Sharp expressivity gap between sine and ReLU networks.** Sine MLPs achieve exact modular addition with width-2 for any fixed length $m$ (Thm. 4.1) and, with bias, width-2 exact realizations uniformly over all lengths (Thm. 4.2). Without bias, a length-agnostic construction of width $\lfloor (p-1)/2 \rfloor$ attains population accuracy $1 - \frac{1}{p}$ for odd $p$ and accuracy $1 - \frac{2}{p}$ for even $p$ (Thm. 4.2). In contrast, ReLU MLPs require width at least $\frac{m-p}{p+2} = \Omega(m/p - 1)$ for exact realization at length $m$ (Thm. 4.3) and cannot simultaneously fit two lengths $m_1, m_2$ with $m_1 \not\equiv m_2 \pmod{p}$ (Thm. 4.4).

2. **Unified underparameterized generalization for broad activations.** Via a multiclass Natarajan-dimension analysis based on pairwise reduction, we prove uniform convergence bounds for two-layer MLPs with a wide family of activations—piecewise-polynomial (include ReLU), trigonometric-polynomial (include sine), and rational–exponential (include sigmoid/SiLU/QuickGELU). The resulting sample complexity is $\widetilde{\Theta}(dp)$ with width $d$ and vocabulary size $p$ (Thm. 5.6; Tab. 1).

3. **Width-independent margin guarantees for overparameterized networks.** Under spectral- and Frobenius-norm constraints for ReLU and a $\|V\|_{1,\infty}$ constraint for sine, we establish multiclass, width-independent margin generalization bounds. Our sine construction attains large normalized margins, leading to population error $\widetilde{\mathcal{O}}(p/\sqrt{n})$ when the normalized margin is $\Omega(1)$ (Thm. 6.2). In contrast, the best known ReLU interpolants achieve normalized margins that decay exponentially with $m$, yielding substantially weaker bounds under comparable norms (Thm. 6.3).

4. **Near-optimal ERM sample complexity for constant-width sine networks.** We prove that any interpolating algorithm over constant-width sine MLPs has sample complexity $\widetilde{\mathcal{O}}(p)$ (Cor. 5.8).

5. **Empirical validation of our theory.** With matched architectures, datasets, and training budgets, sine MLPs consistently outperform ReLU MLPs on modular addition across under- and overparameterized regimes; in the latter, larger normalized margins track improved test accuracy (Figs. 1-3). Sine MLPs also retain near-perfect accuracy far beyond training lengths, while ReLU MLPs collapse to chance (Figs. 4-5). These advantages extend to Transformers, where sine activations demonstrate significantly better sample efficiency than ReLU and GELU baselines (Fig. 6).

## 2 RELATED WORK

**Modular arithmetic as a probe of algorithmic learning and grokking.** Delayed generalization ("grokking") was first highlighted on modular arithmetic (Power et al., 2022). Reverse-engineering reveals Fourier-style internal mechanisms—tokens represented as phases and addition as rotation (Nanda et al., 2023; Zhong et al., 2023)—while a unifying view shows MLPs and transformers can implement an approximate CRT with coset-tracking neurons using only $\mathcal{O}(\log p)$ frequencies (McCracken et al., 2025). For $m{=}2$, analyses indicate an initial kernel regime followed by feature learning (Mohamadi et al., 2024), consistent with effective-theory explanations of grokking (Liu et al., 2022a). Margin-based perspectives explain the emergence of Fourier features (Morwani et al., 2024; Li et al., 2025), and optimizer/regularization choices modulate the dynamics (Thilak et al., 2022), with related phenomena observed beyond algorithmic data (Liu et al., 2022b). Fourier-style embeddings accelerate modular-addition learning and reduce grokking (Zhou et al., 2024a). These observations motivate architectures with an explicit periodic bias.

**Periodic representations and encodings.** Periodic structure is widely used in modern models: sinusoidal and rotary positional encodings (Vaswani et al., 2017; Su et al., 2021); random Fourier features and sinusoidal encodings to address spectral bias (Rahimi & Recht, 2007; Tancik et al., 2020;

Rahaman et al., 2019); and periodic activations for implicit neural representations (SIREN) (Sitzmann et al., 2020). Spectral parameterizations underlie NeRF and neural operators (Mildenhall et al., 2020; Li et al., 2021b). We instantiate this bias in a minimal algorithmic setting—two-layer MLPs with shared embeddings—showing that sine activations align with modular addition and enable compact constructions with favorable sample complexity.

**Capacity and generalization.** Classical tools bound expressivity via growth functions and sign-pattern counting for semialgebraic classes (Warren, 1968; Goldberg & Jerrum, 1995; Anthony & Bartlett, 2009), while multiclass uniform convergence is governed by the Natarajan dimension (Natarajan, 1989; Haussler & Long, 1995; Shalev-Shwartz & Ben-David, 2014). For piecewise-linear/polynomial networks, nearly tight VC bounds scale like $\Theta(WL \log W)$ up to factors (Bartlett et al., 2017b). We adapt these techniques to integer-valued shared-embedding inputs and analyze both ReLU and sine units, obtaining uniform-convergence bounds and width-independent margin guarantees tailored to our setting.

**Margins, overparameterization, and length generalization.** In overparameterized regimes, margins and layerwise norms are more predictive of generalization than raw width (Neyshabur et al., 2018b). Representative results include spectral-norm and $L_{2,1}$ analyses (Bartlett et al., 2017a), size-independent Rademacher bounds under Frobenius and $L_{1,\infty}$ controls (Golowich et al., 2017), and PAC-Bayesian robustness to weight noise (Neyshabur et al., 2018a). Length extrapolation is a distinct stressor: SCAN/CFQ and LRA expose brittleness and positional-encoding sensitivity (Lake & Baroni, 2018; Keysers et al., 2020; Tay et al., 2021); ALiBi and Hard-ALiBi improve scaling with length (Press et al., 2022; Jelassi et al., 2024), whereas scaling alone often fails and performance can follow term frequency rather than structure (Zhou et al., 2024b; Razeghi et al., 2022). Our analysis and experiments show that periodic activations provide a principled route to strong length extrapolation.

## 3 MODEL SETUP

**Notation.** For an integer $p \geq 2$, let $[p] := \{0, \ldots, p-1\}$ and $e_i$ denote the $i$-th standard basis vector of $\mathbb{R}^p$. For nonnegative $f, g$, we write $f(n) = \mathcal{O}(g(n))$ (respectively $f(n) = \Omega(g(n))$) if there exists an absolute constant $C > 0$ such that for all $n \geq 0$, $f(n) \leq Cg(n)$ (respectively $f(n) \geq Cg(n)$). We use $f(n) = \Theta(g(n))$ when both bounds hold. We write $f(n) = \widetilde{\mathcal{O}}(g(n))$ to suppress absolute constants (independent of the model architecture and data) and polylog factors. The symbols $\widetilde{\Omega}(\cdot)$ and $\widetilde{\Theta}(\cdot)$ are defined analogously.

**Task and data.** Fix an integer $p \geq 2$ and vocabulary $\mathcal{V} = [p]$. Each example is a length-$m$ sequence $s_{1:m} \in [p]^m$ with $s_1, \ldots, s_m \overset{\text{i.i.d.}}{\sim} \text{Unif}([p])$. We use one-hot encoding and a shared, position-independent input embedding, so the network observes only the bag-of-tokens vector

$$x = \sum_{i=1}^{m} e_{s_i} \in \{0, 1, \ldots, m\}^p, \qquad \|x\|_1 = m.$$

The effective instance space[1] is defined as

$$\mathcal{X}_m = \left\{x \in \{0, 1, \ldots, m\}^p : \|x\|_1 = m\right\}, \qquad |\mathcal{X}_m| = \binom{m+p-1}{p-1}.$$

Labels are modular sums $y \equiv \left(\sum_{i=1}^{m} s_i\right) (\text{mod } p) \in [p]$. Let $\mathcal{D}_m$ denote the induced population distribution on $\mathcal{X}_m \times [p]$. Given $n$ training samples we draw

$$S = \left\{(x^{(i)}, y^{(i)})\right\}_{i=1}^{n} \overset{i.i.d.}{\sim} \mathcal{D}_m^n.$$

**Model.** We study two-layer MLPs of width $d$ with a shared input embedding, comparing standard ReLU and sinusoidal activations. Parameters are $\theta = (W, V)$ with $W \in \mathbb{R}^{d \times p}$ and $V \in \mathbb{R}^{p \times d}$. For $x \in \mathbb{R}^p$ and an activation $\sigma \in \{\text{ReLU}, \sin\}$, the score vector[2] is $s^\theta(x) = V\sigma(Wx) \in \mathbb{R}^p$.

---

[1]In the out-of-domain regime, it is extended to $\mathcal{X} := \bigcup_{m \geq 2} \mathcal{X}_m$.

[2]Unless otherwise noted, MLPs are defined without first-layer bias. When bias is used, we write $\theta = (W, V, b)$ and define $s^\theta(x) = V\sigma(Wx + b)$, where $b \in \mathbb{R}^d$.

We refer to networks with $\sigma = \sin$ as *sine networks* and those with $\sigma = \text{ReLU}$ as *ReLU networks*. The induced predictor is $h_\theta(x) = \text{uargmax}_{\ell \in [p]} s_\ell^\theta(x) = \begin{cases} \ell, & \text{if } s_\ell^\theta(x) > s_k^\theta(x) \text{ for all } k \neq \ell, \\ \bot, & \text{otherwise,} \end{cases}$

where $\bot$ denotes an invalid prediction (counted as an error). The predictor returns a valid label only if the maximum score is unique, so $h_\theta$ is well defined. The hypothesis class of score functions is

$$\mathcal{H}_\Theta = \{ s^\theta : \theta = (W, V) \in \mathbb{R}^{d \times p} \times \mathbb{R}^{p \times d} \}.$$

**Training.** We minimize the empirical cross-entropy loss over $S = \mathcal{D}_{\text{train}}$, treating $s^\theta(x)$ as the logits for the $p$-class classification problem with labels in $[p]$. Optimization employs AdamW and Muon; implementation details and hyperparameters are provided in App. C.

## 4 EXPRESSIVITY OF SINE AND ReLU MLPS

We establish a sharp expressivity gap for modular addition under a shared, position-independent embedding. For sine MLPs, we provide explicit constructions showing they are highly efficient: width-2 suffices for exact realization at any fixed length, and with bias, for all lengths simultaneously (Thms. 4.1–4.2). Conversely, we prove that ReLU MLPs face fundamental limitations: their width must scale linearly with sequence length $m$ to interpolate, and they fail to generalize across incongruent lengths (Thms. 4.3–4.4). We outline the logic of these proofs in App. D and provide full details in App. E.

**Theorem 4.1** (Exact realization at fixed length by a width-2 sine network). *For any fixed $m \geq 2$ and $p \geq 2$, there exists a width-2 sine network $s^\theta(x) = V \sin(Wx)$ that exactly realizes $Y \equiv \left( \sum_{i=1}^m s_i \right)$ (mod $p$) for all $x = (s_1, \cdots, s_m) \in \mathcal{X}_m$, i.e., $\mathbb{P}_{(X,Y) \sim \mathcal{D}_m}[h_\theta(X) = Y] = 1$.*

**Theorem 4.2** (Uniform-in-length expressivity of sine networks). *Fix $p \geq 2$ and consider two-layer sine MLPs with prediction rule $h_\theta(x) = \text{uargmax}_{\ell \in [p]} s_\ell^\theta(x)$.*

**With bias.** *There exists a width-2 sine network $s^\theta(x) = V \sin(Wx + b)$ that exactly realizes $Y \equiv \left( \sum_{i=1}^m s_i \right) (\text{mod } p)$ for all $m \geq 2$, that is,*

$$\mathbb{P}_{(X,Y) \sim \mathcal{D}_m}[h_\theta(X) = Y] = 1.$$

**Without bias.** *There exists a sine network $s^\theta(x) = V \sin(Wx)$ of width $d = \lfloor (p-1)/2 \rfloor$ such that, for all $m \geq 2$,*

*1. If $p$ is odd, then $\mathbb{P}_{(X,Y) \sim \mathcal{D}_m}[h_\theta(X) = Y] \geq 1 - \frac{1}{p}$.*

*2. If $p$ is even, then $\mathbb{P}_{(X,Y) \sim \mathcal{D}_m}[h_\theta(X) = Y] \geq 1 - \frac{2}{p}$.*

**Theorem 4.3** (Width lower bound for modular addition with ReLU networks). *If a ReLU network $s^\theta(x) = V \text{ReLU}(Wx)$ exactly realizes modular addition on $\mathcal{X}_m$, then necessarily*

$$d \geq \frac{m - p}{p + 2} = \Omega\left( \frac{m}{p} - 1 \right).$$

**Theorem 4.4** (Impossibility of exact realization at two incongruent lengths for ReLU networks). *Let $m_1, m_2$ with $m_1 \not\equiv m_2 \pmod{p}$. There is no ReLU network $s^\theta(x) = V \text{ReLU}(Wx)$ such that*

$$h_\theta(x) = y(x) \quad \text{for all } x \in \mathcal{X}_{m_1} \cup \mathcal{X}_{m_2}.$$

However, high expressivity does not guarantee generalization. Indeed, constant-size sine networks can shatter infinite sets when inputs are unbounded:

**Example 4.5** (Lem. 7.2 (Anthony & Bartlett, 2009); see also App. H.2). *The class $\mathcal{F} = \{x \mapsto \text{sgn}(\sin(ax)) : a \in \mathbb{R}^+\}$ of functions defined on $\mathbb{N}$ has $\text{VCdim}(F) = \infty$.*

Leveraging the structure of our integer-valued, bounded input space, the following section establishes generalization via uniform-convergence bounds that scale linearly with parameter counts.

## 5 GENERALIZATION IN THE UNDERPARAMETERIZED REGIME

We establish uniform convergence guarantees for two-layer MLPs with shared embeddings across a broad class of activations. We characterize the learnability of these models via the Natarajan dimension. Our proof strategy bounds the growth function by counting sign patterns induced by pairwise margins (Lem. G.6) and invokes the Multiclass Fundamental Theorem (Thm. G.18). We provide a high-level roadmap of this reduction in App. D and detailed proofs in App. G.

**Definition 5.1** (VC-dimension). The *VC-dimension* of $\mathcal{B} \subseteq \{-1, +1\}^{\mathcal{Z}}$, denoted $\mathrm{VCdim}(\mathcal{B})$, is the maximal size of a set $T \subset \mathcal{Z}$ that is *shattered* by $\mathcal{B}$, meaning the restriction of $\mathcal{B}$ to $T$ realizes all sign patterns: $\mathcal{B}_{|_T} = \{-1, +1\}^T$.

**Definition 5.2** (Natarajan-dimension). The *Natarajan-dimension* of $\mathcal{H} \subseteq [p]^{\mathcal{X}}$, denoted $\mathrm{Ndim}(\mathcal{H})$, is the maximal size of a set $S \subset \mathcal{X}$ that is *N-shattered* by $\mathcal{H}$. A set $S$ is N-shattered if there exist $f_1, f_2 : S \to [p]$ with $f_1(x) \neq f_2(x)$ such that for every binary selector $b \in \{1, 2\}^S$, there is some $h \in \mathcal{H}$ satisfying $h(x) = f_{b(x)}(x)$ for all $x \in S$.

**Definition 5.3** (Piecewise-polynomial activation). A function $\sigma : \mathbb{R} \to \mathbb{R}$ is *piecewise polynomial with at most $L \geq 1$ pieces and maximal piece degree $r \geq 1$* if there exist breakpoints

$$-\infty = b_0 < b_1 < \cdots < b_{L-1} < b_L = +\infty$$

and polynomials $P_1, \ldots, P_L$ with $\deg P_\ell \leq r$ such that $\sigma(t) = P_\ell(t)$ for all $t \in (b_{\ell-1}, b_\ell], \ell \in [L]$.

**Definition 5.4** (Trigonometric-polynomial activation). Let $K \in \mathbb{N}_0$. A function $\sigma : \mathbb{R} \to \mathbb{R}$ is a *trigonometric polynomial of degree at most $K$* if

$$\sigma(t) = a_0 + \sum_{k=1}^{K} \big(a_k \cos(kt) + b_k \sin(kt)\big)$$

for some real coefficients $a_0, (a_k)_{k \leq K}, (b_k)_{k \leq K}$.

**Definition 5.5** (Polynomial–rational–exponential activation). Fix $k \in \mathbb{R} \setminus \{0\}$, $c \geq 0$, $\tau > 0$, $a, b \in \mathbb{R}$, and a polynomial $P$ with degree $r := \deg P \in \mathbb{N}_0$. Define

$$\sigma(t) = P(t) \frac{ae^{kt} + b}{ce^{kt} + \tau}.$$

**Theorem 5.6** (Uniform convergence for broad activation families). *Let $\sigma$ be one of: piecewise-polynomial (Def. 5.3), trigonometric-polynomial (Def. 5.4), or polynomial–rational–exponential (Def. 5.5). Let $\mathcal{H}_\sigma$ be the corresponding two-layer class. Then for every $\delta \in (0, 1)$, with probability at least $1 - \delta$ over the random draw of the training set $\mathcal{D}_{train} \sim \mathcal{D}_m^n$,*

$$\sup_{h \in \mathcal{H}_\sigma} \left| \mathbb{P}_{(X,Y) \sim \mathcal{D}_m} [h(X) \neq Y] - \mathbb{P}_{(X,Y) \sim \mathcal{D}_{train}} [h(X) \neq Y] \right| \leq \widetilde{\mathcal{O}}\left( \sqrt{\frac{dp + \log(1/\delta)}{n}} \right).$$

As direct corollaries of Thm. 5.6, we have:

**Corollary 5.7.** *Two-layer MLPs with activation ReLU ($\sigma(t) = \max\{0, t\}$), monomial ($\sigma(t) = t^m$), sine ($\sigma(t) = \sin t$), Sigmoid ($\sigma(t) = \frac{e^t}{e^t+1}$), SiLU ($\sigma(t) = t \, \mathrm{sigmoid}(t)$), QuickGELU ($\sigma(t) = t \, \mathrm{sigmoid}(\beta t), \ \beta > 0$) have sample complexity $\widetilde{\mathcal{O}}(dp)$.*

**Corollary 5.8** (Sample-complexity upper bound for ERM with constant-width sine networks). *Fix a constant width $d \geq 2$. With probability at least $1 - \delta$ over the random draw of the training set $\mathcal{D}_{train} \sim \mathcal{D}_m^n$, for all interpolating ERM solutions $\hat{\theta}$:*

$$\mathbb{P}_{(X,Y) \sim \mathcal{D}_m} \big[h_{\hat{\theta}}(X) \neq Y\big] \leq \widetilde{\mathcal{O}}\left( \sqrt{\frac{p + \log(1/\delta)}{n}} \right),$$

*where $\widetilde{\mathcal{O}}(\cdot)$ hides polylogarithmic factors in $n$, $m$, and $\delta^{-1}$. Consequently, the sample complexity is $\widetilde{\mathcal{O}}(p)$.*

*Remark* 5.9 (Near-optimality). Cor. 5.8 is nearly optimal for label-permutation equivariant algorithms (Def. F.2): it matches the information-theoretic lower bound $\Omega(p)$ established in Thm. F.11. Rmk. F.3 provides notable examples of label-permutation equivariant learners, including AdaGrad, Adam, and GD/SGD with momentum under i.i.d. final-layer initialization.

Table 1: Capacity bounds for two-layer MLPs with $W$ parameters and width $d$. $\widetilde{\Theta}(\cdot)$ suppresses polylog factors in $W$, $d$, and the bound on the input. **Bold** entries are our contributions; Natarajan lower bounds come from Thm. H.2 and upper bounds from Thm. H.6. Full details are in App. H.

| Activation | Input type | VCdim[3] | Ndim |
|---|---|---|---|
| Piecewise linear | real-valued | $\Theta(W \log W)$ | $\widetilde{\Theta}(W)$ |
| Piecewise polynomial | real-valued | $\Theta(W \log(W))$ | $\widetilde{\Theta}(W)$ |
| Pfaffian, incl. standard sigmoid | real-valued | $\mathcal{O}(d^2 W^2)$ | — |
| Standard sigmoid | real-valued | $\Omega(W \log W)$ | $\Omega(W \log W)$ |
| **Standard sigmoid** | **integer-valued, bounded** | $\Omega(W), \mathcal{O}(W)$ | $\widetilde{\Theta}(W)$ |
| Sine | integer-valued, unbounded | $\infty$ | $\infty$ |
| **Trigonometric polynomial** | **integer-valued, bounded** | — | $\widetilde{\mathcal{O}}(W)$ |
| **Rational exponential** | **integer-valued, bounded** | — | $\widetilde{\mathcal{O}}(W)$ |

## 6 GENERALIZATION IN THE OVERPARAMETERIZED REGIME

Uniform-convergence yields stronger bounds for two-layer sine MLPs than for ReLU networks, yet these still scale with hidden width. What happens as width becomes very large? To bridge this gap, we establish width-independent, margin-based generalization bounds. Our analysis builds on the $\ell_\infty$ vector-contraction bound for Rademacher complexity (Foster & Rakhlin, 2019). For sine MLPs, we control the contracted complexity via the Dudley entropy integral and covering numbers adapted to periodic activations. For ReLU MLPs, we leverage positive homogeneity to apply a layer-wise peeling argument (Golowich et al., 2017). We provide a high-level roadmap of these logical steps in App. D before detailing the full proofs in App. J.

Let $v_j$ denote the $j$-th row of $V$ for $j \in [p]$. We write $\|V\|_{1,\infty} := \max_{j \in [p]} \|v_j\|_1$ (the maximum row $\ell_1$-norm), $\|V\|_2$ for the spectral norm, and $\|W\|_F$ for the Frobenius norm.

**Definition 6.1** (Empirical margin). Let $(x, y)$ be a labeled example with $y \in [p]$ and score vector $s^\theta(x) \in \mathbb{R}^p$. The multiclass margin is $\gamma_\theta(x, y) := s^\theta_y(x) - \max_{k \in [p] \setminus \{y\}} s^\theta_k(x)$.

For a finite sample $S = \{(x^{(i)}, y^{(i)})\}_{i=1}^n$, define the (empirical) margin of $S$ as

$$\gamma_\theta(S) := \min_{i \in [n]} \gamma_\theta(x^{(i)}, y^{(i)}).$$

We say that the classifier interpolates $S$ if $\gamma_\theta(S) > 0$.

**Theorem 6.2** (Two-layer sin MLP, margin-based generalization). *Consider the two-layer MLP $s^\theta(x) = V \sin(Wx) \in \mathbb{R}^p$ on $\mathcal{X}_m$. Fix $\delta \in (0, 1)$ and assume $d \geq 2p$. With probability at least $1 - \delta$ over the random draw of the training set $\mathcal{D}_{train} \sim \mathcal{D}_m^n$, for all interpolating solutions $\theta$ with normalized margin $\overline{\gamma}_{\theta, \sin} := \frac{\gamma_\theta(\mathcal{D}_{train})}{\|V\|_{1,\infty}} = \Omega(1)$, it holds that*

$$\mathbb{P}_{(X,Y) \in \mathcal{D}_m}\big[h_\theta(X) \neq Y\big] \leq \widetilde{\mathcal{O}}\left(p\sqrt{\frac{1}{n}}\right),$$

*where $\widetilde{\mathcal{O}}(\cdot)$ hides polylogarithmic factors in $n$, $m$, and $\delta^{-1}$.*

**Theorem 6.3** (Two-layer ReLU MLP, margin-based generalization). *Assume $p > m$, $n > m^2$, and $n \geq 17$. Fix $\delta \in (0, 1)$. Suppose the width satisfies $d \geq 64\,p\,m^{\frac{m}{2}+2}\,4.67^m$. With probability at least $1 - \delta$ over the random draw of the training set $\mathcal{D}_{train} \sim \mathcal{D}_m^n$, for all interpolating solutions $\theta$ with normalized margin $\overline{\gamma}_{\theta, \text{ReLU}} := \frac{\gamma_\theta(\mathcal{D}_{train})}{\|V\|_2 \|W\|_F} = \Omega\left(\frac{1}{\sqrt{p}} \cdot \frac{1}{m^{1.5m+2.5}\,6.34^m}\right)$, it holds that*

$$\mathbb{P}_{(X,Y) \sim \mathcal{D}_m}\big[h_\theta(X) \neq Y\big] \leq \widetilde{\mathcal{O}}\left(p\,m^{1.5m+2.5}6.34^m\sqrt{\frac{m}{n}}\right),$$

*where $\widetilde{\mathcal{O}}(\cdot)$ hides polylogarithmic factors in $n$ and $\delta^{-1}$.*

---

[3]VC-dimension lower bounds are existential: for given size and depth budgets, there exists a network that shatters a set of the claimed cardinality. Upper bounds are universal: they hold for every network in the family.

*Remark* 6.4. The exponential dependence on $m$ in Thm. 6.3 is a proof artifact rather than a fundamental limitation. Our interpolation guarantee uses an explicit construction akin to the "Pizza" algorithm (Zhong et al., 2023), an implementation of the aCRT template (McCracken et al., 2025), which certifies learnability via a mechanistic scheme (approximating periodic embeddings with ReLUs) but yields conservative margin estimates. Empirically (Figs. 3, 11, 13), trained ReLU networks attain much larger margins at smaller widths, indicating that the theoretical gap arises from the inefficiency of this construction rather than intrinsic model constraints.

## 7 EXPERIMENTS

To evaluate our theory, we train two-layer MLPs with sine and ReLU activations on modular addition under three regimes: (i) underparameterized, (ii) overparameterized—both defined by the number of parameters relative to the training set size—and (iii) out-of-domain regime that tests extrapolation to sequence lengths unseen during training, *i.e.*, length extrapolation. Full setup details and additional figures are provided in App. C.

### 7.1 UNDERPARAMETERIZED REGIME.

We evaluate our sample complexity predictions by training matched architectures that differ only in their nonlinearity (sine vs. ReLU), using AdamW with zero weight decay on identical datasets and with identical optimization hyperparameters (Fig. 1).

**Sine networks are consistently more sample efficient than ReLU networks.** Across widths and training sizes, sine networks consistently outperform ReLU in both training and test accuracy, attaining a given accuracy at substantially smaller widths. For a fixed training set size, reducing the width—provided it remains sufficient for optimization—improves test accuracy for both activations, consistent with our uniform convergence guarantee in Sec. 5.

### 7.2 OVERPARAMETERIZED REGIME.

To validate the margin-based bounds in Sec. 6, we train wide two-layer MLPs with Muon[4] and sweep over decoupled weight decay rates. For sine models we apply weight decay only to the second layer; for ReLU we decay both layers. We report the 0.5th-percentile rather than the minimum margin because the latter is often dominated by rare outliers; a small quantile yields a stable large-margin proxy and, by Cor. J.4, only adds an additive $0.5\%$ term to the population error. We log training and test accuracies, the 0.5th-percentile of the training margin $\gamma_{\text{train}}^{0.5\%} := \text{Percentile}_{0.5} \left\{ \gamma_\theta(x^{(i)}, y^{(i)}) \right\}_{i=1}^n$, together with normalized margins that factor out layer scales: ReLU: $\widehat{\gamma}_{\text{ReLU}} = \frac{\gamma_{\text{train}}^{0.5\%}}{\|V\|_2 \|W\|_F}$, Sine: $\widehat{\gamma}_{\sin} = \frac{\gamma_{\text{train}}^{0.5\%}}{\|V\|_{1,\infty}}$.

**Normalized margins track generalization in the overparameterized regime.** Figs. 2 and 3 show that, as weight decay increases through a moderate range, normalized margins grow and test accuracy improves; with excessively large decay, training accuracy falls and generalization degrades. These trends align with the prediction that, in the overparameterized regime, generalization is governed by effective layer scales and margins.

### 7.3 OUT-OF-DOMAIN (OOD) REGIME (LENGTH GENERALIZATION).

We study length generalization, i.e., the ability to generalize to test datasets with sequence lengths unseen during training. The training sequence length $m$ is sampled uniformly from $\{2, 3, 4, 5, 7, 13, 19\}$, and we report the population accuracy of the trained model on a uniform distribution over data of fixed lengths, for lengths up to 811.

**Sine networks achieve near-perfect length generalization, while ReLU networks struggle in-distribution.** We compare the length generalization capability of MLPs with sine and ReLU activations in Fig. 4. Once the data budget exceeds a threshold, sine MLPs achieve perfect accuracy

---

[4]With decoupled weight decay, Muon's induced spectral geometry upper bounds $\|V\|_2\|W\|_F$ and $\|V\|_{1,\infty}$ up to dimension-dependent constants; see App. C.

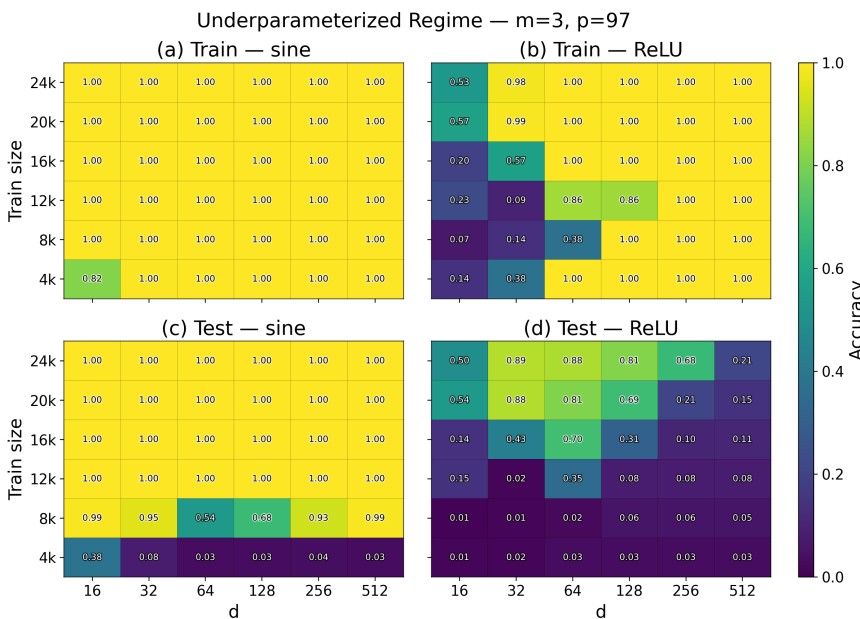

Figure 1: Accuracies for two-layer sine and ReLU MLPs in the underparameterized regime.

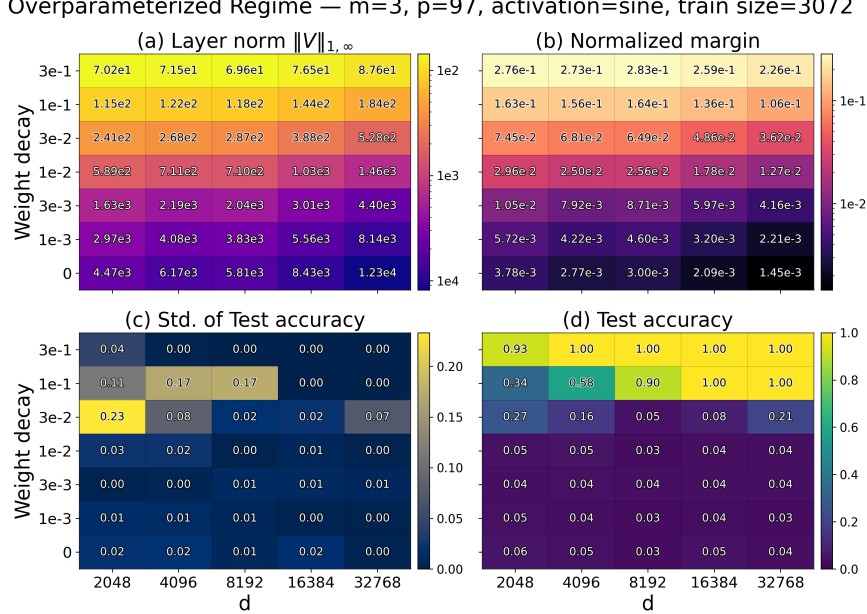

Figure 2: Two-layer sine networks in the overparameterized regime. Clockwise from top left: layer norm, normalized margin, test accuracy, and standard deviation of test accuracy.

on all seen lengths and remain essentially perfect on much longer unseen lengths. In contrast, ReLU MLPs struggle even in-domain and quickly degrade to chance-level accuracy on longer OOD lengths. These behaviors align closely with our expressivity results in Sec. 4. Thm. 4.2 shows that a width-2 sine network without bias can learn modular addition uniformly over all sequence lengths with high accuracy, implying that sine MLPs can map the shared embedding to a periodic representation of the modular sum. Conversely, Thm. 4.3 implies that the required ReLU width must grow linearly with $m$ to interpolate, and Thm. 4.4 shows that no fixed-width ReLU network can exactly match the ground truth at two incongruent lengths, indicating that ReLU MLPs admit no comparably simple periodic parametrization.

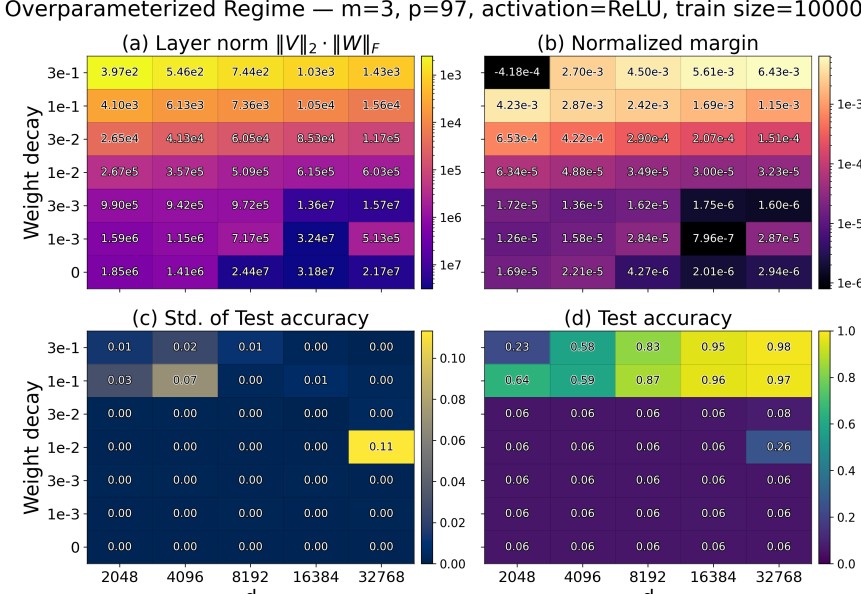

Figure 3: Two-layer ReLU networks in the overparameterized regime (panels as in Fig. 2).

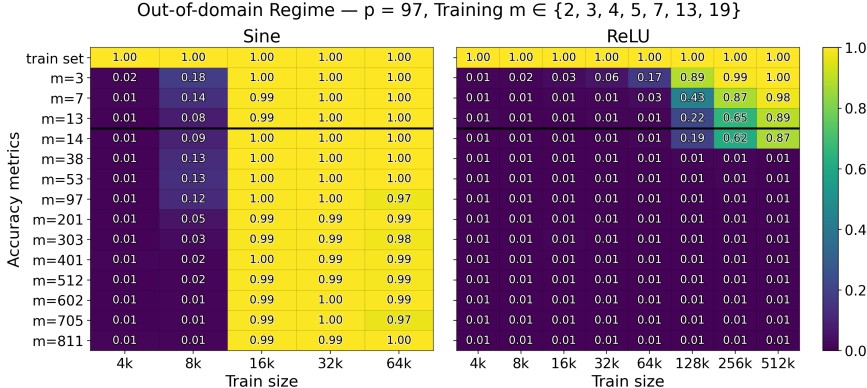

Figure 4: Out-of-domain accuracies of two-layer sine and ReLU MLPs, with no bias; each heatmap cell reports accuracy under the Best-over-WD scheme.

**Bias improves the robustness of sine MLP length generalization.** Figure 5 compares sine MLPs with and without a first-layer bias. Empirically, adding the bias preserves the excellent in-distribution performance while making length generalization more robust: accuracy remains high over a wider range of OOD lengths and weight-decay values. This phenomenon is naturally connected to our expressivity results. Thm. 4.2 shows that the bias effectively allows sines with different phases, and thus a richer periodic basis that implicitly includes cosine-like components, which makes it easier for the network to implement a modular rule that remains consistent across many sequence lengths. This additional expressivity provides a plausible explanation for the more stable and robust length generalization observed in biased sine MLPs.

### 7.4 TRANSFORMER BASELINE.

To verify that our observations are not specific to MLPs, we additionally train a 1-layer, 1-head decoder-only Transformer on modular addition and vary the feed-forward activation among *sine*, *ReLU*, and *GELU*.

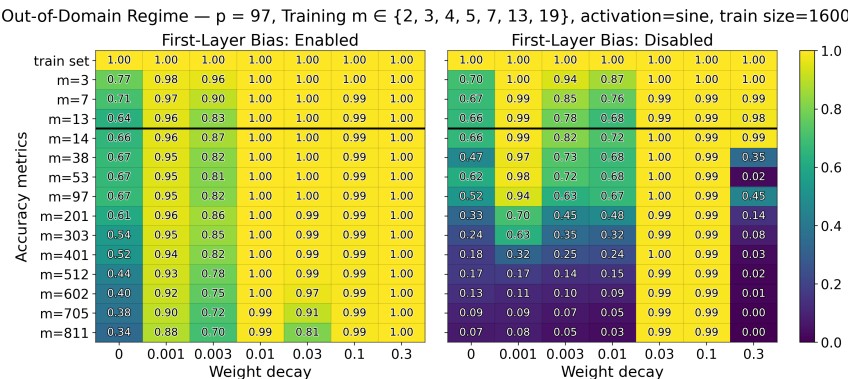

Figure 5: Out-of-domain accuracies for two-layer sine MLPs with and without first-layer bias.

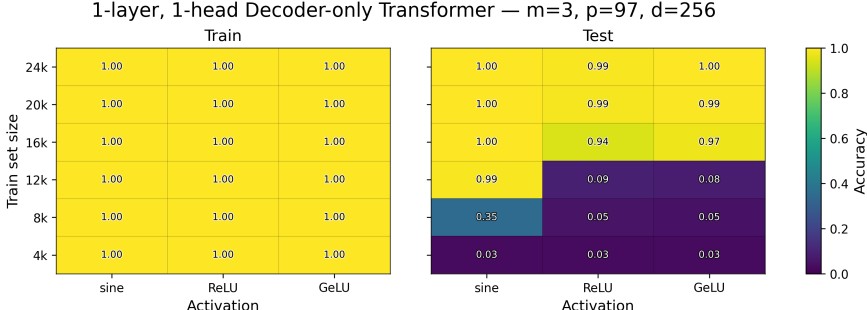

Figure 6: Train and test accuracy on modular addition for a 1-layer, 1-head decoder-only Transformer with FFN activations *sine*, *ReLU*, and *GELU*.

**Sine activation also improves sample efficiency in Transformers.** Fig. 6 reports train and test accuracies as a function of training-set size. Consistent with the MLP results, the sine activation attains high test accuracy with substantially fewer samples, while ReLU and GELU require more data and remain notably worse at moderate data budgets. This corroborates our theory across architectures: when optimization succeeds, the sine nonlinearity yields solutions that generalize more reliably on modular addition.

## 8 CONCLUSION

We establish that periodic activations offer provable advantages for learning modular addition. Our analysis reveals a sharp expressivity gap under a shared, position-independent embedding: width-2 sine MLPs suffice for exact modular addition (Thms. 4.1, 4.2), whereas ReLU networks require width scaling linearly with sequence length and cannot generalize across incongruent lengths (Thms. 4.3, 4.4). We complement this with statistical guarantees, deriving $\widetilde{\Theta}(dp)$ uniform convergence bounds (Thm. 5.6); specialized to sine with constant width, any interpolating ERM learner achieves nearly optimal $\widetilde{\mathcal{O}}(p)$ sample complexity (Thm. 5.8). In the overparameterized regime, we establish width-independent margin bounds (Thm. 6.2). Empirically, sine networks outperform ReLU models in sample efficiency (Fig. 1), and their superior test accuracy tracks with larger normalized margins in the overparameterized regime (Figs. 2–3). In the OOD regime, sine MLPs generalize far beyond training lengths, while ReLU networks degrade to chance (Figs. 4, 5). These benefits extend to Transformer architectures, where sine activations yield significantly better sample efficiency than standard ReLU and GELU baselines (Fig. 6). Together, our results support a robust design principle: encoding periodicity directly into the architecture maximizes both expressivity and learnability for periodic tasks.

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

## A  DISCLOSURE OF LARGE LANGUAGE MODEL USAGE

**Tool and scope.** We used Gemini 2.5 Pro, Gemini 3.0 Pro, and GPT-5 (high) as general-purpose assist tools for (i) code assistance (e.g., suggesting small snippets, refactoring, debugging hints, writing docstrings/comments, and drafting unit-test scaffolds) and (ii) writing assistance (e.g., copy-editing, grammar/fluency improvements, and localized rephrasing for clarity). Prompts sometimes included short excerpts of our own draft text or code necessary to request the above assistance.

**What the LLM did not do.** The LLM did not originate the paper's core research ideas, hypotheses, methodological designs, experimental protocols, analyses, or conclusions; it did not write sections containing novel scientific claims; and it did not determine which results to report or how to interpret them.

**Verification and oversight.** All LLM-suggested text and code were independently reviewed and edited by the authors. For code, the authors carefully checked and verified correctness (including running and testing LLM-suggested snippets before inclusion). Any factual statements in edited prose were cross-checked by the authors against our own results or appropriate sources. No LLM outputs were accepted without human scrutiny.

**Assessment of significance.** While the LLM provided editing assistance and code-level suggestions, its role does not rise to the level of a contributing author under the ICLR policy. The intellectual contributions (problem formulation, algorithmic design, experiments, and interpretation) are those of the human authors.

**Reproducibility note.** LLM assistance was limited to improving clarity and developer ergonomics; it does not affect the reproducibility of our methods or results. All final code and experiments are authored, verified, and maintained by the authors.

## B  ADDITIONAL RELATED WORK

**Mechanistic and optimization-centric perspectives.** Mechanistic studies catalog multiple circuit families that realize modular addition (Nanda et al., 2023; Zhong et al., 2023). Analyses tie the features that emerge during training to margin maximization and frequency selection (Morwani

et al., 2024; Li et al., 2025), and show that optimizer choice and regularization can change the time-to-grok and failure modes (Thilak et al., 2022; Abbe et al., 2023). Our formulation codifies these observations by hard-wiring a periodic inductive bias in a two-layer MLP and then proving expressivity and generalization benefits.

**Capacity results beyond the main text.** Lower bounds based on bit-extraction and upper bounds via growth-function arguments together yield near-matching VC estimates for piecewise-linear networks (Bartlett et al., 2017b). For piecewise-polynomial activations, classical results give $\mathcal{O}(WL^2 + WL \log W)$ upper bounds with refined scaling via unit counts (Anthony & Bartlett, 2009; Bartlett et al., 2017b). For Pfaffian activations (e.g., sigmoid, $\tanh$), capacity is polynomial in the parameter size (Karpinski & Macintyre, 1997; Anthony & Bartlett, 2009). These follow the general line of bounding polynomial sign patterns (Warren, 1968; Goldberg & Jerrum, 1995; Anthony & Bartlett, 2009), and in multiclass settings are summarized by the Natarajan dimension (Natarajan, 1989; Haussler & Long, 1995; Shalev-Shwartz & Ben-David, 2014).

**Learning parity and modular structure with gradient methods.** Fourier characters (parities) drive SQ hardness: under uniform inputs, even weak learning of related classes is impossible in the SQ model (O'Donnell, 2014; Blum et al., 1994; Kearns, 1998; Reyzin, 2020). With noise, LPN stays difficult—BKW is sub-exponential (Blum et al., 2003). Gradient descent on shallow nets emphasizes low-degree components, matching SQ lower bounds (Vempala & Wilmes, 2019); training time under SGD relates to leap complexity (Abbe et al., 2023), and fixed large-support parities can be provably hard (Shoshani & Shamir, 2025). In contrast, minibatch SGD can efficiently learn quadratic XOR via a find-then-tune phase with near-optimal samples (Glasgow, 2023).

**Implicit bias of optimizers.** Optimization geometry induces solution selection (Gunasekar et al., 2018). For AdamW, analyses identify convergence to KKT points of an $\ell_\infty$-constrained problem, implying an $\ell_\infty$ max-margin bias (Xie & Li, 2024); related max-margin behavior appears for Adam on separable data (Zhang et al., 2024). Muon has been argued to enforce effective spectral-norm control and to favor max-margin solutions in spectral geometry (Chen et al., 2025; Fan et al., 2025). These observations motivate the norm choices in our width-independent margin bounds.

**Margin-based generalization guarantees and path-norm viewpoints.** Beyond parameter counts, generalization can be controlled by margins and layerwise scales (Neyshabur et al., 2018b). Prior bounds include spectral and row-wise $L_{2,1}$ results for Lipschitz activations (Bartlett et al., 2017a), size-independent Rademacher bounds for positively homogeneous nets and, separately, $L_{1,\infty}$ constraints for Lipschitz activations (Golowich et al., 2017), and PAC–Bayesian variants robust to weight perturbations (Neyshabur et al., 2018a). Path norms yield rescaling-invariant capacity measures (Neyshabur et al., 2015b), connect to Barron-space approximation/estimation in two-layer models (E et al., 2022), underpin Path-SGD (Neyshabur et al., 2015a), and extend to modern DAG ReLU networks (Gonon et al., 2024).

**Gradient descent dynamics and empirical margins.** On separable data with cross-entropy, gradient descent drives norm growth while the predictor direction converges to the hard-margin SVM (Soudry et al., 2018). For positively homogeneous networks, gradient flow increases a layer-normalized margin and converges to a KKT point of a margin maximization problem (Lyu & Li, 2020); directional convergence and alignment extend to deep homogeneous settings (Ji & Telgarsky, 2020). In the mean-field limit for infinitely wide two-layer nets, the limiting classifier is max-margin in an appropriate function space (Chizat & Bach, 2020). For non-homogeneous networks, normalized margins still grow once the risk is small, with directional convergence to a KKT point (Cai et al., 2025). BatchNorm alters the bias, encouraging more uniform margins and faster directional convergence (Cao et al., 2023).

**Additional context on OOD generalization and length extrapolation.** Self-attention at fixed size has formal expressivity limits (Hahn, 2020), while under an idealized norm-regularized inference scheme, causal transformers can provably length-generalize for certain Limit Transformer functions (Huang et al., 2025). Practical mitigations include ALiBi/Hard-ALiBi (Press et al., 2022; Jelassi et al., 2024), prompting strategies that elicit multi-step reasoning (Wei et al., 2022), and pe-

riodic compression for long chains of thought within a fixed context window (e.g., PENCIL) (Yang et al., 2025).

# C  EXPERIMENTAL SETUP AND ADDITIONAL RESULTS

## C.1  EXPERIMENTAL SETUP

In this section, we explain the configuration used in all experiments.

**Data.** For each run we generate a static training set of size $n$ once and reshuffle it at the start of every epoch; the test set contains 10,000 i.i.d. samples from each $\mathcal{D}_m$.

**Initialization and reproducibility.** Unless otherwise specified, we run each hyperparameter configuration with three random seeds $\{1337, 1338, 1339\}$ and report metrics averaged over these runs. For each run, we generate static training and test input sequences i.i.d. with `torch.randint` and initialize all weights i.i.d. from $\mathcal{N}(0, 0.01^2)$. This setup ensures reproducible initializations and, within each sweep over training set sizes at fixed $(m, p)$, that smaller training sets are strict subsets of larger ones.

**Precision and implementation.** We implement all experiments in PyTorch, with TF32 disabled and `float32` precision throughout. Metrics are logged with Weights & Biases. Each run uses a single NVIDIA GPU (RTX A4000 or A6000, RTX 5000 or 6000 Ada, RTX Pro 6000 Blackwell Max-Q, L40S, A100, H100, or H200).

**Training.** We use mini-batch training with a fixed batch size of 1024. Models are trained for up to 300,000 epochs, and we report metrics at the final checkpoint.

### C.1.1  UNDERPARAMETERIZED REGIME.

Unless otherwise specified, we use AdamW with a constant learning rate of $10^{-3}$ and *zero* weight decay. All other AdamW hyperparameters are left at their PyTorch defaults (betas $(0.9, 0.999)$, $\varepsilon = 10^{-8}$). We do not use learning rate schedules, warmup, or gradient clipping. When comparing activations, we match $(m, p, d, n)$ and optimizer settings. We report train and test accuracy. We also evaluate vanilla SGD (learning rate $0.1$, momentum $0$, dampening $0$, weight decay $0$, Nesterov disabled) in Fig. 9.

### C.1.2  OVERPARAMETERIZED REGIME.

We use Muon with a constant learning rate of $10^{-3}$ and vary the decoupled weight decay. Momentum, Nesterov momentum, and Newton–Schulz steps are left at the library defaults (momentum $0.95$, Nesterov enabled, 5 Newton–Schulz steps). We do not use learning rate schedules, warmup, or gradient clipping. We report train and test accuracy and the 0.5th-percentile of the training margin, $\gamma_{\text{train}}^{0.5\%}$. We report the layer norms $\|W\|_F$ and $\|V\|_2$ for ReLU models, and $\|V\|_{1,\infty}$ for sine models, enabling computation of the normalized margins used in Sec. 7.

**Optimizer choice (overparameterized regime).** We adopt Muon to match the norm-based denominators used in our normalized margins. For any $A \in \mathbb{R}^{m \times n}$,

$$\|A\|_F \leq \sqrt{\text{rank}(A)}\,\|A\|_2 \leq \sqrt{\min\{m,n\}}\,\|A\|_2, \quad \|A\|_{1,\infty} := \max_i \|A_{i,:}\|_1 \leq \sqrt{n}\,\max_i \|A_{i,:}\|_2 \leq \sqrt{n}\,\|A\|_2.$$

Hence $\|V\|_2\|W\|_F$ and $\|V\|_{1,\infty}$ are controlled, up to explicit dimension factors, under the spectral geometry induced by Muon with decoupled weight decay.

### C.1.3  OUT-OF-DOMAIN (OOD) REGIME.

We fix the network width $d = 1024$ and train with the Muon optimizer at a constant learning rate of $10^{-3}$, and vary only the decoupled weight decay. We sweep decoupled weight decay over $\{0.3, 0.1, 0.03, 0.01, 0.003, 0.001, 0\}$, applying it to the second layer $V$ for sine MLPs and to both layers $V$ and $W$ for ReLU. When the first-layer bias is enabled for sine MLPs, we optimize the bias with AdamW (no weight decay) while keep Muon for all other parameters. Training uses a

static multi-length set with $m \in \{2, 3, 4, 5, 7, 13, 19\}$. For $p = 97$, the total sample budget is $n \in \{4\text{k}, 8\text{k}, 16\text{k}, 32\text{k}, 64\text{k}\}$; for $p = 53$, $n \in \{1\text{k}, 2\text{k}, 4\text{k}, 8\text{k}, 16\text{k}\}$. In each case, the budget is split as evenly as possible across the $m$ values, and each per-$m$ shard is reshuffled every epoch. For evaluation, we construct—once per $m$ and seed—fixed held-out test sets for OOD lengths $m \in \{14, 38, 53, 97, 201, 303, 401, 512, 602, 705, 811\}$. We also track training and in-domain test accuracy for $m \in \{3, 7, 13\}$. To ensure determinism and independence, we use independent CPU generators seeded with $\texttt{seed} \times 1009 + m$ for the training data of length $m$ and $\texttt{seed} \times 2009 + m$ for the corresponding test data; this yields per-$m$ test sets that are identical across epochs, prevents leakage via seed collisions, and ensures that, within each $m$, smaller training sets are strict prefixes of larger ones.

**Reporting conventions.** For some plots, we additionally use a *best-over-WD* scheme: for each accuracy metric, we compute the accuracy averaged over seeds for every weight decay value and report the maximum of these averages.

### C.1.4    TRANSFORMER ARCHITECTURE AND TRAINING DETAILS

**Task and tokenization.** For a given modulus $p$ and number of summands $m$, each example is a sequence of length $2m$: $x_1, +, x_2, +, \ldots, x_m, =$, where $x_i \in \{0, \ldots, p-1\}$. The vocabulary has size $p+2$ and includes two special symbols for "+" and "=". The model predicts the residue $(\sum_{i=1}^{m} x_i) \bmod p$ from the final position. We train with the standard cross-entropy loss on the last token's logits and report accuracy.

**Embeddings.** We use a single-layer *decoder-only* Transformer with one self-attention head. Tokens are embedded via a learned lookup table $\text{tok\_emb} \in \mathbb{R}^{(p+2) \times d}$ and added to a learned positional embedding $\text{pos\_emb} \in \mathbb{R}^{(2m) \times d}$. We use fixed $d = 256$ in Fig. 6.

$$H^{(0)} = E_{\text{tok}}(s_{1:L}) + E_{\text{pos}}(1{:}L) \in \mathbb{R}^{L \times d}.$$

**Layer normalization.** Position-wise LN over the feature dim $d$ with trainable $\gamma, \beta \in \mathbb{R}^d$ (init $\gamma = 1$, $\beta = 0$, $\varepsilon = 10^{-5}$):

$$\text{LN}(h) = \gamma \odot \frac{h - \mu(h)}{\sqrt{\text{Var}(h) + \varepsilon}} + \beta.$$

**Attention block.** We apply pre-norm LayerNorm, then a single-head causal self-attention with projections $Q = HW_Q$, $K = HW_K$, $V = HW_V$ ($W_Q, W_K, W_V \in \mathbb{R}^{d \times d}$, no biases). Attention logits are scaled by $1/\sqrt{d}$ and masked with a lower-triangular causal mask. The attention output is added to the residual stream.

$$\tilde{H} = \text{LN}(H^{(0)}), \; Q = \tilde{H}W_Q, \; K = \tilde{H}W_K, \; V = \tilde{H}W_V,$$

$$M_{ij} = \begin{cases} 0 & j \le i \\ -\infty & j > i \end{cases}, \quad P = \text{softmax}\left(\frac{QK^\top}{\sqrt{d}} + M\right), \quad H^{(1)} = H^{(0)} + PV.$$

**Feed-forward block.** After a second pre-norm LayerNorm, we apply a two-layer MLP with expansion factor 4, $\text{FFN}(h) = W_2 \phi(W_1 h)$, where $W_1 \in \mathbb{R}^{4d \times d}$, $W_2 \in \mathbb{R}^{d \times 4d}$. We sweep the activation $\phi \in \{\text{sine}, \text{ReLU}, \text{GELU}\}$. The FFN output is added to the residual stream, followed by a final LayerNorm.

$$\hat{H} = \text{LN}(H^{(1)}), \quad H^{(2)} = H^{(1)} + W_2 \phi(W_1 \hat{H}), \quad H^{(3)} = \text{LN}(H^{(2)}).$$

**Output layer.** A learned, untied linear head $W_{\text{out}} \in \mathbb{R}^{(p+2) \times d}$ maps hidden states to logits in $\mathbb{R}^{p+2}$; we evaluate the last position only. No dropout or label smoothing is used.

$$Z_{:,t} = W_{\text{out}} h_t^{(3)} \in \mathbb{R}^{p+2}, \qquad \hat{y} = \text{uargmax}_{c \in \{0, \ldots, p-1\}} Z_{c,L}.$$

**Overall model.** For a sequence $s_{1:L}$ with $L = 2m$ tokens drawn from a vocabulary of size $p+2$,

$$\text{TF}_\theta(s_{1:L}) = W_{\text{out}} \circ \text{LN} \circ \big(\text{Id} + (W_2 \circ \phi \circ W_1) \circ \text{LN}\big) \circ \big(\text{Id} + \text{Attn} \circ \text{LN}\big) \circ (\text{tok\_emb} + \text{pos\_emb})(s_{1:L}).$$

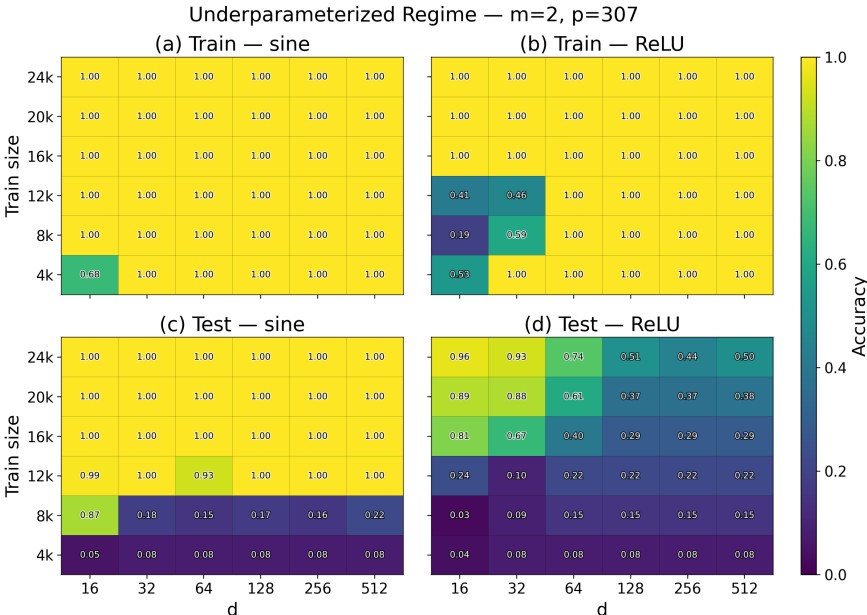

Figure 7: Underparameterized regime ($m = 2$, $p = 307$). Final train/test accuracies for two-layer MLPs with sine vs. ReLU activations under matched budgets.

**Initialization and precision.** All linear and embedding weights are initialized i.i.d. $\mathcal{N}(0, \sigma^2)$ with $\sigma = 10^{-2}$; LayerNorm scales are initialized to one and biases to zero. Training uses `float32` precision without mixed precision or TF32.

**Optimization and data.** We generate a *fixed* training set of size $n$ once per run and a fixed test set of 10,000 examples; the training set is reshuffled each epoch via index permutation. We use AdamW with constant learning rate, decoupled weight decay, batch size 1024, and gradient-norm clipping (1.0). We average metrics over seeds $\{1337, 1338, 1339\}$.

$$S_{\text{train}} = \{(s^{(j)}, y^{(j)})\}_{j=1}^{n}, \quad g \leftarrow \nabla_\theta \mathcal{L}, \quad g \leftarrow \text{clip}_{\|g\|_2 \leq 1.0}(g), \quad \theta \leftarrow \text{AdamW}(\theta, g; \eta, \lambda).$$

**Hyperparameters for Fig. 6.** We use $(m, p, d) = (3, 97, 256)$, train for a fixed number of epochs with AdamW (learning rate $10^{-4}$, weight decay 0.0, batch size 1024), and report train/test accuracy averaged over the above seeds. The x-axis of Fig. 6 orders FFN activations as *sine, ReLU,* and *GELU*; rows correspond to $n \in \{4k, 8k, 12k, 16k, 20k, 24k\}$.

## C.2 Additional results

We provide additional figures for our experiments.

Figs. 7 and 8 present multiple plots for the underparameterized sweeps at $(m, p) = (2, 307)$ and $(4, 53)$, respectively. In both cases, sine networks dominate ReLU networks for matched width and training budget, and the advantage widens as width decreases, until optimization begins to fail.

Fig. 9 reports underparameterized sweeps at $(m, p) = (3, 97)$ with vanilla SGD. On the test set, sine consistently outperforms ReLU at matched width and training budget, with performance peaking at small–moderate widths and degrading at large width despite perfect training accuracy. Compared with AdamW, the qualitative picture is unchanged, but SGD generalization is slower and more learning-rate sensitive. ReLU test accuracy under SGD closely matches ReLU under AdamW, whereas sine under SGD improves much more slowly after interpolation and never reaches the test accuracy of sine under AdamW.

Figs. 10–13 present multiple plots for the overparameterized sweeps at $(m, p) = (2, 307)$ and $(4, 53)$, respectively. In both cases, as weight decay increases through a moderate range, normalized margins grow and test accuracy improves; for excessively large decay, training accuracy falls

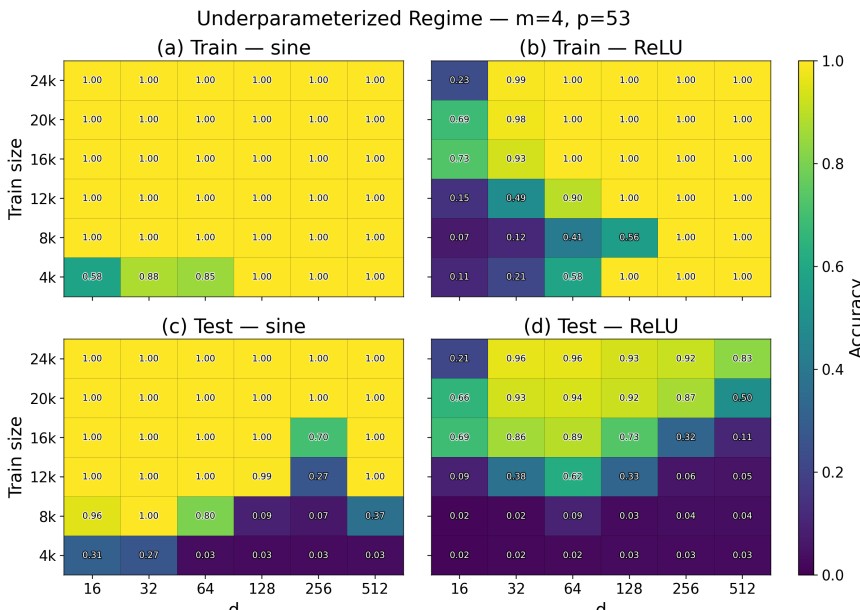

Figure 8: Underparameterized regime ($m = 4$, $p = 53$).

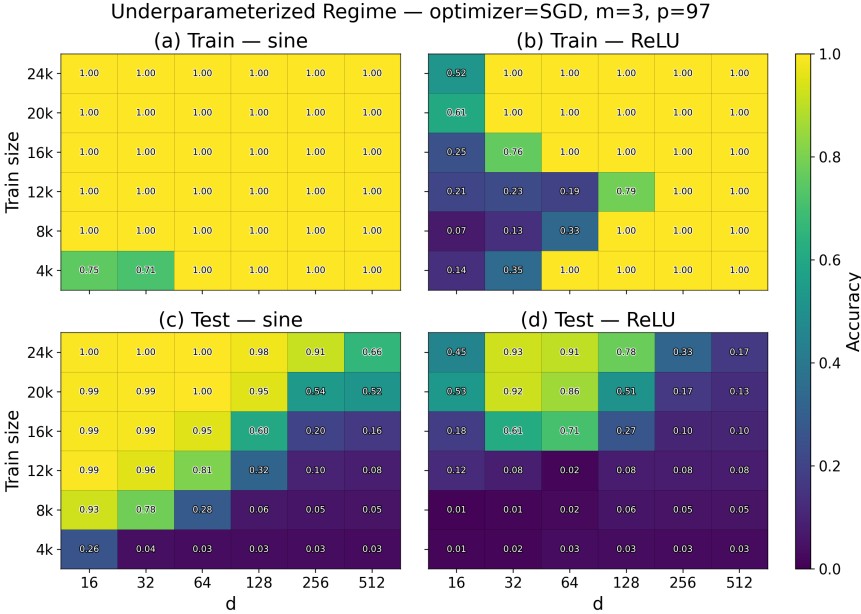

Figure 9: Underparameterized regime with Vanilla SGD ($m = 3$, $p = 97$).

and generalization degrades. These trends align with the prediction that, in the overparameterized regime, generalization is governed by effective layer scales and margins.

Fig. 14 provides additional plots for the out-of-domain sweep, including per-length accuracies across the weight-decay grid. Once the data budget reaches 8k examples, sine MLPs achieve perfect accuracy on all seen lengths and remain essentially perfect on unseen lengths far beyond the training support. In contrast, ReLU MLPs struggle even in-domain and quickly collapse to chance accuracy on larger OOD lengths.

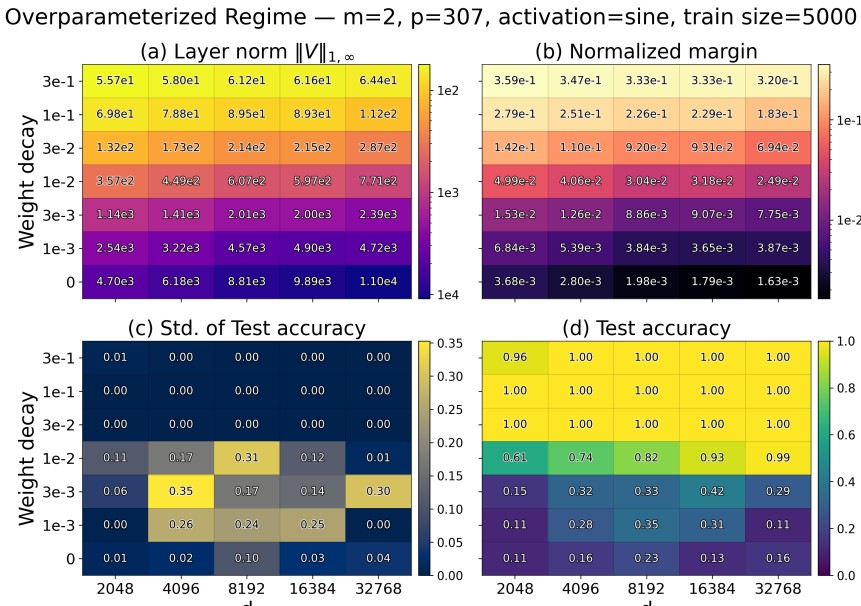

Figure 10: Two-layer sine networks in the overparameterized regime.

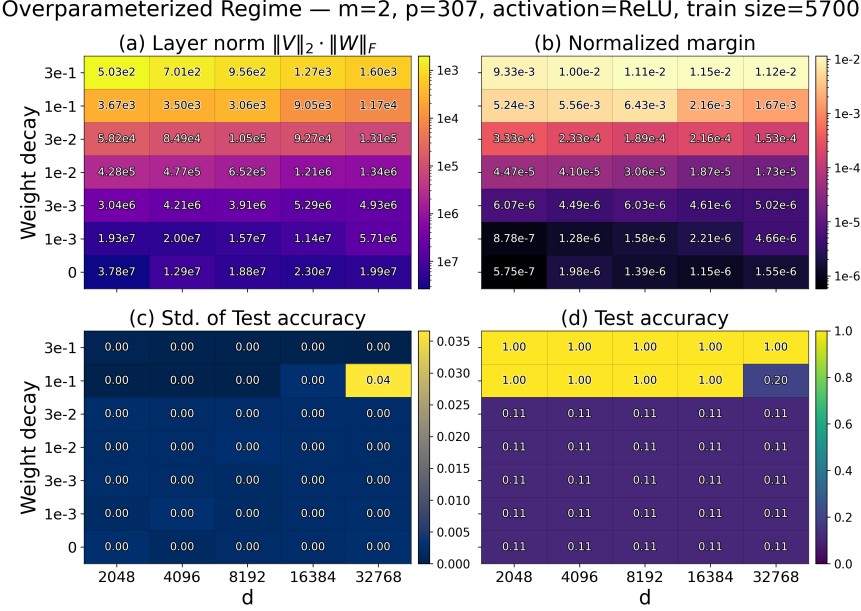

Figure 11: Two-layer ReLU networks in the overparameterized regime.

Fig. 15 shows additional plots comparing out-of-domain accuracies for two-layer sine MLPs with and without a first-layer bias. Enabling a first-layer bias in sine MLPs substantially improves robustness, leading to solutions that generalize stably and consistently.

# D  PROOF OUTLINES

This appendix outlines the logical structure of our theoretical results, clarifying the connections between the main text theorems and the detailed proofs in subsequent sections.

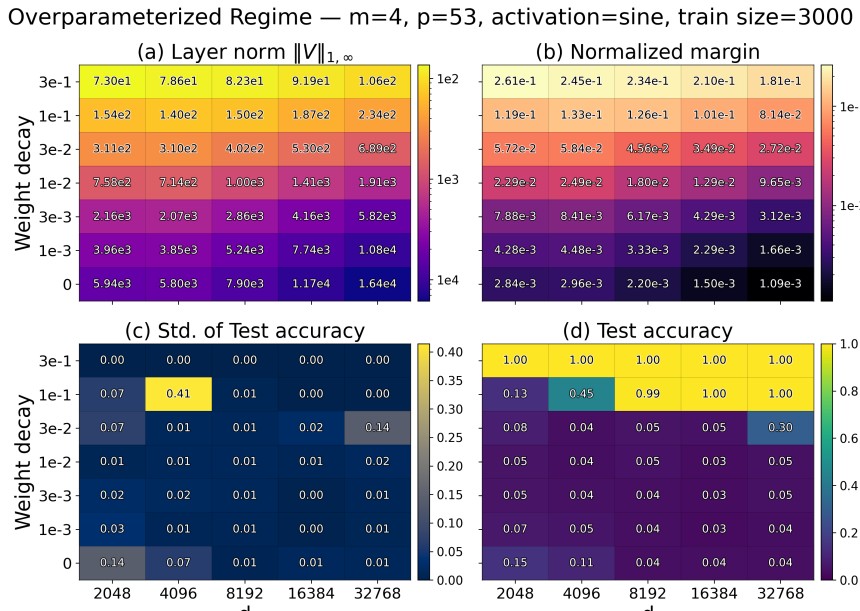

Figure 12: Two-layer sine networks in the overparameterized regime.

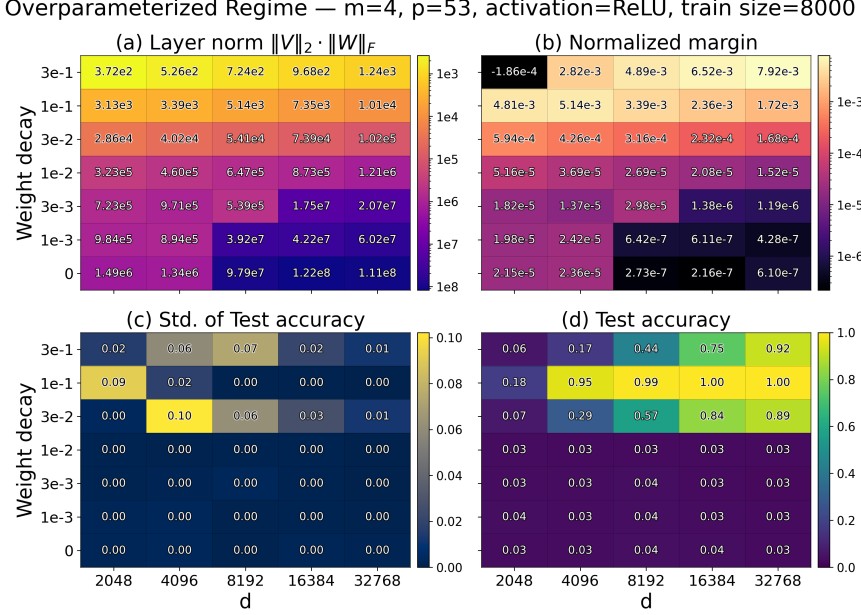

Figure 13: Two-layer ReLU networks in the overparameterized regime.

## D.1 EXPRESSIVITY (SEC. 4)

**Sine Networks (Thms. 4.1 and 4.2).** The proofs for sine networks are constructive. The core intuition is that the periodicity of the activation function naturally aligns with the modular arithmetic task.

1. **Encoding residues:** For an input $x \in \mathcal{X}_m$, the dot product $w^T x$ represents a sum of integers. By choosing weights proportional to frequencies $2\pi/p$, we ensure that inputs summing to $k$ and $k + p$ map to the same phase angle on the unit circle.

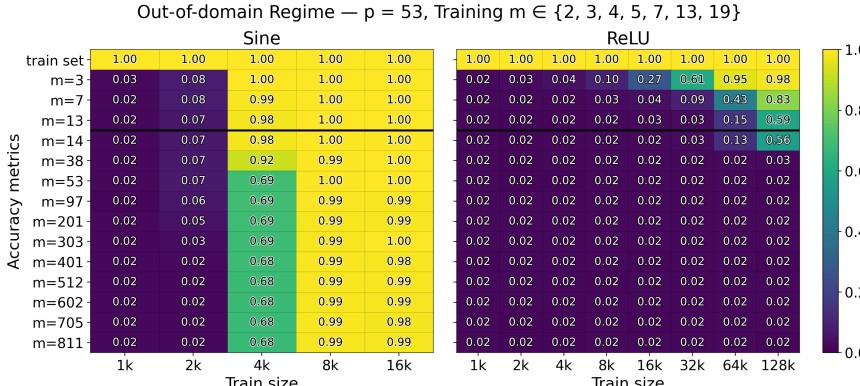

Figure 14: Out-of-domain accuracies of two-layer sine and ReLU MLPs, with no bias; each heatmap cell reports accuracy under the Best-over-WD scheme.

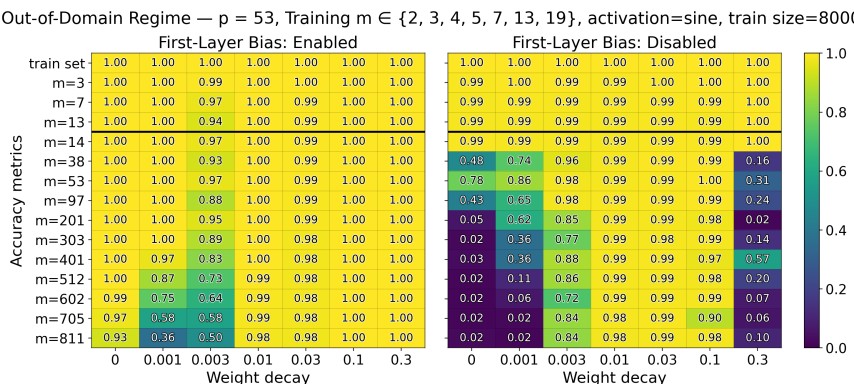

Figure 15: Out-of-domain accuracies for two-layer sine MLPs with and without first-layer bias.

2. **Decoding via orthogonality:** A single sine neuron captures the vertical projection of this phase. To fully distinguish $p$ classes, a width of 2 allows the network to compute both sine and cosine components, effectively implementing a Fourier basis that isolates specific residues.

3. **Uniformity:** For Thm. 4.2, we extend this construction to hold for all lengths $m$ simultaneously. With bias, we can realign the phases for any $m$. Without bias, we rely on a width-$\lfloor (p-1)/2 \rfloor$ construction that mimics a discrete Fourier transform to separate residue classes.

**ReLU Networks (Thms. 4.3 and 4.4).** The proofs for ReLU networks utilize geometric counting arguments and linear algebra.

1. **Counting linear regions:** A ReLU network partitions the input space into linear regions where the function is affine. To approximate the "sawtooth" function of modular addition exactly, the network must change slope (oscillate) $\Omega(m)$ times along specific directions. Thm. 4.3 establishes that a width-$d$ network cannot generate enough linear regions to match this complexity when $d$ is small relative to $m/p$.

2. **Incompatibility of lengths:** Thm. 4.4 demonstrates that the affine transformation required to fit length $m_1$ creates errors at length $m_2$ if $m_1 \not\equiv m_2 \pmod{p}$.

## D.2 UNDERPARAMETERIZED GENERALIZATION (SEC. 5)

**Uniform Convergence (Thm. 5.6).** The argument follows classical techniques (Warren, 1968; Goldberg & Jerrum, 1995; Anthony & Bartlett, 2009; Shalev-Shwartz & Ben-David, 2014). The proof of Thm. 5.6 proceeds via the following three-step reduction:

1. **Generalization Bound via Natarajan Dimension:** We first invoke the Multiclass Fundamental Theorem (Thm. G.18), which uniformly bounds the generalization gap by $\widetilde{\mathcal{O}}(\sqrt{d_N/n})$, where $d_N$ is the Natarajan dimension of the hypothesis class $\mathcal{H}$.

2. **Reduction to Pairwise VC-Dimension:** Directly computing $d_N$ is difficult. We utilize a reduction (Lem. G.6) which bounds the Natarajan dimension of a $p$-class model by the VC-dimension of its induced binary pairwise comparisons.

3. **Bounding VC-Dimension via Parameter Counting:** For the activations considered (trigonometric, piecewise-polynomial, rational-exponential), the pairwise difference function $f(x, \theta) = s_i^\theta(x) - s_j^\theta(x)$ is semi-algebraic (specifically, a polynomial or rational function of the parameters $\theta$). We apply classical bounds on the VC-dimension of polynomial function classes (Thm. G.11).

## D.3 OVERPARAMETERIZED GENERALIZATION (SEC. 6)

**Margin-based Bounds (Thms. 6.2 and 6.3).** These proofs show that all interpolating networks with large normalized margins generalize (Sec. J), and that at least one such network exists (Sec. I).

1. **Rademacher Generalization Bounds:** We first connect population loss with Rademacher complexity via standard generalization bounds (Thm. J.3).

2. **Ramp Loss Surrogate:** We treat the 0-1 multiclassification loss as upper bounded by the ramp loss (Cor. J.4).

3. **Vector Contraction:** We use the vector-contraction inequality for Rademacher complexity (Thm. 1 of Foster & Rakhlin 2019).

4. **Sine MLPs:** We bound the contracted complexity using the Dudley entropy integral (Lem. J.8) and covering numbers for sine networks (Lem. J.7), yielding a bound independent of width $d$.

5. **ReLU MLPs:** We apply a "peeling" argument tailored to positive homogeneous functions (Lem. 1 of Golowich et al. 2017; see Lem. J.12). This technique allows us to strip away the activation function and bound complexity using the spectral and Frobenius norms of the weight matrices, though the resulting bound depends on $m$ due to the complexity of our construction (Sec. I).

# E PROOFS IN EXPRESSIVITY

Throughout, denote the ground-truth label as

$$y(x) := \sum_{i=1}^{m} s_i \pmod{p} \in [p],$$

## E.1 LOW-WIDTH SINE CONSTRUCTION FOR A FIXED-LENGTH INPUT (THM. 4.1)

*Proof of Thm. 4.1.* Let $\phi := \frac{2\pi}{p}$. Define $W \in \mathbb{R}^{2 \times p}$ and $V \in \mathbb{R}^{p \times 2}$ by, for each $r \in [p]$ and $q \in [p]$,

$$W_{1,r} := (\phi r) \pmod{2\pi} \in [-\pi, \pi), \qquad W_{2,r} := \left(\phi r + \frac{\pi}{2m}\right) \pmod{2\pi} \in [-\pi, \pi),$$

$$V_{q,1} := \sin(\phi q), \qquad V_{q,2} := \cos(\phi q).$$

Adding integer multiples of $2\pi$ to coordinates of $W$ does not change $\sin(Wx)$ because $x \in \mathbb{Z}^p$. Therefore, for any $x \in \mathcal{X}_m$,

$$\sin\big((Wx)_1\big) = \sin\left(\phi \sum_{r=0}^{p-1} r x_r\right) = \sin(\phi y(x))$$

and, since $\|x\|_1 = m$,

$$\sin\big((Wx)_2\big) = \sin\left(\phi \sum_{r=0}^{p-1} r x_r + \frac{\pi}{2m} \sum_{r=0}^{p-1} x_r\right) = \cos(\phi y(x)).$$

Thus, for each $q \in [p]$,

$$
\begin{aligned}
s_q^\theta(x) &= V_{q,1} \sin(\phi y(x)) + V_{q,2} \cos(\phi y(x)) \\
&= \sin(\phi q) \sin(\phi y(x)) + \cos(\phi q) \cos(\phi y(x)) \\
&= \cos\big(\phi(y(x) - q)\big).
\end{aligned}
$$

Therefore $h_\theta(x) = \operatorname{uargmax}_{q \in [p]} s_q^\theta(x) = y(x)$.

The margin satisfies

$$\min_x \left(s_{y(x)}^\theta(x) - \max_{q \neq y(x)} s_q^\theta(x)\right) = 1 - \cos\left(\frac{2\pi}{p}\right) \geq \frac{8}{p^2},$$

using $1 - \cos t \geq \frac{2}{\pi^2} t^2$ for $t \in [0, \pi]$. Moreover, $\|W\|_\infty \leq \pi$, $\|V\|_\infty \leq 1$. $\qquad\square$

### E.2 A LENGTH-AGNOSTIC SINE NETWORK (THM. 4.2)

We will use two elementary lemmas.

**Lemma E.1** (Uniformity of the modular sum). *For fixed $m \in \mathbb{N}$, if $s_1, \ldots, s_m \overset{\text{i.i.d.}}{\sim} \operatorname{Unif}([p])$, then $y(x) \equiv \sum_{i=1}^m s_i \pmod{p}$ is uniform on $[p]$.*

*Proof.* Let $U \sim \operatorname{Unif}([p])$ and write $S_m := \sum_{i=1}^m s_i \pmod{p}$. For any $\omega := e^{2\pi i/p}$ and $t \in \{1, \ldots, p-1\}$, the discrete fourier transform of $S_m$ is

$$\mathbb{E}\,\omega^{t S_m} = \big(\mathbb{E}\,\omega^{tU}\big)^m = \left(\frac{1}{p} \sum_{u=0}^{p-1} \omega^{tu}\right)^m = 0.$$

Hence the discrete Fourier coefficients of $S_m$ vanish at all nonzero frequencies, and $S_m$ is uniform on $[p]$. $\qquad\square$

**Lemma E.2** (Sine Gram identity on $\mathbb{Z}/p\mathbb{Z}$). *Let $p \geq 2$ and $K := \lfloor (p-1)/2 \rfloor$. For $a, b \in [p]$ define*

$$S(a, b) := \sum_{k=1}^K \sin\left(\frac{2\pi k}{p} a\right) \sin\left(\frac{2\pi k}{p} b\right).$$

*Then:*

    *1. If $p$ is odd,*

$$S(a,b) = \begin{cases} \dfrac{p}{4}, & a \equiv b \not\equiv 0 \pmod{p}, \\[2mm] -\dfrac{p}{4}, & a \equiv -b \not\equiv 0 \pmod{p}, \\[2mm] 0, & \text{otherwise (in particular if } a = 0 \text{ or } b = 0). \end{cases}$$

    *2. If $p$ is even,*

$$S(a,b) = \begin{cases} \dfrac{p}{4}, & a \equiv b \not\equiv 0, \frac{p}{2} \pmod{p}, \\[2mm] -\dfrac{p}{4}, & a \equiv -b \not\equiv 0, \frac{p}{2} \pmod{p}, \\[2mm] 0, & \text{otherwise (in particular if } a \in \{0, \frac{p}{2}\} \text{ or } b \in \{0, \frac{p}{2}\}). \end{cases}$$

*Proof.* Using $\sin u \sin v = \frac{1}{2}\big(\cos(u-v) - \cos(u+v)\big)$,

$$S(a,b) = \frac{1}{2}\sum_{k=1}^{K}\cos\Big(\frac{2\pi k}{p}(a-b)\Big) - \frac{1}{2}\sum_{k=1}^{K}\cos\Big(\frac{2\pi k}{p}(a+b)\Big).$$

For odd $p$ we have $K = \frac{p-1}{2}$ and, for $u \in [p]$,

$$T_1(u) := \sum_{k=1}^{K}\cos\Big(\frac{2\pi k}{p}u\Big) = \begin{cases} \frac{p-1}{2}, & u \equiv 0, \\ -\frac{1}{2}, & u \not\equiv 0, \end{cases}$$

which follows by taking real parts in $\sum_{k=0}^{p-1} e^{2\pi i k u/p} = 0$ and pairing $k$ with $p-k$. Thus $S(a,b) = \frac{1}{2}\big(T_1(a-b) - T_1(a+b)\big)$, yielding the stated values.

For even $p$ we have $K = p/2 - 1$ and, for $u \in [p]$,

$$T_0(u) := \sum_{k=1}^{K}\cos\Big(\frac{2\pi k}{p}u\Big) = \frac{-1 - \cos(\pi u)}{2} = \begin{cases} -1, & u \text{ even but } u \not\equiv 0, \\ 0, & u \text{ odd}, \\ \frac{p}{2} - 1, & u \equiv 0, \end{cases}$$

using $1 + 2\sum_{k=1}^{p/2-1}\cos(2\pi k u/p) + \cos(\pi u) = 0$ for $u \not\equiv 0$. Hence $S(a,b) = \frac{1}{2}\big(T_0(a-b) - T_0(a+b)\big)$, which gives the stated cases. When $a \in \{0, p/2\}$ or $b \in \{0, p/2\}$, every summand vanishes (since $\sin(0) = \sin(\pi k) = 0$), so $S(a,b) = 0$. $\square$

*Proof of Thm. 4.2.*

**With bias.** Let $s^\theta(x) = V\sin(Wx + b)$ and set $\phi := \frac{2\pi}{p}$. Define $W \in \mathbb{R}^{2\times p}$ and $V \in \mathbb{R}^{p\times 2}$ by, for each $r, q \in \{0, \ldots, p-1\}$,

$$W_{1,r} := (\phi r) \pmod{2\pi} \in [-\pi, \pi), \qquad W_{2,r} := (\phi r) \pmod{2\pi} \in [-\pi, \pi),$$

$$V_{q,1} := \sin(\phi q), \qquad V_{q,2} := \cos(\phi q), \qquad b_1 = 0, \qquad b_2 = \frac{\pi}{2}.$$

The same calculation as in the proof of Thm. 4.1 shows that $s_\ell(x) = \cos\big(\phi(y(x) - \ell)\big)$. For the correct label $\ell = y(x)$, the score is 1. For any incorrect label $\ell \neq y(x)$, the score is strictly less than 1. Thus $h_\theta(x) = y(x)$ for every $m$.

**Without bias.** Let $s^\theta(x) = V\sin(Wx)$, $K := \lfloor (p-1)/2 \rfloor$, and $\alpha = (0, 1, \ldots, p-1)^\top \in \mathbb{Z}^p$. Define a width-$K$ sine network by

$$W \in \mathbb{R}^{K\times p}, \quad W_{k,:} = \frac{2\pi k}{p}\alpha^\top \ (k = 1, \ldots, K), \qquad V \in \mathbb{R}^{p\times K}, \quad V_{\ell k} = \sin\Big(\frac{2\pi k}{p}\ell\Big).$$

For any $x \in \mathcal{X}_m$, the $k$-th hidden unit is

$$\phi_k(x) = \sin\Big(\frac{2\pi k}{p}\langle\alpha, x\rangle\Big) = \sin\Big(\frac{2\pi k}{p}y(x)\Big),$$

so for $\ell \in [p]$,

$$s_\ell(x) = (V\phi(x))_\ell = \sum_{k=1}^{K}\sin\Big(\frac{2\pi k}{p}\ell\Big)\sin\Big(\frac{2\pi k}{p}y(x)\Big) = S\big(\ell, y(x)\big),$$

with $S$ from Lem. E.2.

If $p$ is odd, Lem. E.2 implies: for $y(x) = 0$, all scores are 0. Under the strict uniqueness rule, this counts as a tie (invalid), so the prediction is wrong. for $y(x) \neq 0$, one has $s_{y(x)}(x) = p/4$, which is strictly greater than $s_{-y(x)}(x) = -p/4$ and $s_\ell(x) = 0$ otherwise. Thus the max is unique and $h_\theta(x) = y(x)$. By Lem. E.1, $Y$ is uniform, so $\mathbb{P}[h_\theta(X) = Y] = 1 - \frac{1}{p}$ for every $m$.

If $p$ is even, Lem. E.2 gives: for $y(x) \in \{0, \frac{p}{2}\}$, all scores are 0, resulting in a tie (invalid prediction) for both residues. for all other residues, $s_{y(x)}(x) = p/4$ is the unique maximum (strictly greater than 0 and $-p/4$). By Lem. E.1, $Y \sim \text{Unif}([p])$, hence $\mathbb{P}[h_\theta(X) = Y] = 1 - \frac{2}{p}$ for every $m$. $\square$

### E.3 ReLU width lower bound (Thm. 4.3)

By a one-dimensional counting-path argument, we show that any ReLU MLP that exactly implements modular addition requires width $\Omega(m/p - 1)$.

*Proof of Thm. 4.3.* Consider the one-dimensional path of bags

$$x^{(s)} := (m - s)e_0 + se_1, \qquad s = 0, 1, \ldots, m,$$

along which the true label is $\ell(s) \equiv s \pmod p$. Let $y \in \mathbb{R}^d$ be the first column of $W$ and $z \in \mathbb{R}^d$ the difference between the second and first columns, $y_k = W_{k,1}$, $z_k = W_{k,2} - W_{k,1}$. Then the $k$-th preactivation is affine in $s$:

$$a_k(s) = [Wx^{(s)}]_k = (m - s)W_{k,1} + sW_{k,2} = m\,y_k + s\,z_k,$$

and the hidden unit $h_k(s) := \mathrm{ReLU}(a_k(s))$ is continuous piecewise-affine with at most one breakpoint at $s_k := -m\,y_k/z_k$ (if $z_k \neq 0$). Consequently, for each class $r \in [p]$ the score

$$f_r(s) := s_r^\theta(x^{(s)}) = \sum_{k=1}^{d} v_{r,k}\,h_k(s)$$

is a continuous piecewise-affine function whose breakpoint set $\mathcal{B} \subset [0, m]$ has cardinality at most $d$.

For $r \in [p]$ define the adjacent-class margin

$$g_r(s) := f_r(s) - f_{r\oplus 1}(s), \qquad r \oplus 1 = \begin{cases} r + 1, & r \leq p - 2, \\ 0, & r = p - 1. \end{cases}$$

Each $g_r$ is continuous piecewise-affine with breakpoints contained in $\mathcal{B}$. Sort $\mathcal{B}$ and note that $[0, m] \setminus \mathcal{B}$ has at most $d + 1$ connected components.

Write $I_s := [s, s + 1]$ for $s = 0, \ldots, m - 1$. Call $I_s$ *spoiled* if $I_s \cap \mathcal{B} \neq \emptyset$ and *clean* otherwise. A noninteger breakpoint lies in exactly one $I_s$, an integer breakpoint $j \in \{1, \ldots, m - 1\}$ lies in both $I_{j-1}$ and $I_j$, and endpoints $0$ or $m$ lie in exactly one $I_s$. Hence the number of spoiled $I_s$ is at most $2|\mathcal{B}| \leq 2d$, and therefore the number of clean $I_s$ is at least $m - 2d$.

Exact realization of the labels along the path requires that for every integer $s$, the correct class score is strictly greater than all others to avoid an invalid prediction ($\perp$). At step $s$, the label is $\ell(s)$, so we must have $f_{\ell(s)}(s) > f_{\ell(s)\oplus 1}(s)$, implying:

$$g_{\ell(s)}(s) > 0.$$

At step $s+1$, the label becomes $\ell(s+1) = \ell(s)\oplus 1$. Thus, we must have $f_{\ell(s)\oplus 1}(s+1) > f_{\ell(s)}(s+1)$, implying:

$$g_{\ell(s)}(s + 1) < 0.$$

If $I_s$ is clean, then $g_{\ell(s)}$ is affine on $I_s$. Since it is strictly positive at $s$ and strictly negative at $s+1$, it must have a *nontrivial* zero $t_s \in (s, s + 1)$ by the Intermediate Value Theorem. Because $I_s$ is clean, we have $t_s \in [0, m] \setminus \mathcal{B}$.

Moreover, a piecewise-affine $g_r$ can have at most one nontrivial zero in each connected component of $[0, m] \setminus \mathcal{B}$, hence at most $d + 1$ such zeros overall. Summing over $r \in [p]$ gives that the number of clean intervals satisfies

$$m - 2d \leq \sum_{r=0}^{p-1}(d + 1) = p(d + 1).$$

Rearranging yields

$$d \geq \frac{m - p}{p + 2} = \Omega\left(\frac{m}{p} - 1\right).$$

$\square$

### E.4 NO SIMULTANEOUS EXACT FIT FOR TWO LENGTHS WITH RELU (THM. 4.4)

*Proof of Thm. 4.4.* ReLU is positively 1-homogeneous: $\mathrm{ReLU}(\alpha z) = \alpha\,\mathrm{ReLU}(z)$ for all $\alpha \geq 0$. Thus, for any $\alpha > 0$ and any $x \in \mathbb{R}^p$,

$$s^\theta(\alpha x) = V\,\mathrm{ReLU}(W(\alpha x)) = \alpha\,V\,\mathrm{ReLU}(Wx) = \alpha\,s^\theta(x),$$

so scaling preserves the uargmax:

$$h_\theta(\alpha x) = h_\theta(x) \qquad \forall\,\alpha > 0. \tag{1}$$

Let $e_1 \in \mathbb{R}^p$ be the first basis vector and set

$$x^{(1)} := m_1 e_1 \in \mathcal{X}_{m_1}, \qquad x^{(2)} := m_2 e_1 \in \mathcal{X}_{m_2}.$$

Then $x^{(2)} = \frac{m_2}{m_1}\,x^{(1)}$, so by equation 1, $h_\theta(x^{(2)}) = h_\theta(x^{(1)})$. However,

$$y(x^{(1)}) \equiv m_1 \pmod{p}, \qquad y(x^{(2)}) \equiv m_2 \pmod{p},$$

and $m_1 \not\equiv m_2 \pmod{p}$ by assumption. Therefore at least one of $x^{(1)}$ or $x^{(2)}$ must be misclassified by any $\theta$, ruling out perfect accuracy on $\mathcal{X}_{m_1} \cup \mathcal{X}_{m_2}$. $\square$

## F  A PAC LOWER BOUND FOR LABEL-PERMUTATION-EQUIVARIANT LEARNER

Although we introduced specific MLPs, the lower bound below holds for any (possibly randomized) *label-permutation-equivariant* learner.

**Definition F.1** (Learning algorithm; Def. 3.1 in (Li et al., 2021a)). A (possibly randomized) supervised learning algorithm $\mathcal{A}$ maps a training sample $S = \{(x^{(i)}, y^{(i)})\}_{i=1}^n \in (\mathcal{X} \times \mathcal{Y})^n$ to a hypothesis $\widehat{h} = \mathcal{A}(S) : \mathcal{X} \to \mathcal{Y}$. For randomized $\mathcal{A}$, the output is a distribution over hypotheses; two randomized algorithms are considered the same if their induced output distributions on functions coincide for every input sample.

**Definition F.2** (Label-permutation equivariance). Let $\mathbb{S}(\mathcal{Y})$ be the group of permutations of the label space $\mathcal{Y}$. A learning algorithm $\mathcal{A}$ is *label-permutation equivariant* if, for every dataset $S = \{(x^{(i)}, y^{(i)})\}_{i=1}^n$ and every $\sigma \in \mathbb{S}(\mathcal{Y})$,

$$\mathcal{A}\Big(\{(x^{(i)}, \sigma(y^{(i)}))\}_{i=1}^n\Big) = \sigma \circ \mathcal{A}(S) \quad \text{as functions } \mathcal{X} \to \mathcal{Y}.$$

For randomized $\mathcal{A}$, equality is in distribution of the output functions.

*Remark* F.3 (Notable algorithms for label-permutation equivariance). Def. F.2 parallels (Li et al., 2021a, Def. 3.2) but with the group acting on labels rather than inputs. Their Appendix C criterion (Thm. C.1; Rem. C.1 for adaptive methods) applies verbatim to actions on output coordinates: with an i.i.d. final-layer initialization, the learner is label-permutation equivariant. Thus AdaGrad and Adam satisfy this. Since orthogonal equivariance strictly contains permutation equivariance, the same reasoning shows GD/SGD with momentum are label-permutation equivariant under i.i.d. final-layer initialization (Li et al., 2021a, Table 1).

The ground-truth labeling function is

$$f(x) = \sum_{k=0}^{p-1} k\,x_k \pmod{p} \in [p].$$

A learner that already knows this rule requires essentially no data, since it can compute $f(x)$ exactly from $x$.

We capture label symmetry by requiring that the learner be label-permutation–equivariant. For the analysis, we use the following standard symmetrization device.

**Lemma F.4** (Equivariant symmetrization). *Fix a (possibly randomized) label-permutation-equivariant algorithm $\mathcal{A}$ and a realized sample $S = \{(X_i, f(X_i))\}_{i=1}^n$. For any $\sigma \in \mathbb{S}([p])$ define the symmetrized output*

$$\widehat{f}_\sigma \ := \ \sigma^{-1} \circ \mathcal{A}\big(\{(X_i, \sigma(f(X_i)))\}_{i=1}^n\big).$$

*Then for deterministic $\mathcal{A}$ one has $\widehat{f}_\sigma = \mathcal{A}(S)$ for every $\sigma$; for randomized $\mathcal{A}$, $\widehat{f}_\sigma$ has the same distribution as $\mathcal{A}(S)$. Consequently, for any event $\mathcal{E}$ depending on the learned function,*

$$\mathbb{P}\big[\mathcal{E}(\mathcal{A}(S))\big] \ = \ \mathbb{P}\big[\mathcal{E}(\widehat{f}_\Sigma)\big] \quad \text{when } \Sigma \sim \mathrm{Unif}(\mathbb{S}_p) \text{ is independent of } S.$$

*Proof.* Deterministic case: by equivariance $\mathcal{A}(\sigma(S)) = \sigma \circ \mathcal{A}(S)$, hence $\sigma^{-1} \circ \mathcal{A}(\sigma(S)) = \mathcal{A}(S)$.

Randomized case: for each fixed $\sigma$, label-permutation equivariance implies $\mathcal{A}(\sigma(S)) \stackrel{d}{=} \sigma \circ \mathcal{A}(S)$. Therefore $\sigma^{-1} \circ \mathcal{A}(\sigma(S)) \stackrel{d}{=} \mathcal{A}(S)$. With $\Sigma$ independent of $S$, $\widehat{f}_\Sigma$ has the same distribution as $\mathcal{A}(S)$. $\square$

We will therefore analyze $\widehat{f}_\Sigma$ for a uniform random permutation $\Sigma$, and, by Lem. F.4, this entails no loss of generality for the original (unsymmetrized) learner.

We measure performance by the population 0–1 risk against the canonical rule $f$,

$$L(\widehat{f}) = \mathbb{P}_X\big[\widehat{f}(X) \neq f(X)\big],$$

where $X \sim \mathcal{D}_X$ is an independent test draw and $\widehat{f}$ is random due to $S, \Sigma$, and any internal randomness of $\mathcal{A}$.

**Lemma F.5.** *If $X$ is generated as above, then $f(X) \sim \mathrm{Unif}([p])$. Hence $f(X_1), \ldots, f(X_n)$ are i.i.d. uniform on $[p]$.*

*Proof.* There exist random variables $s_1, \ldots, s_m \in [p]$ such that

$$X_k = \sum_{i=1}^m \mathbf{1}\{s_i = k\} \qquad (k = 0, \ldots, p-1).$$

Therefore

$$f(X) = \sum_{k=0}^{p-1} k\, X_k \equiv \sum_{i=1}^m s_i \pmod{p}.$$

Notice that $s_1 \sim \mathrm{Unif}([p])$ and $s_1 \perp (s_2, \ldots, s_m)$. Let $T := \sum_{i=2}^m s_i \pmod{p}$. Then $(s_1 + T) \bmod p$ is uniform on $[p]$, so $f(X) \sim \mathrm{Unif}([p])$. The i.i.d. statement follows from the i.i.d. draws of $X_i$. $\square$

**Lemma F.6.** *Fix any realized training set $S$. Let $R \subseteq [p]$ be the set of residues that appear among $f(X_1), \ldots, f(X_n)$, and let $U = [p] \setminus R$ with $K = |U|$. Conditional on $S$ and on the values $\{\Sigma(u) : u \in R\}$ revealed by the permuted sample $\Sigma(S)$, the restriction $\Sigma|_U$ is a uniformly random bijection from $U$ to $[p] \setminus \Sigma(R)$.*

*Proof.* $\Sigma$ is uniform over $\mathbb{S}_p$, independent of the data. Conditioning on $\{\Sigma(u) : u \in R\}$ leaves all completions of $\Sigma$ on $U$ equally likely. There are $K!$ such completions. $\square$

**Lemma F.7** (Risk lower bound via unseen residues). *Let $K$ be the number of unseen residues determined by $S$. For any (possibly randomized) label-permutation-equivariant learner, over the random draw,*

$$\mathbb{E}_\Sigma\left[L(\widehat{f}_\Sigma) \mid S\right] \ \geq \ \frac{K-1}{p}.$$

*Proof.* Condition on a realized $S$ and its unseen set $U$ (size $K$). By Lem. F.5, $\mathbb{P}\left[f(X) = u\right] = 1/p$ for each $u \in [p]$. For any unseen $u \in U$, Lem. F.6 implies $\Sigma(u)$ is uniform over a set of $K$ labels, independent of $X$ given $f(X) = u$. Thus, for any prediction rule measurable with respect to $S, \Sigma$,

and $X$, the success probability at residue $u$ is at most $1/K$, so the misclassification probability is at least $(K-1)/K$. Summing over $u \in U$,

$$\mathbb{E}_\Sigma \left[ L(\widehat{f}_\Sigma) \mid S \right] \geq \sum_{u \in U} \mathbb{P}\left[f(X) = u\right] \cdot \frac{K-1}{K} = \frac{K}{p} \cdot \frac{K-1}{K} = \frac{K-1}{p}.$$

$\square$

**Lemma F.8.** *Let $K$ be the number of residues in $[p]$ not hit by $f(X_1), \ldots, f(X_n)$. Then*

$$\mathbb{E}[K] = p\left(1 - \frac{1}{p}\right)^n,$$

$$\mathrm{Var}(K) \leq \mathbb{E}[K],$$

$$\mathbb{P}[K \leq \mathbb{E}[K] - t] \leq \exp\left(-\frac{2t^2}{n}\right) \quad \text{for all } t \geq 0.$$

*Proof.* Let $I_u = \mathbf{1}\{\text{residue } u \text{ is unseen}\}$ for $u \in [p]$. By Lem. F.5, the $n$ residues are i.i.d. uniform, so

$$\mathbb{P}\left[I_u = 1\right] = \left(1 - \frac{1}{p}\right)^n =: q, \qquad \mathbb{P}\left[I_u = I_v = 1\right] = \left(1 - \frac{2}{p}\right)^n =: q_2 \quad (u \neq v).$$

Thus $\mathbb{E}[K] = \sum_u \mathbb{E}[I_u] = pq$ and

$$\mathrm{Var}(K) = \sum_u \mathrm{Var}(I_u) + \sum_{u \neq v} \mathrm{Cov}(I_u, I_v) = pq(1-q) + p(p-1)(q_2 - q^2) \leq pq(1-q) \leq pq = \mathbb{E}[K],$$

since $(1 - 2/p)^n \leq (1 - 1/p)^{2n}$. For concentration, expose the independent residues $Z_i = f(X_i) \in [p]$. The mapping $(Z_1, \ldots, Z_n) \mapsto K$ is 1-Lipschitz (changing one residue can alter the number of unseen residues by at most 1), so McDiarmid's inequality yields the tail bound. $\square$

**Lemma F.9** (Logarithmic inequality). *For $x \in (0,1)$, $\log(1-x) \geq -\frac{x}{1-x}$. Hence, for integers $p \geq 2$ and all $n \geq 0$,*

$$\left(1 - \frac{1}{p}\right)^n \geq \exp\left(-\frac{n}{p-1}\right).$$

*Proof.* Define $h(x) = \log(1-x) + \frac{x}{1-x}$. Then $h'(x) = \frac{x}{(1-x)^2} \geq 0$ and $h(0) = 0$, so $h(x) \geq 0$ on $(0,1)$. $\square$

**Lemma F.10** (Permutation exposure martingale bound). *Let $U = \{u_1, \ldots, u_K\} \subseteq [p]$ and consider a real-valued function $G$ of the random restriction $\Sigma|_U$, where $\Sigma|_U : U \to [p] \setminus \Sigma(R)$ is a uniformly random bijection (conditional on $S$). Reveal $\Sigma(u_1), \ldots, \Sigma(u_K)$ sequentially and let $M_i = \mathbb{E}[G \mid \Sigma(u_1), \ldots, \Sigma(u_i)]$. If for all $i$ one has*

$$|M_i - M_{i-1}| \leq c_i,$$

*then for all $t \geq 0$,*

$$\mathbb{P}[G \leq \mathbb{E}[G] - t] \leq \exp\left(-\frac{2t^2}{\sum_{i=1}^K c_i^2}\right).$$

*Proof.* $(M_i)_{i \in \mathbb{N}}$ is a Doob martingale with bounded differences; apply Azuma–Hoeffding. $\square$

**Theorem F.11** (PAC lower bound for label-permutation-equivariant learner). *Fix $\varepsilon \in (0, \frac{1}{2})$ and $\delta \in (0, \frac{1}{2})$. There exists an integer $p_0 = p_0(\varepsilon, \delta)$ such that for all $p \geq p_0$, every label-permutation-equivariant learner $\mathcal{A}$ that (with probability at least $1 - \delta$ over the random draw of the training samples $S$ and any internal randomness) achieves population risk at most $\varepsilon$ against the canonical rule $f$ must use*

$$n \geq (p-1)\left(\log \frac{1}{\varepsilon} - 1\right) = \Omega(p).$$

*Equivalently, for every $n \leq (p-1)\big(\log\frac{1}{\varepsilon} - 1\big)$ and every such learner,*

$$\mathbb{P}\big[L(\widehat{f}) \leq \varepsilon\big] \; \leq \; \exp\Big(-\frac{(e-1)^2}{2\log(1/\varepsilon)}\,\varepsilon^2\,p\Big) \; + \; \exp(-c_1\,\varepsilon\,p) \; + \; \exp\Big(-\frac{c_2\,\varepsilon^2\,p}{\log(1/\varepsilon)}\Big), \quad (2)$$

*for absolute constants $c_1, c_2 > 0$. In particular, $\mathbb{P}\Big[L(\widehat{f}) \leq \varepsilon\Big] \leq \delta$ for all sufficiently large $p$.*

*Proof of Thm. F.11.* By Lem. F.4, it suffices to analyze $\widehat{f}_\Sigma$ with $\Sigma \sim \mathrm{Unif}(\mathbb{S}_p)$ independent of $S$, since $L(\widehat{f}_\Sigma)$ has the same distribution as $L(\mathcal{A}(S))$. Fix $S$ and let $U$ be the unseen residue set, $|U| = K$. Consider $G(\Sigma) = L(\widehat{f}_\Sigma)$ with $X \sim \mathcal{D}_X$ independent of $(S, \Sigma)$. Changing $\Sigma(u_i)$ can only swap at most two preimages, so it cannot affect the event $\widehat{f}_\Sigma(X) = f(X)$ unless $f(X)$ equals one of those two preimages, which has probability at most $2/p$ by Lem. F.5. Hence in Lem. F.10 we may take $c_i = 2/p$, so for all $t \geq 0$,

$$\mathbb{P}_\Sigma\Big[L(\widehat{f}_\Sigma) \; \leq \; \mathbb{E}_\Sigma[L \mid S] - t \;\Big|\; S\Big] \; \leq \; \exp\Big(-\frac{p^2 t^2}{2K}\Big). \quad (3)$$

Here and below, probabilities $\mathbb{P}_\Sigma[\cdot \mid S]$ are over the random restriction $\Sigma|_U$ (conditional on $S$), while $X \sim \mathcal{D}_X$ is independent of $(S, \Sigma)$.

By Lem. F.7, $\mathbb{E}_\Sigma[L \mid S] \geq (K-1)/p$. Set

$$t \; = \; \max\Big\{\frac{K-1}{p} - \varepsilon,\, 0\Big\}.$$

Plugging this into equation 3 yields the conditional bound

$$\mathbb{P}_\Sigma\big[L(\widehat{f}_\Sigma) \leq \varepsilon \mid S\big] \; \leq \; \mathbf{1}\{K \leq \varepsilon p + 1\} \; + \; \exp\Big(-\frac{\big(\max\{(K-1) - \varepsilon p, 0\}\big)^2}{2K}\Big). \quad (4)$$

Let $\mu = \mathbb{E}[K] = p(1 - \frac{1}{p})^n$. By Lem. F.9, for $n \leq (p-1)\big(\log\frac{1}{\varepsilon} - 1\big)$,

$$\mu \; \geq \; p\exp\Big(-\frac{n}{p-1}\Big) \; \geq \; p\exp\Big(-\log\frac{1}{\varepsilon} + 1\Big) \; = \; e\,\varepsilon p.$$

By Lem. F.8 (McDiarmid over the $n$ i.i.d. residues),

$$\mathbb{P}\big(K \leq \varepsilon p + 1\big) \; \leq \; \exp\Big(-\frac{2(\mu - \varepsilon p - 1)^2}{n}\Big). \quad (5)$$

Since $\mu \geq e\varepsilon p$, for all $p \geq \frac{2}{(e-1)\varepsilon}$ we have $\mu - \varepsilon p - 1 \geq \frac{e-1}{2}\varepsilon p$. Using also $n \leq p\log(1/\varepsilon)$ gives from equation 5

$$\mathbb{P}\big(K \leq \varepsilon p + 1\big) \; \leq \; \exp\Big(-\frac{(e-1)^2}{2\log(1/\varepsilon)}\,\varepsilon^2\,p\Big).$$

Split on $\{K \geq \mu/2\}$ vs. $\{K < \mu/2\}$. By Lem. F.8 with $t = \mu/2$,

$$\mathbb{P}\big[K < \mu/2\big] \; \leq \; \exp\Big(-\frac{\mu^2}{2n}\Big) \; \leq \; \exp\Big(-\frac{c_2\,\varepsilon^2\,p}{\log(1/\varepsilon)}\Big) \quad (6)$$

for a universal $c_2 > 0$, since $\mu \geq e\varepsilon p$ and $n \leq p\log(1/\varepsilon)$. On $\{K \geq \mu/2\}$ we have $K \geq \frac{e}{2}\varepsilon p$ and thus, for all sufficiently large $p$,

$$\frac{\big((K-1) - \varepsilon p\big)^2}{2K} \; \geq \; \frac{\big((\frac{e}{2} - 1)\varepsilon p - 1\big)^2}{e\,\varepsilon p} \; \geq \; c_1\,\varepsilon\,p$$

for a universal $c_1 > 0$. Hence

$$\mathbb{E}\Big[\exp\Big(-\frac{\big(\max\{(K-1) - \varepsilon p, 0\}\big)^2}{2K}\Big)\Big] \; \leq \; \exp(-c_1\,\varepsilon p) \; + \; \exp\Big(-\frac{c_2\,\varepsilon^2\,p}{\log(1/\varepsilon)}\Big). \quad (7)$$

Taking expectations in equation 4 and since $n \leq p \log(1/\varepsilon)$,

$$\mathbb{P}\big[L(\widehat{f}_\Sigma) \leq \varepsilon\big] \leq \exp\left(-\frac{((e-1)\varepsilon p)^2}{2\,n}\right) + \exp(-c_1\,\varepsilon p) + \exp\left(-\frac{c_2\,\varepsilon^2\,p}{\log(1/\varepsilon)}\right)$$

$$\leq \exp\left(-\frac{(e-1)^2}{2\,\log(1/\varepsilon)}\,\varepsilon^2\,p\right) + \exp(-c_1\,\varepsilon p) + \exp\left(-\frac{c_2\,\varepsilon^2\,p}{\log(1/\varepsilon)}\right).$$

Finally, by Lem. F.4, $L(\widehat{f}_\Sigma)$ has the same distribution as $L(\widehat{f})$, yielding equation 2. In particular, for any $\delta \in (0, \frac{1}{2})$, choosing $p \geq p_0(\varepsilon, \delta)$ sufficiently large makes the right-hand side at most $\delta$. This shows that if $\mathbb{P}\left[L(\widehat{f}) \leq \varepsilon\right] \geq 1 - \delta$ then necessarily $n > (p-1)(\log(1/\varepsilon) - 1)$, proving the sample-complexity lower bound $n = \Omega(p)$. $\qquad\square$

# G    PROOFS IN UNDERPARAMETERIZED DOMAIN

## G.1    UPPER BOUND OF NATARAJAN-DIMENSION

Let $\mathcal{X}$ be an instance space, let $p \in \mathbb{N}$ with $p \geq 2$, and let $[p] = \{1, \dots, p\}$ be the label set. Fix a domain $\mathcal{U}$ and a function class $\mathcal{F}$. For a finite $T \subseteq \mathcal{U}$, write $\mathcal{F}_{|T} := \{ f_{|T} : f \in \mathcal{F} \}$ for its restriction and $|\mathcal{F}_{|T}|$ for the number of distinct labelings on $T$ realized by $\mathcal{F}$.

**Definition G.1** (Growth function). Let $\mathcal{B} \subseteq \{-1, +1\}^{\mathcal{Z}}$ be a binary hypothesis class on a domain $\mathcal{Z}$. For $n \in \mathbb{N}$, the *growth function* of $\mathcal{B}$ is

$$\Pi_\mathcal{B}(n) := \max\big\{ \big|\mathcal{B}_{|T}\big| : T \subseteq \mathcal{Z}, \, |T| = n \big\}.$$

**Definition G.2** (Network class and pairwise reduction). Let $\mathcal{H}_\Theta \subseteq ([p] \cup \{\bot\})^{\mathcal{X}}$ be a network class realized by score vectors $s^\theta(x) = (s_1^\theta(x), \dots, s_p^\theta(x)) \in \mathbb{R}^p$, $\theta \in \Theta$, $x \in \mathcal{X}$. The predictor is defined by strict maximization:

$$h_\theta(x) = \operatorname{uargmax}_{\ell \in [p]} s_\ell^\theta(x) = \begin{cases} \ell, & \text{if } s_\ell^\theta(x) > s_k^\theta(x) \text{ for all } k \neq \ell, \\ \bot, & \text{otherwise,} \end{cases}$$

Define the *pairwise reduction* on the domain

$$\mathcal{Z}_{\mathrm{pair}} := \mathcal{X} \times \{(i, j) \in [p] \times [p] : i < j\}$$

by the binary reduction class $\mathcal{G}_\Theta \subseteq \{-1, +1\}^{\mathcal{Z}_{\mathrm{pair}}}$ for $\mathcal{H}_\Theta$ consisting of functions

$$g_\theta(x, i, j) = \operatorname{sgn}\big(s_i^\theta(x) - s_j^\theta(x)\big) = \begin{cases} +1, & \text{if } s_i^\theta(x) \geq s_j^\theta(x), \\ -1, & \text{if } s_i^\theta(x) < s_j^\theta(x). \end{cases}$$

**Definition G.3** (Number of realized multiclass labelings). For $S = \{x^{(1)}, \dots, x^{(n)}\} \subset \mathcal{X}$ and hypothesis class $\mathcal{H} \subseteq [p]^{\mathcal{X}}$, define

$$\Lambda_\mathcal{H}(S) := \big|\mathcal{H}_{|S}\big| = \left|\Big\{ \big(h(x^{(1)}), \dots, h(x^{(n)})\big) \in [p]^n : h \in \mathcal{H} \Big\}\right|.$$

The lemma below connects the Natarajan-dimension to the growth function of the reduction class, which is a key tool in this section.

**Lemma G.4** (Natarajan shattering and labelings). *If a finite set $S = \{x^{(1)}, \dots, x^{(n)}\} \subset \mathcal{X}$ is Natarajan-shattered by a hypothesis $\mathcal{H} \subset [p]^{\mathcal{X}}$, then $\Lambda_\mathcal{H}(S) \geq 2^n$.*

*Proof of Lem. G.4.* By Def. 5.2, there exist $f_1, f_2 \in [p]^S$ with $f_1(x) \neq f_2(x)$ for all $x \in S$ such that for every selector $b : S \to \{1, 2\}$ there is $h_b \in \mathcal{H}$ with $h_b(x) = f_{b(x)}(x)$ for all $x \in S$. Define $\Phi : \{1, 2\}^S \to \mathcal{H}_{|S}$ by $\Phi(b) = h_b|_S$.

If $b \neq b'$, pick $x_0 \in S$ with $b(x_0) \neq b'(x_0)$. Then

$$\Phi(b)(x_0) = h_b(x_0) = f_{b(x_0)}(x_0) \neq f_{b'(x_0)}(x_0) = h_{b'}(x_0) = \Phi(b')(x_0),$$

so $\Phi(b) \neq \Phi(b')$. Thus $\Phi$ is injective and

$$\Lambda_\mathcal{H}(S) = |\mathcal{H}_{|S}| \geq |\Phi(\{1, 2\}^S)| = |\{1, 2\}^S| = 2^{|S|} = 2^n.$$

$\square$

**Lemma G.5** (Labelings and pairwise reduction). *Fix $S = \{x^{(1)}, \ldots, x^{(n)}\} \subset \mathcal{X}$ and a hypothesis class $\mathcal{H}_\Theta \subseteq [p]^{\mathcal{X}}$. Then*

$$\Lambda_{\mathcal{H}_\Theta}(S) \leq \Pi_{\mathcal{G}_\Theta}\big(n\,p(p-1)/2\big).$$

*Proof of Lem. G.5.* Set

$$T := S \times \{(i,j) \in [p] \times [p] : i < j\} \subset \mathcal{Z}_{\mathrm{pair}}.$$

For each $h \in (\mathcal{H}_\Theta)_{|S}$ define the fiber

$$W(h) := \{\theta \in \Theta : h_{\theta|S} = h\}, \quad \text{and} \quad A(h) := \{g_{\theta|T} : \theta \in W(h)\} \subseteq (\mathcal{G}_\Theta)_{|T}.$$

Now we will show that if $h \neq h'$, then $A(h) \cap A(h') = \emptyset$.

Pick $x \in S$ with $h(x) = i$ and $h'(x) = j \neq i$. Without loss of generality, assume $i < j$. For any $\theta \in W(h)$, the predictor yields a valid output $i$, which implies strict maximality: $s_i^\theta(x) > s_k^\theta(x)$ for all $k \neq i$. In particular, $s_i^\theta(x) > s_j^\theta(x)$, hence $g_\theta(x, i, j) = +1$.

Conversely, for any $\theta' \in W(h')$, the predictor yields $j$, which implies $s_j^{\theta'}(x) > s_i^{\theta'}(x)$. Therefore, $s_i^{\theta'}(x) < s_j^{\theta'}(x)$, hence $g_{\theta'}(x, i, j) = -1$.

Thus every element of $A(h)$ has $+1$ and every element of $A(h')$ has $-1$ at the coordinate $(x, i, j) \in T$, so $A(h) \cap A(h') = \emptyset$.

Since $|(\mathcal{H}_\Theta)_{|S}| \leq p^n < \infty$ and each $A(h) \neq \emptyset$, fix an arbitrary choice function $\Psi$ selecting one element of $A(h)$ for each $h \in (\mathcal{H}_\Theta)_{|S}$. Then the map

$$\Psi : (\mathcal{H}_\Theta)_{|S} \longrightarrow (\mathcal{G}_\Theta)_{|T}, \qquad h \longmapsto \Psi(h)$$

is well-defined and injective. Therefore,

$$\Lambda_{\mathcal{H}_\Theta}(S) = |(\mathcal{H}_\Theta)_{|S}| \leq |(\mathcal{G}_\Theta)_{|T}| \leq \Pi_{\mathcal{G}_\Theta}(|T|) = \Pi_{\mathcal{G}_\Theta}\big(n\tbinom{p}{2}\big) = \Pi_{\mathcal{G}_\Theta}\big(n\,p(p-1)/2\big).$$

$\square$

Together with Lems. G.4 and G.5, we have Lem. G.6:

**Lemma G.6** (Natarajan shattering and the growth function). *If $S = \{x^{(1)}, \ldots, x^{(n)}\} \subset \mathcal{X}$ is Natarajan-shattered by a $p$-class network class $\mathcal{H}_\Theta$, then*

$$2^n \leq \Pi_{\mathcal{G}_\Theta}\big(n\,p(p-1)/2\big),$$

*where $\mathcal{G}_\Theta$ is the reduction class of $\mathcal{H}_\Theta$.*

**Definition G.7** (k-combination of $\mathrm{sgn}(\mathcal{F})$). Let $\mathcal{Z}$ be any domain and let $\mathcal{F} \subseteq \mathbb{R}^{\mathbb{R}^D \times \mathcal{Z}}$ be a class of real-valued functions of the form $(a, z) \mapsto f(a, z)$, with $a \in \mathbb{R}^D$ and $z \in \mathcal{Z}$. A binary class $\mathcal{H} \subseteq \{-1, +1\}^{\mathcal{Z}}$ is a *k-combination of* $\mathrm{sgn}(\mathcal{F})$ if there exist a Boolean map $g : \{-1, +1\}^k \to \{-1, +1\}$ and functions $f_1, \ldots, f_k \in \mathcal{F}$ such that for every $h \in \mathcal{H}$ there is $a \in \mathbb{R}^D$ with

$$h(z) = g\big(\mathrm{sgn}(f_1(a, z)), \ldots, \mathrm{sgn}(f_k(a, z))\big) \qquad \text{for all } z \in \mathcal{Z}.$$

We say $f \in \mathcal{F}$ is $C^D$ in its parameters if, for every fixed $z$, the map $a \mapsto f(a, z)$ is $C^D$.

**Definition G.8** (Regular zero-set intersections; Def. 7.3 (Anthony & Bartlett, 2009)). For differentiable $f_1, \ldots, f_k : \mathbb{R}^D \to \mathbb{R}$, the family $\{f_1, \ldots, f_k\}$ has *regular zero-set intersections* if for every nonempty $I \subseteq \{1, \ldots, k\}$, the Jacobian of $(f_i)_{i \in I}$ has full row rank $|I|$ at every $a$ with $f_i(a) = 0$ for all $i \in I$.

**Definition G.9** (Solution set components bound; Def. 7.5 (Anthony & Bartlett, 2009)). Let $\mathcal{G}$ be a set of real-valued functions on $\mathbb{R}^D$. We say $\mathcal{G}$ has *solution set components bound* $B$ if for any $1 \leq k \leq D$ and any $\{f_1, \ldots, f_k\} \subseteq \mathcal{G}$ that has regular zero-set intersections,

$$\mathrm{CC}\left(\bigcap_{i=1}^{k}\{a \in \mathbb{R}^D : f_i(a) = 0\}\right) \leq B,$$

where $\mathrm{CC}(\cdot)$ is the number of connected components.

**Theorem G.10** (General Growth function upper bound; Thm. 7.6 (Anthony & Bartlett, 2009))**.** *Let* $\mathcal{F} \subseteq \mathbb{R}^{\mathbb{R}^D \times \mathcal{Z}}$ *be closed under addition of constants, assume every* $f \in \mathcal{F}$ *is* $C^D$ *in* $a$, *and let*

$$\mathcal{G} := \big\{ a \mapsto f(a, z) : f \in \mathcal{F}, \ z \in \mathcal{Z} \big\}.$$

*If* $\mathcal{G}$ *has a solution set components bound* $B$ *and* $\mathcal{H} \subseteq \{-1, +1\}^{\mathcal{Z}}$ *is a* $k$-*combination of* $\mathrm{sgn}(\mathcal{F})$, *then for all* $N \geq D/k$,

$$\Pi_{\mathcal{H}}(N) \ \leq \ B \sum_{i=0}^{D} \binom{Nk}{i} \ \leq \ B \Big( \frac{e\,Nk}{D} \Big)^D.$$

**Theorem G.11** (General Growth function upper bound; Thm. 8.3 (Anthony & Bartlett, 2009))**.** *Let* $\mathcal{F} \subseteq \mathbb{R}^{\mathbb{R}^D \times \mathcal{Z}}$ *be a class of functions mapping from* $\mathbb{R}^D \times \mathcal{Z}$ *to* $\mathbb{R}$ *such that, for all* $z \in \mathcal{Z}$ *and* $f \in \mathcal{F}$, *the map* $a \mapsto f(a, z)$ *is a polynomial on* $\mathbb{R}^D$ *of degree at most* $r$. *Suppose that* $\mathcal{H}$ *is a* $k$-*combination of* $\mathrm{sgn}(\mathcal{F})$. *Then, if* $N \geq D/k$,

$$\Pi_{\mathcal{H}}(N) \ \leq \ 2 \left( \frac{2eNkr}{D} \right)^D.$$

*Remark* G.12. If $N < D/k$, we have a trivial bound $\Pi_{\mathcal{H}}(N) \leq 2^N < 2^{D/k} \leq 2^D$, so for all $N \in \mathbb{N}$, $\Pi_{\mathcal{H}}(N) \leq \max \left\{ 2^D, 2\left( \frac{2eNkr}{D} \right)^D \right\}$.

**Lemma G.13** (Absorbing $\log n$; Lem. A.2 (Shalev-Shwartz & Ben-David, 2014))**.** *Let* $A \geq 1$, $B \geq 0$, *and* $u > 0$. *If* $u < A \log u + B$, *then*

$$u \ < \ 4A \log\big(2A\big) \ + \ 2B.$$

**Lemma G.14** (Trigonometric Sum Polynomialization)**.** *Let* $p \geq 1$ *and* $m \geq 0$ *be integers. For any vector of non-negative integers* $x = (x_1, \ldots, x_p)$ *such that* $\sum_{v=1}^{p} x_v = m$, *there exist polynomials*

$$S_x, C_x \in \mathbb{Z}[c_1, s_1, \ldots, c_p, s_p]$$

*of total degree at most* $m$ *that satisfy*

$$S_x(c_1, s_1, \ldots, c_p, s_p) = \sin \Big( \sum_{v=1}^{p} x_v \alpha_v \Big) \quad and \quad C_x(c_1, s_1, \ldots, c_p, s_p) = \cos \Big( \sum_{v=1}^{p} x_v \alpha_v \Big)$$

*for all real angles* $\alpha_1, \ldots, \alpha_p$, *where* $c_v := \cos(\alpha_v)$ *and* $s_v := \sin(\alpha_v)$.

*Proof of Lem. G.14.* We prove the lemma by induction on $m = \sum_{v=1}^{p} x_v$. The uniqueness of the polynomials is guaranteed by the deterministic recursive construction.

Base Case ($m = 0$):
If $m = 0$, then $x = \mathbf{0}$ is the only possible vector. The sum of angles is $\sum x_v \alpha_v = 0$. The defined polynomials are $S_{\mathbf{0}} = 0$ and $C_{\mathbf{0}} = 1$. These are integer-coefficient polynomials of degree 0. They correctly evaluate to $\sin(0) = 0$ and $\cos(0) = 1$.

Inductive Step:
Assume the claim holds for all vectors $y$ with component sum $m - 1$. Let $x$ be a vector with component sum $m$. Let $u = \min\{v \mid x_v > 0\}$ and define $y = x - e_u$. The components of $y$ sum to $m - 1$. By the induction hypothesis, there exist polynomials $S_y$ and $C_y$ with integer coefficients and degree at most $m - 1$ that represent $\sin(\sum y_v \alpha_v)$ and $\cos(\sum y_v \alpha_v)$.

We define $S_x$ and $C_x$ as per the recursion:

$$S_x := s_u C_y + c_u S_y \qquad C_x := c_u C_y - s_u S_y$$

1. Coefficients and Degree: Since $S_y$ and $C_y$ have integer coefficients, and $s_u, c_u$ are variables, $S_x$ and $C_x$ are also polynomials with integer coefficients. Their total degrees are bounded by:

$$\deg(S_x) \leq 1 + \max(\deg(C_y), \deg(S_y)) \leq 1 + (m - 1) = m$$

The same bound holds for $\deg(C_x)$.

2. Trigonometric Identity: By the angle-addition identities and the induction hypothesis:

$$S_x = \sin(\alpha_u) \cos \Big( \sum_{v=1}^{p} y_v \alpha_v \Big) + \cos(\alpha_u) \sin \Big( \sum_{v=1}^{p} y_v \alpha_v \Big)$$

$$= \sin \Big( \alpha_u + \sum_{v=1}^{p} y_v \alpha_v \Big) = \sin \Big( \sum_{v=1}^{p} x_v \alpha_v \Big)$$

Similarly,

$$C_x = \cos(\alpha_u) \cos \Big( \sum_{v=1}^{p} y_v \alpha_v \Big) - \sin(\alpha_u) \sin \Big( \sum_{v=1}^{p} y_v \alpha_v \Big)$$

$$= \cos \Big( \alpha_u + \sum_{v=1}^{p} y_v \alpha_v \Big) = \cos \Big( \sum_{v=1}^{p} x_v \alpha_v \Big)$$

This completes the induction. □

### G.1.1 PIECEWISE-POLYNOMIAL ACTIVATIONS

**Theorem G.15** (Two-layer piecewise-polynomial activations). *Let $\sigma$ be as in Def. 5.3. For the two-layer MLP defined in the model setup,*

$$\mathrm{Ndim}(\mathcal{H}_\Theta) \leq 2dp \left( 6 \log(6dp) + \log(2eL) + 2 \log(epr) \right) = \widetilde{\mathcal{O}}(dp)$$

*Proof of Thm. G.15.* Let $S = \{x^{(1)}, \ldots, x^{(n)}\} \subset \mathcal{X}_m$ be Natarajan-shattered. By Lem. G.6, this implies $2^n \leq \Pi_{\mathcal{G}_\Theta}(n\binom{p}{2})$. The parameter space $W \in \mathbb{R}^{dp}$ is partitioned into regions by the zero sets of $\{w_i x^{(j)} - b_\ell\}$ for $j \in [n], i \in [d], \ell \in [L-1]$. The number of regions, $R_S$, is the number of sign patterns on a sample of size $m = nd(L-1)$ by affine functions of $W$.

Notice that $\left\{ w \mapsto \mathrm{sgn}\,(wx - b_\ell) : w \in \mathbb{R}^{1 \times p}, x \in S, \ell \in [L-1] \right\} \subset \{-1, +1\}^{\mathcal{X}_m}$ is a 1-combination of $\mathrm{sgn}\left( \left\{ w \mapsto (wx - b_\ell) : w \in \mathbb{R}^{1 \times p}, x \in S, \ell \in [L-1] \right\} \subset \mathbb{R}^{\mathcal{X}_m} \right)$.

By Thm. G.11, $R_S \leq \max \left\{ 2^{dp}, 2 \left( \frac{2e \cdot nd(L-1)}{dp} \right)^{dp} \right\}$. Within each region, $s_i^\theta(x) - s_j^\theta(x)$ is a polynomial in $\theta \in \mathbb{R}^{2dp}$ of degree at most $r+1$. Let $N = n\binom{p}{2}$, $D = 2dp$. The growth function is bounded by the product of the number of regions and the maximum growth function within a region. Applying Thm. G.11 in each region:

$$\Pi_{\mathcal{G}_\Theta}(N) \leq R_S \cdot \max_R \Pi_R(N) \leq \max \left\{ 2^{dp}, 2 \left( \frac{2e \cdot nd(L-1)}{dp} \right)^{dp} \right\} \max \left\{ 2^{2dp}, 2 \left( \frac{eN(r+1)}{dp} \right)^{2dp} \right\}$$

Substituting $N = \frac{np(p-1)}{2}$ into the inequality $2^n \leq \Pi_{\mathcal{G}_\Theta}(N)$ gives $2^n \leq (2enL)^{dp} (enpr)^{2dp}$. Taking the logarithm of both sides yields

$$n \leq 3dp \log(n)/\log(2) + dp(\log(2eL) + 2 \log(epr))/\log(2).$$

Lem. G.13 implies that

$$n \leq 2dp \left( 6 \log(6dp) + \log(2eL) + 2 \log(epr) \right) / \log(2) = \widetilde{\mathcal{O}}(dp).$$

Take supreum over $S$ yields the result.

□

### G.1.2 TRIGONOMETRIC-POLYNOMIAL ACTIVATIONS

**Theorem G.16** (Two-layer trigonometric-polynomial activations). *Let $\sigma$ be as in Def. 5.4. For the two-layer MLP from the model setup,*

$$\mathrm{Ndim}(\mathcal{H}_\Theta) \leq 2dp \Big( 6 \log(6dp) + 2 \log\big(ep(Km+1)\big) \Big) = \widetilde{\mathcal{O}}(dp).$$

*Proof of Thm. G.16.* Let $S = \{x^{(1)}, \ldots, x^{(n)}\} \subset \mathcal{X}_m$ be Natarajan-shattered. By Lem. G.6, this implies $2^n \leq \Pi_{\mathcal{G}_\Theta}(n\binom{p}{2})$.

For $j \in [d], v \in [p]$ set $c_{j,v} := \cos(w_{j,v})$ and $s_{j,v} := \sin(w_{j,v})$, and regard

$$a := \big(V, (c_{j,v})_{j,v}, (s_{j,v})_{j,v}\big) \in \mathbb{R}^{3dp}$$

as the (relaxed) parameter vector; ignoring the constraints $c_{j,v}^2 + s_{j,v}^2 = 1$ can only increase the growth function. For any $(i, y \neq y')$,

$$s_y^\theta(x^{(i)}) - s_{y'}^\theta(x^{(i)}) = \sum_{j=1}^d (V_{yj} - V_{y'j})\sigma(\langle w_j, x^{(i)}\rangle).$$

Writing $\sigma$ as in Def. 5.4 and applying Lem. G.14 to $kx^{(i)}$ shows that each term $\cos(k\langle w_j, x^{(i)}\rangle)$ and $\sin(k\langle w_j, x^{(i)}\rangle)$ is a polynomial in $\big((c_{j,v})_v, (s_{j,v})_v\big)$ of degree at most $km \leq Km$. Hence every pairwise margin is a polynomial in $a$ of degree at most $Km + 1$.

The reduction class $\mathcal{G}_\Theta$ is a 1-combination of $\mathrm{sgn}(\mathcal{F})$ with $\mathcal{F}$ being a family of polynomials of degree at most $Km + 1$ in $D = 3dp$ parameters. Applying Thm. G.11 with $N = n\binom{p}{2}$, $k = 1$, $r = Km + 1$,

$$\Pi_{\mathcal{G}_\Theta}(N) \leq \max\Big\{2^D, 2\Big(\frac{2eNr}{D}\Big)^D\Big\} \leq 2\,(pKnm)^{3dp}.$$

Combine with $2^n \leq \Pi_{\mathcal{G}_\Theta}(N)$, take logs: $n\log(2) \leq \log(2) + 3dp\log(pKm) + 3dp\log(n)$. Use Lem. G.13 to absorb the $\log n$ term, yielding

$$n \leq 12dp\log(6dp)/\log(2) + 2\,(\log(2) + 3dp\log(pKm))\,/\log(2) = \widetilde{\mathcal{O}}\,(dp)$$

Taking the supremum over shattered $S$ gives the claim. $\qquad\square$

### G.1.3 RATIONAL-EXPONENTIAL ACTIVATIONS

**Theorem G.17** (Two-layer polynomial–rational–exponential activations)**.** *Let $\sigma$ be as in Def. 5.5. For the two-layer MLP from the model setup,*

$$\mathrm{Ndim}(\mathcal{H}_\Theta) \leq 2dp\Big(6\log(6dp) + 2\log\big(ep(dm + r + 1)\big)\Big) = \widetilde{\mathcal{O}}(dp).$$

*Proof of Thm. G.17.* Let $S = \{x^{(1)}, \ldots, x^{(n)}\} \subset \mathcal{X}_m$ be Natarajan-shattered and set $N := n\binom{p}{2}$. For each example $i$ and hidden unit $j$, put $z_{j,i} := e^{k\langle w_j, x^{(i)}\rangle} > 0$. Since $c \geq 0$ and $\tau > 0$, the product

$$D_i(W) := \prod_{j=1}^d \big(cz_{j,i} + \tau\big) > 0.$$

Multiplying any pairwise margin $G_{i,y,y'} := s_y^\theta(x^{(i)}) - s_{y'}^\theta(x^{(i)})$ by $D_i(W)$ preserves its sign and yields

$$\widehat{G}_{i,y,y'}(W, V) = D_i(W)G_{i,y,y'}(W, V) = \sum_{j=1}^d (V_{yj} - V_{y'j})P(\langle w_j, x^{(i)}\rangle)(az_{j,i} + b)\prod_{\ell \neq j}(cz_{\ell,i} + \tau).$$

Introduce relaxed variables $u_{j,v} := e^{kw_{j,v}} \in (0, \infty)$. Then

$$z_{j,i} = e^{k\langle w_j, x^{(i)}\rangle} = \prod_{v=1}^p u_{j,v}^{x_v^{(i)}},$$

a monomial of total degree $m$ in $U_j := (u_{j,1}, \ldots, u_{j,p})$. Consequently, each summand in $\widehat{G}_{i,y,y'}$ is a product of: (i) a linear term in $V$; (ii) the degree-$r$ polynomial $P(\langle w_j, x^{(i)}\rangle)$ in $W_j$; (iii) a factor $(az_{j,i} + b)\prod_{\ell \neq j}(cz_{\ell,i} + \tau)$ of total degree $dm$ in the $U$-variables. Thus every $\widehat{G}_{i,y,y'}$ is a polynomial in

$$a := \big(V, (u_{j,v})_{j,v}, (w_{j,v})_{j,v}\big) \in \mathbb{R}^{3dp}$$

of degree at most $\rho := dm + r + 1$. Treating $a$ as the parameter vector, the reduction class $\mathcal{G}_\Theta$ is a 1-combination of $\mathrm{sgn}(\mathcal{F})$ with $\mathcal{F}$ being a family of polynomials of degree at most $\rho$ in $D = 3dp$ parameters. Applying Thm. G.11 with $k = 1$, $D = 3dp$, $N = n\binom{p}{2}$,

$$\Pi_{\mathcal{G}_\Theta}(N) \leq \max\Big\{ 2^D, 2\Big(\frac{2eN\rho}{D}\Big)^D \Big\} \leq (np(dm + r + 1))^{3dp} .$$

Combine with Lem. G.6 and absorb the $\log n$ term via Lem. G.13 to obtain

$$n \leq 12dp \log(6dp/\log(2))/\log(2) + 6dp \log(p(dm + r + 1))/\log(2) = \widetilde{\mathcal{O}}(dp)$$

Taking the supremum over shattered $S$ gives the claim. $\qquad\square$

### G.1.4 UNIFORM CONVERGENCE GUARANTEES

Let $\mathcal{H}_\sigma \subseteq [p]^{\mathcal{X}}$ be a multiclass hypothesis class with Natarajan-dimension $\mathrm{Ndim}(\mathcal{H}_\sigma) < \infty$. Let $h \in \mathcal{H}$, denote by $\mathbb{P}_{(x,y)\in\mathcal{D}}[h(x) \neq y]$ the population 0–1 risk and by $\mathbb{P}_{(x,y)\in\mathcal{D}_{\mathrm{train}}}[h(x) \neq y]$ the empirical 0–1 risk computed from an i.i.d. sample of size $n$.

**Theorem G.18** (The Multiclass Fundamental Theorem, Thm. 29.3 of (Shalev-Shwartz & Ben–David, 2014), Uniform convergence). *There exists a universal constant $C > 0$ such that, for every $\delta \in (0, 1)$, with probability at least $1 - \delta$,*

$$\sup_{h\in\mathcal{H}_\sigma} \big| \mathbb{P}_{(x,y)\in\mathcal{D}}[h(x) \neq y] - \mathbb{P}_{(x,y)\in\mathcal{D}_{train}}[h(x) \neq y] \big| \;\leq\; C\sqrt{\frac{\mathrm{Ndim}(\mathcal{H}_\sigma)\log p + \log(1/\delta)}{n}} .$$

*Proof of Thm. 5.6.* By Thm. G.15, G.16, G.17, $\mathrm{Ndim}(\mathcal{H}_\sigma) = \widetilde{\mathcal{O}}(dp)$. Substituting this into the multiclass uniform convergence bound (Thm. G.18) yields

$$\sup_{h\in\mathcal{H}_\sigma} \Big| \mathbb{P}_{(x,y)\sim\mathcal{D}}\big[h(x) \neq y\big] - \mathbb{P}_{(x,y)\sim\mathcal{D}_{train}}\big[h(x) \neq y\big] \Big| \;\leq\; C\sqrt{\frac{\mathrm{Ndim}(\mathcal{H}_\sigma)\log p + \log(1/\delta)}{n}} = \widetilde{\mathcal{O}}\Bigg( \sqrt{\frac{dp + \log(1/\delta)}{n}} \Bigg),$$

where the $\log p$ factor is absorbed into $\widetilde{\mathcal{O}}(\cdot)$. $\qquad\square$

### G.2 LOWER BOUND OF NATARAJAN-DIMENSION

Let $\Theta$ denote the backbone parameter space determined by the architecture. The multiclass hypothesis class is

$$\mathcal{H} = \big\{ h_{\theta,V} \;:\; \theta \in \Theta, \; V \in \mathbb{R}^{p\times d} \big\}.$$

**Definition G.19** (Associated binary class). We consider the binary (realizable) subclass of halfspaces in the representation:

$$\mathcal{M} \;=\; \Big\{ x \mapsto \mathbf{1}\big\{\langle v, \; f_\theta(x)\rangle \geq 0\big\} \;:\; (\theta, v) \in \Theta \times \mathbb{R}^d \Big\} \;\subseteq\; \{0,1\}^{\mathcal{X}}.$$

**Lemma G.20** (VC-dimension is bounded by the Natarajan-dimension). *For the multiclass hypothesis class $\mathcal{H}$ with $p \geq 2$,*

$$\mathrm{VCdim}(\mathcal{M}) \;\leq\; \mathrm{Ndim}(\mathcal{H}).$$

*Proof.* Let $S \subseteq \mathcal{X}$ be a finite set that is VC-shattered by $\mathcal{M}$.

Fix $f_1, f_2 \in [p]^S$ by $f_1(x) \equiv 1$ and $f_2(x) \equiv 2$ for all $x \in S$ (possible since $p \geq 2$). Let $b : S \to \{1, 2\}$ be an arbitrary selector. Define the induced binary labeling

$$y_b(x) \;=\; \mathbf{1}\{b(x) = 1\} \;\in\; \{0, 1\} \qquad (x \in S).$$

Since $S$ is VC-shattered by $\mathcal{M}$, there exist $\theta_b \in \Theta$ and $v_b \in \mathbb{R}^d$ such that

$$y_b(x) \;=\; \mathbf{1}\big\{\langle v_b, f_{\theta_b}(x)\rangle \geq 0\big\} \quad \text{for all } x \in S.$$

Construct $V_b \in \mathbb{R}^{p\times d}$ such that the first row is $v_b$, and all remaining rows are 0. Then, for each $x \in S$,

$$h_{\theta_b, V_b}(x) \;=\; \begin{cases} 1, & \text{if } y_b(x) = 1 \; (b(x) = 1), \\ 2, & \text{if } y_b(x) = 0 \; (b(x) = 2), \end{cases}$$

i.e., $h_{\theta_b, V_b}(x) = f_{b(x)}(x)$ for all $x \in S$. Since $b$ was arbitrary, $S$ is Natarajan-shattered by $\mathcal{H}$ with witnesses $(f_1, f_2)$. Therefore $|S| \leq \mathrm{Ndim}(\mathcal{H})$. Taking the supremum over all such $S$ gives $\mathrm{VCdim}(\mathcal{M}) \leq \mathrm{Ndim}(\mathcal{H})$.

$\square$

## H  CAPACITY BOUNDS IN TAB. 1

**Notation and scope.** All entries are for two-layer MLPs (one hidden layer) with $W$ trainable parameters and width $d$. Throughout, $\widetilde{\Theta}(\cdot)$ hides polylogarithmic factors in $W$, $d$, and the bound $M$ on the input.

### H.1  TWO-LAYER MLP SETUP IN TAB. 1

**Model.** For width $d$, input dimension $n$, output dimension $K \geq 1$, and elementwise activation $\sigma$. The score map is

$$s_\theta(x) = V_2\, \sigma(V_1 x) \in \mathbb{R}^K, \qquad V_1 \in \mathbb{R}^{d \times n},\ V_2 \in \mathbb{R}^{K \times d}$$

Total parameters

$$W\ =\ Kd + dn$$

and the table reports bounds as functions of $W$.

**Input regimes.** All constraints apply to the vector *entering the first linear layer*.

1. Real-valued: $\mathcal{X} = \mathbb{R}^n$.
2. Integer-valued, bounded by $M$: $\mathcal{X} = \{x \in \mathbb{Z}^n : \|x\|_\infty \leq M\}$.
3. Integer-valued, unbounded: $\mathcal{X} = \mathbb{Z}^n$.

**Representation, scores, and hypothesis classes.** Let $\Theta = \{(V_1, V_2) : V_1 \in \mathbb{R}^{d \times n},\ V_2 \in \mathbb{R}^{K \times d}\}$. We write the learned representation and scores as

$$f_\theta(x) = \sigma(V_1 x) \in \mathbb{R}^d, \qquad s_\theta(x) = V_2\, f_\theta(x) \in \mathbb{R}^K.$$

For $K \geq 2$ (multiclass), the hypothesis class is

$$\mathcal{H} = \Big\{ h_\theta : \mathcal{X} \to [K] \cup \{\bot\}, \quad h_\theta(x) = \psi_{\mathrm{uargmax}}\big(s_\theta(x)\big) \ \Big|\ \theta \in \Theta \Big\},$$

where $\psi_{\mathrm{uargmax}}(u)$ returns the index of the unique maximum of the vector $u$, or $\bot$ (invalid) if the maximum is not unique.

**Definition H.1** (Associated binary class restated; Def. G.19)**.** We consider the binary (realizable) subclass of halfspaces in the representation:

$$\mathcal{M}\ =\ \Big\{ x \mapsto \mathbf{1}\big\{\langle v,\ f_\theta(x)\rangle \geq 0\big\}\ :\ (\theta, v) \in \Theta \times \mathbb{R}^d \Big\}\ \subseteq\ \{0,1\}^{\mathcal{X}}.$$

**Bound on integer inputs.** In the bounded–integer regime we fix $M \in \mathbb{N}$ and take

$$\mathcal{X} = \big\{x \in \mathbb{Z}^n :\ \|x\|_\infty \leq M\big\}.$$

Throughout, $\widetilde{\Theta}(\cdot)$ and $\widetilde{\mathcal{O}}(\cdot)$ hide polylogarithmic factors in $W$, $d$, and (when applicable) $M$.

**Output Type and Complexity Measure.** Predictions are obtained from scores via a fixed decoder:

1. **VC-dimension (binary subclass):** take $K = 1$ and $\psi_{\mathrm{sign}}(z) = \mathrm{sgn}(z)$ (assigning $\mathrm{sgn}(0) = 1$); VCdim is measured on $\{\psi_{\mathrm{sign}} \circ s_\theta\}$.
2. **Natarajan dimension (multiclass):** take any $K \geq 2$ and $\psi_{\mathrm{uargmax}}$ (strict uniqueness) as defined above; Ndim is measured on $\{\psi_{\mathrm{uargmax}} \circ s_\theta\}$.

**Scope of the table.** Tab. 1 concerns the general class above, only the total parameter count $W$ and the width $d$ matter for the rates.

## H.2 SOURCES FOR THE VC-DIMENSION BOUNDS

**Piecewise linear, real-valued** (VCdim $= \Theta(W \log W)$). The nearly-tight bounds for piecewise-linear networks are summarized by (Bartlett et al., 2017b, Eq. (2)); for fixed depth $L = 2$ this specializes to $\Theta(W \log W)$.

**Piecewise-polynomial, real-valued** (VCdim $= \Theta(W \log W)$). (Anthony & Bartlett, 2009, Thm. 8.8) prove an upper bound of $\mathcal{O}(WL \log W + WL^2)$ for networks with piecewise-polynomial activations of bounded degree and a bounded number of pieces; in the depth-2 case this simplifies to $\mathcal{O}(W \log W)$. A matching lower bound of $\Omega(W \log W)$ for two-layer linear-threshold networks (a special case with degree 0) appears in (Anthony & Bartlett, 2009, Thm. 6.4). Using a refined bit-extraction technique, (Bartlett et al., 2017b, Thm. 3) further gives an explicit construction achieving $\Omega(WL \log(W/L))$ for ReLU networks, which in particular gives $\Omega(W \log W)$ for depth-2 networks.

**Pfaffian activations (incl. standard sigmoid), real-valued** (VCdim $= \mathcal{O}(d^2 W^2)$). A general upper bound $\mathcal{O}(W^2 k^2)$ for standard sigmoid networks is given in (Anthony & Bartlett, 2009, Thm. 8.13), where $k$ is the number of computation units; in a two-layer networks, $k = d$. The Pfaffian extension follows from Khovanskii's *Fewnomials*: the theorem underlying Lemma 8.15 (see also (Anthony & Bartlett, 2009, §8.6)) bounds the number of connected components (Betti numbers) of semi-Pfaffian sets defined by functions from a fixed Pfaffian chain. Plugging this component bound into the standard growth function argument used for the exponential case yields the same $\mathcal{O}(d^2 W^2)$ VC-dimension bound for networks whose activations lie in a fixed Pfaffian chain (with order/degree independent of the data).

**Standard sigmoid, real-valued** (VCdim $= \Omega(W \log W)$). The reduction from linear-threshold to smooth sigmoids (Anthony & Bartlett, 2009, Thm. 6.5) implies that the two-layer linear-threshold lower bound (Anthony & Bartlett, 2009, Thm. 6.4) carries over to standard sigmoid networks on the same finite set of inputs. This yields $\Omega(W)$lower bound, and $\Omega(W \log W)$ under the construction of "bit extraction"(Bartlett et al., 2017b, Rmk. 4).

**Standard sigmoid, integer-valued bounded inputs** (VCdim $= \widetilde{\Theta}(W)$). For two-layer standard sigmoid networks with integer-valued inputs and first-layer fan-in $\leq N$, (Anthony & Bartlett, 2009, Thm. 8.11) gives VCdim $\leq 2W \log_2(60ND) = \widetilde{\mathcal{O}}(W)$. The paragraph following the theorem constructs a two-layer linear-threshold network with VCdim $= \Omega(W)$; (Anthony & Bartlett, 2009, Thm. 6.5) transfers this lower bound to sigmoids. Hence the bound is $\widetilde{\Theta}(W)$.

**Sine, integer-valued unbounded inputs** (VCdim $= \infty$). (Anthony & Bartlett, 2009, Lemma 7.2) shows that $\{x \mapsto \mathrm{sgn}(\sin(ax))\}$ has infinite VC-dimension. Thus, restricting to two labels, the corresponding multiclass Natarajan-dimension is also infinite.

## H.3 NATARAJAN-DIMENSION LOWER BOUNDS

We now transfer known VC-dimension lower bounds for the binary subclass to the multiclass setting via Lem. G.20.

**Theorem H.2** (Natarajan-dimension lower bounds for two-layer MLPs). *Consider the two-layer MLP family in App. H.1, with width $d$, total parameter count $W = Kd + dn$, and $K \geq 2$ classes decoded by $\psi_{\mathrm{argmax}}$. Let $\mathcal{H}$ denote the resulting multiclass hypothesis class and let $\mathcal{M}$ be the associated binary subclass of halfspaces in the learned representation (Def. G.19). Then*

$$\mathrm{Ndim}(\mathcal{H}) \geq \mathrm{VCdim}(\mathcal{M}),$$

*and, in particular, for the activation/input regimes appearing in Tab. 1 for which VC-dimension lower bounds are known, the following existential lower bounds hold:*

1. ***Piecewise linear, real-valued inputs:*** $\mathrm{Ndim}(\mathcal{H}) \geq \Omega(W \log W)$.

2. ***Piecewise polynomial, real-valued inputs:*** $\mathrm{Ndim}(\mathcal{H}) \geq \Omega(W \log W)$.

3. ***Standard sigmoid, real-valued inputs:*** $\mathrm{Ndim}(\mathcal{H}) \geq \Omega(W \log W)$.

4. ***Standard sigmoid, integer-valued bounded inputs:*** $\mathrm{Ndim}(\mathcal{H}) \geq \Omega(W)$.

5. ***Sine, integer-valued unbounded inputs:*** $\mathrm{Ndim}(\mathcal{H}) = \infty$.

*All bounds are stated for the two-layer architecture in App. H.1.*

*Proof.* By Lem. G.20, $\mathrm{VCdim}(\mathcal{M}) \leq \mathrm{Ndim}(\mathcal{H})$, where $\mathcal{M}$ is the binary subclass.

It remains to instantiate $\mathrm{VCdim}(\mathcal{M})$ in each regime. By construction (Def. G.20 and App. H.1), $\mathcal{M}$ is exactly the binary two-layer network class used in the VC-dimension entries of Tab. 1. Therefore, from Sec. H.2:

1. For piecewise-linear and piecewise-polynomial activations with real-valued inputs, $\mathrm{VCdim}(\mathcal{M}) \geq \Omega(W \log W)$, yielding the first two claims.

2. For standard sigmoids with real-valued inputs, $\mathrm{VCdim}(\mathcal{M}) \geq \Omega(W \log W)$, yielding the third claim.

3. For standard sigmoids with bounded integer inputs, $\mathrm{VCdim}(\mathcal{M}) \geq \Omega(W)$, yielding the fourth claim.

4. For sine activations with unbounded integer inputs, the binary subclass has infinite VC-dimension, hence $\mathrm{Ndim}(\mathcal{H}) = \infty$.

These bounds are existential: for each $(W, d)$ there exist network parameters achieving the stated shattering. $\qquad\square$

## H.4    NATARAJAN-DIMENSION UPPER BOUNDS

We restate the Natarajan-dimension upper bounds for the two-layer MLP in App. H.1. Throughout, the score map is $s_\theta(x) = V_2\,\sigma(V_1 x) \in \mathbb{R}^K$ with width $d$, output size $K \geq 2$. The proofs are essentially the same as in Thms. G.15, G.17, and G.16, with only the notational substitution of the updated score map and parameter count.

**Theorem H.3** (Piecewise-polynomial activations, real-valued inputs)**.** *Assume the model of App. H.1 with scores $s_\theta(x) = V_2\,\sigma(V_1 x) \in \mathbb{R}^K$, prediction by $\psi_{\mathrm{uargmax}}$, and total parameters $W = dn + Kd$. Let $\sigma$ be piecewise-polynomial on $\mathbb{R}$ with at most $L$ pieces and maximal piece degree $r$, where $L, r$ are absolute constants (do not grow with $n, d, K$). For the input domain $\mathcal{X} = \mathbb{R}^n$, the multiclass hypothesis class $\mathcal{H} = \{\psi_{\mathrm{uargmax}} \circ s_\theta\}$ satisfies*

$$\mathrm{Ndim}(\mathcal{H}) \leq \widetilde{\mathcal{O}}(W),$$

*where $\widetilde{\mathcal{O}}(\cdot)$ hides polylogarithmic factors in $W$, $d$ (and in the structural constants $L, r$).*

**Theorem H.4** (Polynomial–rational–exponential activations (incl. logistic sigmoid), bounded integer-valued inputs)**.** *Assume the model of App. H.1 with scores $s_\theta(x) = V_2\,\sigma(V_1 x) \in \mathbb{R}^K$, prediction by $\psi_{\mathrm{uargmax}}$, and total parameters $W = dn + Kd$. Let $\sigma$ be of the form in Def. 5.5, i.e.,*

$$\sigma(t) = P(t)\,\frac{ae^{kt} + b}{ce^{kt} + \tau},$$

*with fixed scalars $k \in \mathbb{R} \setminus \{0\}$, $c \geq 0$, $\tau > 0$, $a, b \in \mathbb{R}$, and $\deg P \leq r$, where $r$ is an absolute constant (do not grow with $n, d, K, M$). For the input domain $\mathcal{X} = \{x \in \mathbb{Z}^n : \|x\|_\infty \leq M\}$ with $M \in \mathbb{N}$, the multiclass hypothesis class $\mathcal{H} = \{\psi_{\mathrm{uargmax}} \circ s_\theta\}$ satisfies*

$$\mathrm{Ndim}(\mathcal{H}) \leq \widetilde{\mathcal{O}}(W),$$

*where $\widetilde{\mathcal{O}}(\cdot)$ hides polylogarithmic factors in $W$, $d$, $M$ (and in the structural constant $r$).*

**Theorem H.5** (Trigonometric-polynomial activations, bounded integer-valued inputs). *Assume the model of App. H.1 with scores $s_\theta(x) = V_2\,\sigma(V_1 x) \in \mathbb{R}^K$, prediction by $\psi_{\mathrm{uargmax}}$, and total parameters $W = dn + Kd$. Let $\sigma$ be a trigonometric polynomial of degree at most $T$, where $T$ is an absolute constant. For the input domain $\mathcal{X} = \{x \in \mathbb{Z}^n : \|x\|_\infty \leq M\}$ with $M \in \mathbb{N}$, the multiclass hypothesis class $\mathcal{H} = \{\psi_{\mathrm{uargmax}} \circ s_\theta\}$ satisfies*

$$\mathrm{Ndim}(\mathcal{H}) \leq \widetilde{\mathcal{O}}(W),$$

*where $\widetilde{\mathcal{O}}(\cdot)$ hides polylogarithmic factors in $W$, $d$, $M$ (and in the structural constant $T$).*

Immediate by combining Thms. H.3, H.4, and H.5, we have

**Theorem H.6** ($\widetilde{\mathcal{O}}(W)$ Natarajan dimension upper bound for two-layer MLPs). *Consider the model of App. H.1 with scores $s_\theta(x) = V_2\,\sigma(V_1 x) \in \mathbb{R}^K$, where $V_1 \in \mathbb{R}^{d \times n}$, $V_2 \in \mathbb{R}^{K \times d}$, and total parameters $W = dn + Kd$. Prediction is by $\psi_{\mathrm{uargmax}}$. Assume throughout that all structural constants below are absolute (do not grow with $n, d, K, M$):*

1. *$\sigma$ is piecewise-polynomial with at most $L$ pieces and maximal piece degree $r$ on $\mathbb{R}$, and $\mathcal{X} = \mathbb{R}^n$;*

2. *$\sigma$ is polynomial–rational–exponential whose polynomial factor $P(t)$ has degree at most $s$, and $\mathcal{X} = \{x \in \mathbb{Z}^n : \|x\|_\infty \leq M\}$;*

3. *$\sigma$ is a trigonometric-polynomial of degree at most $T$, and $\mathcal{X} = \{x \in \mathbb{Z}^n : \|x\|_\infty \leq M\}$.*

*Then the multiclass hypothesis class $\mathcal{H} = \{\psi_{\mathrm{uargmax}} \circ s_\theta\}$ satisfies*

$$\mathrm{Ndim}(\mathcal{H}) \leq \widetilde{\mathcal{O}}(W),$$

*where $\widetilde{\mathcal{O}}(\cdot)$ hides polylogarithmic factors in $W$, $d$, and, when applicable, $M$ (as well as in the structural constants $L, r, T, s$).*

# I  HIGH-MARGIN INTERPOLATING SOLUTIONS

In this section, we present high-margin interpolating solutions for two-layer MLPs with sine and ReLU activations for a fixed length.

## I.1  SINE ACTIVATION $\sigma(z) = \sin z$

**Theorem I.1** (High margin, $d = 2p$). *There exists a construction with hidden dimension $d = 2p$ and sine activation computes $\sum_{i=1}^m s_i \bmod p$ for all $x = (s_1, \cdots, s_m) \in \mathcal{X}_m$, achieving margin $\gamma = p$ and $\|V\|_2 = \sqrt{p}$, $\|W\|_F \leq \pi\sqrt{2}p$.*

*Proof.* Index hidden units by $h \in \{1, \ldots, 2p\}$ and group them as $(2k-1, 2k)$ for frequencies $k \in \{1, \ldots, p\}$. Let $\phi_k = \frac{2\pi k}{p}$.

**First-Layer Weights** $W \in \mathbb{R}^{2p \times p}$. For each $k \in \{1, \ldots, p\}$ and $r \in [p]$,

$$W_{2k-1, r} = (\phi_k r) \bmod 2\pi \in [-\pi, \pi), \qquad W_{2k, r} = \left(\phi_k r + \frac{\pi}{2m}\right) \bmod 2\pi \in [-\pi, \pi).$$

Then

$$(Wx)_{2k-1} = \phi_k S, \qquad (Wx)_{2k} = \phi_k S + \frac{\pi}{2},$$

and

$$\sigma\big((Wx)_{2k-1}\big) = \sin(\phi_k S), \qquad \sigma\big((Wx)_{2k}\big) = \cos(\phi_k S).$$

The first layer satisfies $\|W\|_\infty \leq \pi$, hence $\|W\|_F \leq \pi\sqrt{(2p)p} = \pi\sqrt{2}\,p$.

**Second-Layer Weights** $V \in \mathbb{R}^{p \times 2p}$. For each $q \in [p]$ and $k \in \{1, \ldots, p\}$,

$$V_{q, 2k-1} = \sin(\phi_k q), \qquad V_{q, 2k} = \cos(\phi_k q).$$

**Verification** For the $q$-th output,

$$
\begin{aligned}
s_q^\theta(x) &= \sum_{k=1}^p \Big[ \sin(\phi_k q)\sin(\phi_k S) + \cos(\phi_k q)\cos(\phi_k S) \Big] \\
&= \sum_{k=1}^p \cos\big(\phi_k(S-q)\big) = \Re\left( \sum_{k=1}^p e^{\,i\frac{2\pi k}{p}(S-q)} \right) \\
&= \begin{cases} p, & S \equiv q \pmod{p}, \\ 0, & \text{otherwise.} \end{cases}
\end{aligned}
$$

Hence $h_\theta(x) = S \bmod p$ with margin $\gamma = p$. The construction achieves $100\%$ accuracy with width $d = 2p$ and satisfies $\|W\|_\infty \le \pi$, $\|V\|_\infty \le 1$. $\qquad\square$

**Lemma I.2** (Singular values of $V$). *In the high-margin construction, all singular values of $V$ are exactly $\sqrt{p}$, so $\|V\|_2 = \sqrt{p}$.*

*Proof.* Compute $VV^\top$ entrywise. For $q, r \in [p]$,

$$
\begin{aligned}
(VV^\top)_{qr} &= \sum_{k=1}^p \Big( \sin(\phi_k q)\sin(\phi_k r) + \cos(\phi_k q)\cos(\phi_k r) \Big) \\
&= \sum_{k=1}^p \cos\big(\phi_k(q-r)\big) \qquad (\text{using } \cos(a-b) = \cos a \cos b + \sin a \sin b) \\
&= \Re\left( \sum_{k=1}^p e^{\,i\frac{2\pi k}{p}(q-r)} \right) \\
&= \begin{cases} 0, & \text{if } q - r \not\equiv 0 \mod p \\ p, & \text{if } q - r \equiv 0 \mod p \end{cases}
\end{aligned}
$$

Therefore all eigenvalues of $VV^\top$ are exactly $p$, so all singular values of $V$ are exactly $\sqrt{p}$. In particular, the spectral norm is $\|V\|_2 = \sqrt{p}$. $\qquad\square$

## I.2 ReLU activation $\sigma(z) = \mathrm{ReLU}(z)$

For a multi-index $a = (a_1, \ldots, a_s) \in \mathbb{Z}_{\ge 0}^s$, denote $|a| := \sum_{i=1}^s a_i$.

**Lemma I.3** (Polynomial sign polarization). *For $s \ge 1$,*

$$
x_1 x_2 \cdots x_s = \frac{1}{s!\,2^s} \sum_{\varepsilon \in \{\pm 1\}^s} \left( \prod_{i=1}^s \varepsilon_i \right) \left( \sum_{i=1}^s \varepsilon_i x_i \right)^s.
$$

*Proof.* Multinomial expansion gives

$$
\left( \sum_{i=1}^s \varepsilon_i x_i \right)^s = \sum_{|k|=s} \frac{s!}{k_1! \cdots k_s!} \prod_{i=1}^s (\varepsilon_i x_i)^{k_i}.
$$

Multiplying by $\prod_{j=1}^s \varepsilon_j$ and summing over $\varepsilon$ gives

$$
\sum_{\varepsilon \in \{\pm 1\}^s} \left( \prod_{j=1}^s \varepsilon_j \right) \left( \sum_{i=1}^s \varepsilon_i x_i \right)^s = \sum_{|k|=s} \frac{s!}{k_1! \cdots k_s!} x_1^{k_1} \cdots x_s^{k_s} \sum_{\varepsilon \in \{\pm 1\}^s} \prod_{i=1}^s \varepsilon_i^{k_i+1}.
$$

Observe that

$$
\sum_{\varepsilon \in \{\pm 1\}^s} \prod_{i=1}^s \varepsilon_i^{k_i+1} = \prod_{i=1}^s \big( (+1)^{k_i+1} + (-1)^{k_i+1} \big) = \begin{cases} 2^s, & \text{if each } k_i + 1 \text{ is even,} \\ 0, & \text{otherwise.} \end{cases}
$$

Because $|k| = s$ and each $k_i \geq 1$ must be odd so that $\sum_{\varepsilon \in \{\pm 1\}^s} \prod_{i=1}^{s} \varepsilon_i^{k_i+1} \neq 0$, the only possibility is $k = (1, \ldots, 1)$.

Therefore,

$$\sum_{\varepsilon \in \{\pm 1\}^s} \left( \prod_{i=1}^{s} \varepsilon_i \right) \left( \sum_{i=1}^{s} \varepsilon_i x_i \right)^s = s! \, 2^s x_1 x_2 \cdots x_s.$$

Dividing by $s! \, 2^s$ yields the stated identity.

□

**Lemma I.4** (Uniform ReLU–spline approximation of power functions). *Let $s \geq 1$, and $\varepsilon > 0$. Partition $[-1, 1]$ uniformly with knots $z_k = -1 + \frac{2k}{N}$, $k = 0, 1, \ldots, N$. Let $g$ be the linear spline that interpolates $f_s(z) = z^s$ at these knots. Then*

$$\|f_s - g\|_{L_\infty([-1,1])} \leq \frac{s(s-1)}{2N^2}.$$

*Moreover, $g$ admits an exact one-hidden-layer ReLU representation on $[-1, 1]$ of the form*

$$\Phi_s(z) = \sum_{i=1}^{M} c_i \, \mathrm{ReLU}(a_i z - b_i),$$

*with at most $M \leq N + 1$ units and*

$$|a_i| \leq 1, \qquad |b_i| \leq 1, \qquad |c_i| \leq \max\left\{ s + \tfrac{1}{2}, \, \frac{2 \, s(s-1)}{N} \right\}.$$

*Choosing*

$$N \geq \max\left\{ 1, \, \left\lceil \sqrt{\frac{s(s-1)}{2\varepsilon}} \right\rceil \right\}$$

*ensures $\|f_s - g\|_{L_\infty([-1,1])} \leq \varepsilon$. Thus, the number of required ReLU units to achieve accuracy $\varepsilon$ is $M = O\left( \frac{s}{\sqrt{\varepsilon}} \right)$.*

*Proof.* The case $s = 1$ is trivial since $f_1(z) = z$ is linear and equals its linear spline interpolant.

For $s \geq 2$, $f_s \in C^2([-1, 1])$ with $f_s''(z) = s(s-1)z^{s-2}$ and $\|f_s''\|_{L_\infty([-1,1])} = s(s-1)$. Fix $z \in [z_k, z_{k+1}]$ and define

$$\varphi(t) = f_s(t) - g(t) - \frac{f_s(z) - g(z)}{(z - z_k)(z - z_{k+1})}(t - z_k)(t - z_{k+1}).$$

Then $\varphi(z_k) = \varphi(z_{k+1}) = \varphi(z) = 0$ and, by Rolle's theorem, there exists $\xi_z \in (z_k, z_{k+1})$ such that

$$|f_s(z) - g(z)| = \left| \frac{f_s''(\xi_z)}{2} (z - z_k)(z_{k+1} - z) \right|.$$

Hence, with $h = \frac{2}{N}$,

$$\max_{z \in [z_k, z_{k+1}]} |f_s(z) - g(z)| \leq \tfrac{1}{2} \|f_s''\|_{L_\infty} \frac{h^2}{4} = \frac{s(s-1)}{2N^2}.$$

Taking the maximum over $k$ yields the stated uniform bound.

For the ReLU representation, write $h = \frac{2}{N}$ and set the interval slopes

$$m_k = \frac{z_{k+1}^s - z_k^s}{h}, \quad k = 0, \ldots, N-1, \qquad \gamma_j = m_j - m_{j-1} = \frac{z_{j+1}^s - 2z_j^s + z_{j-1}^s}{h}, \quad j = 1, \ldots, N-1.$$

Then $g$ admits the exact expansion on $[-1, 1]$:

$$g(z) = c_1 \, \mathrm{ReLU}(z + 1) + c_2 \, \mathrm{ReLU}(1 - z) + \sum_{j=1}^{N-1} \gamma_j \, \mathrm{ReLU}(z - z_j),$$

with

$$c_2 = \frac{f_s(-1)}{2} = \frac{(-1)^s}{2}, \qquad c_1 = m_0 + \frac{(-1)^s}{2},$$

and $(a_1, b_1) = (1, -1)$, $(a_2, b_2) = (-1, -1)$, $(a_{j+2}, b_{j+2}) = (1, z_j)$ for $j = 1, \ldots, N-1$.

Since $|z_j| \leq 1$, we have $|a_i| \leq 1$ and $|b_i| \leq 1$. By the mean value theorem, $|m_0| \leq \|f_s'\|_{L_\infty} = s$, hence $|c_1| \leq s + \frac{1}{2}$ and $|c_2| \leq \frac{1}{2} \leq s + \frac{1}{2}$.

Moreover, define $\Psi \in C^2[z_j - h, z_j + h]$ where

$$\Psi(t) = f_s(t) - \left( f_s(z_j) + \frac{f_s(z_j + h) - f_s(z_j - h)}{2h}(t - z_j) + \frac{f_s(z_j + h) - 2f_s(z_j) + f_s(z_j - h)}{2h^2}(t - z_j)^2 \right).$$

By Rolle's theorem,

$$\gamma_j = \frac{f_s(z_j + h) - 2f_s(z_j) + f_s(z_j - h)}{h} = h\, f_s''(\xi_j) \quad \text{for some } \xi_j \in (z_j - h, z_j + h),$$

so $|\gamma_j| \leq h\, \|f_s''\|_{L_\infty} = \frac{2}{N} s(s-1)$. Counting two boundary hinges and $N-1$ interior hinges gives $M \leq N + 1$ units.

$\square$

**Lemma I.5** (Polarized Newton expansion for $f_{\cos}$ and $f_{\sin}$). *Let $m \geq 1$. For angles $(\theta_1, \ldots, \theta_m)$, define*

$$C_k = \sum_{i=1}^m \cos(k\theta_i), \qquad S_k = \sum_{i=1}^m \sin(k\theta_i).$$

*Let*

$$\mathcal{K}_m := \left\{ k = (k_1, \ldots, k_m) \in \mathbb{Z}_{\geq 0}^m : \sum_{j=1}^m j\, k_j = m \right\}.$$

*We index*

$$\varepsilon = (\varepsilon_{1,1}, \ldots, \varepsilon_{1,k_1}, \varepsilon_{2,1}, \ldots, \varepsilon_{m,k_m}) \in \{\pm 1\}^{|k|}.$$

*For $p = (p_1, \ldots, p_m)$ we denote $p \leq k$ if $0 \leq p_j \leq k_j$. Then*

$$f_{\cos} = \cos\left( \sum_{i=1}^m \theta_i \right) = \sum_{k \in \mathcal{K}_m} \sum_{\substack{p \leq k \\ |p| - |k| \text{ even}}} \sum_{\varepsilon \in \{\pm 1\}^{|k|}} \alpha_{k,p,\varepsilon} G_{k,p,\varepsilon}^{|k|}$$

$$f_{\sin} = \sin\left( \sum_{i=1}^m \theta_i \right) = \sum_{k \in \mathcal{K}_m} \sum_{\substack{p \leq k \\ |p| - |k| \text{ odd}}} \sum_{\varepsilon \in \{\pm 1\}^{|k|}} \beta_{k,p,\varepsilon} G_{k,p,\varepsilon}^{|k|}$$

*Where*

$$G_{k,p,\varepsilon} = \sum_{j=1}^m \left( \sum_{\ell=1}^{p_j} \varepsilon_{j,\ell} \right) C_j + \sum_{j=1}^m \left( \sum_{\ell=p_j+1}^{k_j} \varepsilon_{j,\ell} \right) S_j$$

$$\alpha_{k,p,\varepsilon} = \frac{(-1)^{m - \sum k_j}}{\prod_{j=1}^m k_j! j^{k_j}} (-1)^{\frac{|k| - |p|}{2}} \left( \prod_{j=1}^m \binom{k_j}{p_j} \right) \frac{1}{|k|!\, 2^{|k|}} \left( \prod_{j=1}^m \prod_{\ell=1}^{k_j} \varepsilon_{j,\ell} \right)$$

$$\beta_{k,p,\varepsilon} = \frac{(-1)^{m - \sum k_j}}{\prod_{j=1}^m k_j! j^{k_j}} (-1)^{\frac{|k| - |p| - 1}{2}} \left( \prod_{j=1}^m \binom{k_j}{p_j} \right) \frac{1}{|k|!\, 2^{|k|}} \left( \prod_{j=1}^m \prod_{\ell=1}^{k_j} \varepsilon_{j,\ell} \right)$$

*and thus*

$$|\alpha_{k,p,\varepsilon}| = |\beta_{k,p,\varepsilon}| = \frac{1}{\left( \prod_{j=1}^m j^{k_j} \right) \left( \prod_{j=1}^m p_j!(k_j - p_j)! \right) |k|! 2^{|k|}} \leq \frac{1}{2}$$

*Furthermore, for $N_{tot}(m)$, the total amount of triples $(k, p, \varepsilon)$ (with $k \in \mathcal{K}_m$, $p \leq k$, $\varepsilon \in \{\pm 1\}^{|k|}$),*

$$N_{tot}(m) = \sum_{k \in \mathcal{K}_m} 2^{|k|} \prod_{j=1}^{m} (k_j + 1) \in [m2^m, 13m2^m].$$

*Proof.* Let $z_j = e^{i\theta_j}$ for $j = 1, \ldots, m$. The $k$-th power sum is $Z_k = \sum_{j=1}^{m} z_j^k = C_k + iS_k$. Let $e_m = \prod_{j=1}^{m} z_j = e^{i \sum_{j=1}^{m} \theta_j}$ be the $m$-th elementary symmetric polynomial in $z_1, \ldots, z_m$. The target functions are $f_{\cos} = \Re(e_m)$ and $f_{\sin} = \Im(e_m)$.

Newton's sum identities provide a formula expressing $e_m$ as a polynomial in the power sums $Z_1, \ldots, Z_m$:

$$e_m = P(Z_1, \ldots, Z_m) = \sum_{k \in \mathcal{K}_m} c_k \prod_{j=1}^{m} Z_j^{k_j}$$

where the coefficients $c_k$ are given by $c_k = \frac{(-1)^{m - \sum k_j}}{\prod_{j=1}^{m} k_j! j^{k_j}}$.

Binomial expansion yields

$$\prod_{j=1}^{m} Z_j^{k_j} = \prod_{j=1}^{m} (C_j + iS_j)^{k_j} = \sum_{p \leq k} \prod_{j=1}^{m} \left( \binom{k_j}{p_j} C_j^{p_j} (iS_j)^{k_j - p_j} \right) = \sum_{p \leq k} i^{|k| - |p|} \left( \prod_{j=1}^{m} \binom{k_j}{p_j} \right) \left( \prod_{j=1}^{m} C_j^{p_j} S_j^{k_j - p_j} \right)$$

For each pair of $(p, k)$, where $p \leq k$ and $k \in \mathcal{K}_m$, let

$$s := \sum_{j=1}^{m} k_j = |k|, \qquad x_{j,\ell} := \begin{cases} C_j, & 1 \leq \ell \leq p_j, \\ S_j, & p_j < \ell \leq k_j. \end{cases}$$

List the $s$ variables as $(x_{1,1}, \ldots, x_{1,k_1}, x_{2,1}, \ldots, x_{m,k_m})$. Applying Lem. I.3 to $x_1 \cdots x_s = \prod_{j=1}^{m} C_j^{p_j} S_j^{k_j - p_j}$ gives

$$\prod_{j=1}^{m} C_j^{p_j} S_j^{k_j - p_j} = \frac{1}{|k|! \, 2^{|k|}} \sum_{(\varepsilon_{j,\ell}) \in \{\pm 1\}^{|k|}} \left( \prod_{j=1}^{m} \prod_{\ell=1}^{k_j} \varepsilon_{j,\ell} \right) \left( \sum_{j=1}^{m} \left( \sum_{\ell=1}^{p_j} \varepsilon_{j,\ell} \right) C_j + \sum_{j=1}^{m} \left( \sum_{\ell=p_j+1}^{k_j} \varepsilon_{j,\ell} \right) S_j \right)^{|k|}$$

Therefore,

$$e_m = \sum_{k \in \mathcal{K}_m} c_k \prod_{j=1}^{m} Z_j^{k_j}$$

$$= \sum_{k \in \mathcal{K}_m} c_k \sum_{p \leq k} i^{|k| - |p|} \left( \prod_{j=1}^{m} \binom{k_j}{p_j} \right) \left( \prod_{j=1}^{m} C_j^{p_j} S_j^{k_j - p_j} \right)$$

$$= \sum_{k \in \mathcal{K}_m} c_k \sum_{p \leq k} i^{|k| - |p|} \left( \prod_{j=1}^{m} \binom{k_j}{p_j} \right) \frac{1}{|k|! \, 2^{|k|}} \sum_{(\varepsilon_{j,\ell}) \in \{\pm 1\}^{|k|}} \left( \prod_{j=1}^{m} \prod_{\ell=1}^{k_j} \varepsilon_{j,\ell} \right) \left( \sum_{j=1}^{m} \left( \sum_{\ell=1}^{p_j} \varepsilon_{j,\ell} \right) C_j + \sum_{j=1}^{m} \left( \sum_{\ell=p_j+1}^{k_j} \varepsilon_{j,\ell} \right) S_j \right)^{|k|}$$

Separate Real and Imaginary part yields the polarized Newton expansion for $f_{\cos}$ and $f_{\sin}$.

For $j \geq 1$,

$$\sum_{k_j \geq 0} (k_j + 1) \, 2^{k_j} \, t^{jk_j} = \sum_{r \geq 0} (r + 1)(2t^j)^r = \frac{1}{(1 - 2t^j)^2}.$$

Multiplying over $j$ gives the ordinary generating function

$$F(t) := \sum_{m \geq 0} N_{\text{tot}}(m) t^m = \prod_{j \geq 1} \frac{1}{(1 - 2t^j)^2} = \frac{1}{(1 - 2t)^2} \cdot H(t),$$

where

$$H(t) := \prod_{j \geq 2}(1 - 2t^j)^{-2} = \sum_{r \geq 0} h_r t^r, \qquad h_r \geq 0.$$

Since $(1 - 2t)^{-2} = \sum_{n \geq 0}(n + 1)2^n t^n$, the Cauchy product gives

$$N_{\text{tot}}(m) = \sum_{r=0}^{m} h_r \, (m - r + 1) \, 2^{\,m-r} \leq (m + 1)2^m \sum_{r=0}^{\infty} h_r \, 2^{-r} = (m + 1)2^m \, H\big(\tfrac{1}{2}\big).$$

Here $H(\tfrac{1}{2}) = \prod_{j \geq 2}(1 - 2^{1-j})^{-2} = \prod_{r \geq 1}(1 - 2^{-r})^{-2} < \infty$ is a finite absolute constant.

By Bernoulli's inequality, for all $x_i \in [0, 1]$,

$$(1 - x_1)(1 - x_2) \cdots (1 - x_s) \geq 1 - (x_1 + x_2 + \cdots + x_s).$$

Now observe that $\prod_{r \geq 1}(1 - 2^{-r}) = \tfrac{3}{8} \prod_{r \geq 3}(1 - 2^{-r})$, we have

$$\prod_{r \geq 1}(1 - 2^{-r}) = \frac{3}{8} \prod_{r \geq 3}(1 - 2^{-r}) \geq \frac{3}{8}\Big(1 - \sum_{r=3}^{\infty} 2^{-r}\Big) = \frac{9}{32}$$

Therefore,

$$H(\tfrac{1}{2}) \leq \frac{1}{(9/32)^2} = \frac{1024}{81} \leq 13$$

Thus

$$N_{\text{tot}}(m) \; \leq \; H\big(\tfrac{1}{2}\big) \, (m + 1) \, 2^m \; \leq \; 13m2^m.$$

On the other hand, taking just the term $k = (m, 0, 0, \dots) \in \mathcal{K}_m$ yields

$$N_{\text{tot}}(m) \; \geq \; 2^{|k|} \prod_j (k_j + 1) = 2^m(m + 1) \geq m2^m,$$

so $m2^m \leq N_{\text{tot}}(m) \leq 13m2^m$.

$\square$

We are finally able to provide interpolations for ReLU networks, whose embedding weights echoes with "Pizza" algorithm in (Zhong et al., 2023).

**Theorem I.6** (ReLU construction). *Fix integers $m \geq 1$ and $p \geq 2$. On*
$$\mathcal{X}_m = \{x \in \{0, 1, \dots, m\}^p : \|x\|_1 = m\},$$
*let the target be $y(x) \equiv (\sum_{i=1}^{m} s_i) \bmod p$ for $x = \sum_{i=1}^{m} e_{s_i}$. For any $\tau \in (0, \tfrac{1}{4}]$, there exists a two-layer ReLU network $s^\theta(x) = V \sigma(Wx) \in \mathbb{R}^p$ such that, for all $x \in \mathcal{X}_m$,*
$$h_\theta(x) = \text{uargmax}_{q \in [p]} s_q^\theta(x) = y(x), \qquad s_{y(x)}^\theta(x) - \max_{q \neq y(x)} s_q^\theta(x) \; \geq \; (1 - 4\tau)\, p.$$

*Moreover, the width $d$ is bounded by*
$$d \; \leq \; 13pm2^m \Big( m\sqrt{\tfrac{em}{\tau}}(1 + 2em)^{\frac{m-1}{2}} + 2 \Big), \tag{8}$$

*and the weights satisfy the bounds*
$$\|W\|_\infty \; \leq \; \frac{2}{m}, \qquad \|W\|_F \; \leq \; \frac{2}{m}p\sqrt{13m2^m \Big( m\sqrt{\tfrac{em}{\tau}}(1 + 2em)^{\frac{m-1}{2}} + 2 \Big)}, \qquad \|V\|_\infty \; \leq \; \frac{(m + \tfrac{1}{2})m^{2m}}{m!\, 2^m}. \tag{9}$$

*In addition, the second layer enjoys the spectral-norm bound*
$$\|V\|_2 \; \leq \; \sqrt{p}\sqrt{13m2^m \Big( m\sqrt{\tfrac{em}{\tau}}(1 + 2em)^{\frac{m-1}{2}} + 2 \Big)} \cdot \frac{\sqrt{2}(m + \tfrac{1}{2})m^{2m}}{m!\, 2^m}. \tag{10}$$

*Proof.* For $t \in \mathbb{Z}$, define $c^{\langle t \rangle}, s^{\langle t \rangle} \in \mathbb{R}^p$ by $c_r^{\langle t \rangle} = \cos(2\pi tr/p)$ and $s_r^{\langle t \rangle} = \sin(2\pi tr/p)$ for $r = 0, \ldots, p-1$. For $x = \sum_{i=1}^m e_{s_i}$ and any $j \geq 1$,

$$\sum_{i=1}^m \cos\!\left(j\tfrac{2\pi\nu}{p}s_i\right) = \langle c^{\langle \nu j \rangle}, x \rangle, \qquad \sum_{i=1}^m \sin\!\left(j\tfrac{2\pi\nu}{p}s_i\right) = \langle s^{\langle \nu j \rangle}, x \rangle, \quad \nu \in [p].$$

Fix $\nu \in [p]$ and apply Lem. I.5 to $\theta_i^{(\nu)} = 2\pi\nu s_i/p$. The multi-indices are $\kappa = (\kappa_1, \ldots, \kappa_m) \in \mathcal{K}_m$ and $\pi = (\pi_1, \ldots, \pi_m) \leq \kappa$, and write $r := |\kappa| = \sum_j \kappa_j$. Define

$$u_{\kappa,\pi,\varepsilon}^{(\nu)} = \sum_{j=1}^m \Big(\sum_{\ell=1}^{\pi_j} \varepsilon_{j,\ell}\Big) c^{\langle \nu j \rangle} + \sum_{j=1}^m \Big(\sum_{\ell=\pi_j+1}^{\kappa_j} \varepsilon_{j,\ell}\Big) s^{\langle \nu j \rangle}.$$

Then

$$C_\nu(x) = \sum_{\substack{\kappa,\pi,\varepsilon \\ |\pi|-|\kappa| \text{ even}}} \alpha_{\kappa,\pi,\varepsilon} \, \langle u_{\kappa,\pi,\varepsilon}^{(\nu)}, x \rangle^r, \qquad S_\nu(x) = \sum_{\substack{\kappa,\pi,\varepsilon \\ |\pi|-|\kappa| \text{ odd}}} \beta_{\kappa,\pi,\varepsilon} \, \langle u_{\kappa,\pi,\varepsilon}^{(\nu)}, x \rangle^r,$$

with

$$|\alpha_{\kappa,\pi,\varepsilon}| = |\beta_{\kappa,\pi,\varepsilon}| = \frac{1}{\Big(\prod_{j=1}^m j^{\kappa_j}\Big)\Big(\prod_{j=1}^m \pi_j! \, (\kappa_j - \pi_j)!\Big) r! \, 2^r} \leq \frac{1}{r! \, 2^r}.$$

As $|\langle c^{\langle \nu j \rangle}, x \rangle|, |\langle s^{\langle \nu j \rangle}, x \rangle| \leq m$, we have $|\langle u_{\kappa,\pi,\varepsilon}^{(\nu)}, x \rangle| \leq mr$ and thus $z_{\kappa,\pi,\varepsilon}^{(\nu)}(x) := \langle u_{\kappa,\pi,\varepsilon}^{(\nu)}, x \rangle/(mr) \in [-1, 1]$.

By Lem. I.4, for each $r \in \{1, \ldots, m\}$ and $\delta > 0$ there exists

$$\Phi_r(z) = \sum_{i=1}^{M_r} c_{r,i} \, \mathrm{ReLU}(a_{r,i}z - b_{r,i}), \qquad |a_{r,i}|, |b_{r,i}| \leq 1,$$

such that $\sup_{|z|\leq 1} |z^r - \Phi_r(z)| \leq \delta$ and

$$M_r \leq \frac{m}{\sqrt{2\delta}} + 2, \qquad |c_{r,i}| \leq r + \tfrac{1}{2}. \tag{11}$$

In Equation 11, summing $|\alpha|$ (or $|\beta|$) over $\varepsilon \in \{\pm 1\}^r$ and summing over $\pi \leq \kappa$ factorizes:

$$\sum_{\substack{\pi \leq \kappa \\ \varepsilon \in \{\pm 1\}^r}} |\alpha_{\kappa,\pi,\varepsilon}| = \frac{1}{r!} \cdot \frac{2^r}{\prod_{j=1}^m j^{\kappa_j} \, \kappa_j!}.$$

Summing over $\kappa \in \mathcal{K}_m$ with $|\kappa| = r$ and using the classical cycle-index identity

$$\sum_{\substack{\kappa \in \mathcal{K}_m \\ |\kappa| = r}} \frac{1}{\prod_{j=1}^m j^{\kappa_j} \, \kappa_j!} = \frac{1}{m!} \begin{bmatrix} m \\ r \end{bmatrix},$$

where $\begin{bmatrix} m \\ r \end{bmatrix}$ are the unsigned Stirling numbers of the first kind.

Now we have

$$\sum_{\substack{\kappa,\varepsilon \\ \pi \leq \kappa}} |\alpha_{\kappa,\pi,\varepsilon}| = \sum_{r=1}^m \frac{2^r}{r! \, m!} \begin{bmatrix} m \\ r \end{bmatrix}, \qquad \sum_{\substack{\kappa,\varepsilon \\ \pi \leq \kappa}} |\beta_{\kappa,\pi,\varepsilon}| = \sum_{r=1}^m \frac{2^r}{r! \, m!} \begin{bmatrix} m \\ r \end{bmatrix}.$$

As each power is approximated within $\delta$ and $|\langle u, x \rangle| \leq mr$, the uniform error is bounded by

$$\sum_{r=1}^m \frac{2^r (mr)^r}{r! \, m!} \begin{bmatrix} m \\ r \end{bmatrix} \cdot \delta.$$

We choose

$$\delta := \frac{\tau}{\Lambda_m}, \qquad \Lambda_m := \sum_{r=1}^m \frac{2^r (mr)^r}{r! \, m!} \begin{bmatrix} m \\ r \end{bmatrix},$$

which ensures $\max\{|C_\nu - \widehat{C}_\nu|, |S_\nu - \widehat{S}_\nu|\} \le \tau$ uniformly on $\mathcal{X}$ for all $\nu$.

We now prove by induction on $m$ that

$$\begin{bmatrix} m \\ r \end{bmatrix} \le \binom{m-1}{r-1} m!, \qquad 1 \le r \le m. \tag{12}$$

For $m = 1$, both sides equal 1. Assume equation 12 holds for $m-1$. Using the recurrence $\begin{bmatrix} m \\ r \end{bmatrix} = \begin{bmatrix} m-1 \\ r-1 \end{bmatrix} + (m-1)\begin{bmatrix} m-1 \\ r \end{bmatrix}$,

$$\begin{bmatrix} m \\ r \end{bmatrix} \le \binom{m-2}{r-2}(m-1)! + (m-1)\binom{m-2}{r-1}(m-1)!$$

$$= (m-1)!\left[\binom{m-2}{r-2} + (m-1)\binom{m-2}{r-1}\right]$$

$$\le m\,(m-1)!\binom{m-1}{r-1} = \binom{m-1}{r-1} m!,$$

since $\binom{m-1}{r-1} = \binom{m-2}{r-2} + \binom{m-2}{r-1}$. This proves equation 12.

Using equation 12 and Stirling's lower bound $r! \ge (r/e)^r$, we have

$$\Lambda_m \le \sum_{r=1}^{m} \frac{2^r (mr)^r}{r!}\binom{m-1}{r-1} \le \sum_{r=1}^{m}(2em)^r\binom{m-1}{r-1} = (2em)\sum_{t=0}^{m-1}\binom{m-1}{t}(2em)^t = (2em)\,(1+2em)^{m-1}.$$

Hence

$$\frac{1}{\sqrt{2\delta}} = \sqrt{\frac{\Lambda_m}{2\tau}} \le \sqrt{\frac{em}{\tau}}(1+2em)^{\frac{m-1}{2}}. \tag{13}$$

For $x \in \mathcal{X}$, $\langle \mathbf{1}, x \rangle = m$. So

$$\mathrm{ReLU}\big(a_{r,i} z_{\kappa,\pi,\varepsilon}^{(\nu)}(x) - b_{r,i}\big) = \sigma\left(\left\langle \frac{a_{r,i}}{mr} u_{\kappa,\pi,\varepsilon}^{(\nu)} - \frac{b_{r,i}}{m}\mathbf{1},\ x \right\rangle\right),$$

Each spline unit is a single ReLU of a linear form. Explicitly, $W \in \mathbb{R}^{d\times p}$ has rows $W_{j,:} = \frac{a_{r,i}}{mr} u_{\kappa,\pi,\varepsilon}^{(\nu)} - \frac{b_{r,i}}{m}\mathbf{1}$ for $j = (\nu,\kappa,\pi,\varepsilon,i)$ with $r = |\kappa|$ and $u_{\kappa,\pi,\varepsilon}^{(\nu)} = \sum_{t=1}^{m}\big(\sum_{\ell=1}^{\pi_t}\varepsilon_{t,\ell}\big)c^{\langle\nu t\rangle} + \sum_{t=1}^{m}\big(\sum_{\ell=\pi_t+1}^{\kappa_t}\varepsilon_{t,\ell}\big)s^{\langle\nu t\rangle}$. Since $\|u_{\kappa,\pi,\varepsilon}^{(\nu)}\|_\infty \le r$ and $|a_{r,i}|, |b_{r,i}| \le 1$, each coordinate obeys $|W_{j,t}| \le \frac{|a_{r,i}|}{mr}r + \frac{|b_{r,i}|}{m} \le \frac{1}{m} + \frac{1}{m} = \frac{2}{m}$, hence $\|W\|_\infty \le \frac{2}{m}$.

For class $q \in [p]$ and hidden index $(\nu,\kappa,\pi,\varepsilon,i)$ set

$$V_{q,(\nu,\kappa,\pi,\varepsilon,i)} = \left[\cos\big(\tfrac{2\pi\nu}{p}q\big)\alpha_{\kappa,\pi,\varepsilon} + \sin\big(\tfrac{2\pi\nu}{p}q\big)\beta_{\kappa,\pi,\varepsilon}\right](mr)^r c_{r,i},$$

so that $s_q^\theta(x) = \sum_{\nu=0}^{p-1}\big[\cos(\tfrac{2\pi\nu}{p}q)\widehat{C}_\nu(x) + \sin(\tfrac{2\pi\nu}{p}q)\widehat{S}_\nu(x)\big]$. Let $q^\star \equiv (\sum_i s_i) \bmod p$. Discrete Fourier orthogonality gives $s_q^\star(x) = \sum_{\nu=0}^{p-1}\cos(\tfrac{2\pi\nu}{p}(\sum_i s_i - q)) = \mathbf{1}\{q = q^\star\}p$. Since each mode is within $\tau$, we have $\max_q |s_q^\theta(x) - s_q^\star(x)| \le 2p\tau$ and thus the claimed margin $(1 - 4\tau)p$.

For each fixed $\nu \in [p]$, by Lem. I.5, there are $N_{\mathrm{tot}}(m)$ triples $(\kappa,\pi,\varepsilon)$, each contributes at most $M_r$ units, with $M_r$ bounded in equation 11. Hence for each $\nu$, the width is at most $N_{\mathrm{tot}}(m)\big(\frac{m}{\sqrt{2\delta}} + 2\big)$.

Summing over $\nu = 0, 1, \ldots, p-1$ and using equation 13,

$$d \le p\,N_{\mathrm{tot}}(m)\Big(\frac{m}{\sqrt{2\delta}} + 2\Big) \le p\,N_{\mathrm{tot}}(m)\Big(m\sqrt{\tfrac{em}{\tau}}(1+2em)^{\frac{m-1}{2}} + 2\Big),$$

and the bound $N_{\mathrm{tot}}(m) \le 13m2^m$ gives equation 8.

Thus, $\|W\|_F \le \|W\|_\infty \sqrt{dp} \le \frac{2}{m}p\sqrt{13m2^m\Big(m\sqrt{\tfrac{em}{\tau}}(1+2em)^{\frac{m-1}{2}} + 2\Big)}$.

Finally, using $|\alpha|, |\beta| \le 1/(r!\,2^r)$ and equation 11,

$$|V_{q,(\nu,\kappa,\pi,\varepsilon,i)}| \le |c_{r,i}|(mr)^r \cdot \frac{1}{r!\,2^r} \le \frac{(r+\frac{1}{2})(mr)^r}{r!\,2^r},$$

so taking the maximum over all hidden indices yields equation 9.

For the spectral norm, denote the matrix

$$T = \begin{pmatrix} c^{\langle 0 \rangle} & c^{\langle 1 \rangle} & s^{\langle 1 \rangle} & \cdots & c^{\langle p-1 \rangle} & s^{\langle p-1 \rangle} \end{pmatrix}$$

So

$$TT^\top = c^{(0)}c^{(0)\top} + \sum_{\nu=1}^{p-1} \left( c^{(\nu)}c^{(\nu)\top} + s^{(\nu)}s^{(\nu)\top} \right) = pI_p, \text{ and thus } \|T\|_2 = \sqrt{p}.$$

Index the hidden units by $j = (\nu, \kappa, \pi, \varepsilon, i)$, with $r = |\kappa|$. For that unit, the corresponding column of $V$ was

$$V_{:,j} = \left[ \alpha_{\kappa,\pi,\varepsilon}(mr)^r c_{r,i} \right] c^{\langle \nu \rangle} + \left[ \beta_{\kappa,\pi,\varepsilon}(mr)^r c_{r,i} \right] s^{\langle \nu \rangle}.$$

Hence $V_{:,j}$ is a linear combination of the two columns of $S_\nu$.

Define $B \in \mathbb{R}^{(2p-1) \times d}$, for each column $j = (\nu, \kappa, \pi, \varepsilon, i)$,

$$B_{k,j} = \begin{cases} \alpha_{\kappa,\pi,\varepsilon}(mr)^r c_{r,i}, & k = 0 \text{ and } \nu = 0, \\ \alpha_{\kappa,\pi,\varepsilon}(mr)^r c_{r,i}, & k = 2\nu \text{ with } \nu \in \{1, \ldots, p-1\}, \\ \beta_{\kappa,\pi,\varepsilon}(mr)^r c_{r,i}, & k = 2\nu - 1 \text{ with } \nu \in \{1, \ldots, p-1\}, \\ 0, & \text{otherwise.} \end{cases}$$

One has $V = TB$, and each column $b_j$ of $B$ has support in at most two rows (one when $\nu = 0$). Thus,

$$\|b_j\|_2 = \sqrt{\alpha_{\kappa,\pi,\varepsilon}^2 + \beta_{\kappa,\pi,\varepsilon}^2} \cdot |c_{r,i}| (mr)^r \leq \frac{\sqrt{2}\,(r + \frac{1}{2})(mr)^r}{r!\, 2^r} \leq \frac{\sqrt{2}\,(m + \frac{1}{2})\,m^{2m}}{m!\, 2^m}.$$

Let $n_\nu$ be the number of hidden units at frequency $\nu$. From the construction,

$$n_\nu \leq N_{\text{tot}}(m)\left( \frac{m}{\sqrt{2\delta}} + 2 \right) \leq 13\, m\, 2^m \left( m\sqrt{\frac{em}{\tau}}\,(1 + 2em)^{\frac{m-1}{2}} + 2 \right).$$

Since $BB^\top$ is block diagonal across frequencies, $\|B\|_2 = \max_\nu \|B_\nu\|_2 \leq \max_\nu \sqrt{n_\nu} \cdot \frac{\sqrt{2}(m+\frac{1}{2})m^{2m}}{m!\, 2^m}$. Therefore

$$\|V\|_2 \leq \|T\|_2 \|B\|_2 \leq \sqrt{p}\,\sqrt{\max_\nu n_\nu} \cdot \frac{\sqrt{2}\,(m + \frac{1}{2})\,m^{2m}}{m!\, 2^m},$$

which gives equation 10.

$\square$

**Corollary I.7** (Explicit two-layer ReLU construction for $m = 2$)**.** *Fix $p \geq 2$. Define the input set*

$$\mathcal{X}_2 = \left\{ x \in \{0, 1, 2\}^p : \|x\|_1 = 2 \right\}.$$

*There exists a two-layer ReLU network $s^\theta(x) = V\,\sigma(Wx) \in \mathbb{R}^p$ of width $d = 36p$ such that, for all $x \in \mathcal{X}_2$,*

$$h_\theta(x) = \text{uargmax}_{q \in [p]} s_q^\theta(x) = \left( \sum_{i=1}^2 s_i \right) \bmod p, \qquad s_{y(x)}^\theta(x) - \max_{q \neq y(x)} s_q^\theta(x) \geq \frac{25}{49}p + \frac{20}{49}.$$

*Moreover, the weights satisfy*

$$\|W\|_\infty \leq 1, \qquad \|V\|_\infty \leq \tfrac{34}{7}, \qquad \|V\|_2 \leq 11\sqrt{p}.$$

*Proof.* For $\nu \in [p]$ let $c^{\langle \nu \rangle}, s^{\langle \nu \rangle} \in \mathbb{R}^p$ be defined by $c_r^{\langle \nu \rangle} = \cos(2\pi \nu r/p)$ and $s_r^{\langle \nu \rangle} = \sin(2\pi \nu r/p)$. For inputs $x \in \mathcal{X}$, write

$$C_k = \langle c^{\langle k\nu \rangle}, x \rangle, \qquad S_k = \langle s^{\langle k\nu \rangle}, x \rangle \qquad (k = 1, 2).$$

From Lem. I.5, for any $\theta_1, \theta_2 \in \mathbb{R}$,

$$\cos(\theta_1 + \theta_2) = \frac{1}{2}(C_1^2 - S_1^2 - C_2) = 2(\frac{1}{2}C_1)^2 - 2(\frac{1}{2}S_1)^2 - \frac{1}{2}C_2$$

$$\sin(\theta_1 + \theta_2) = C_1 S_1 - \frac{1}{2}S_2 = 4\left(\left(\frac{C_1 + S_1}{4}\right)^2 - \left(\frac{C_1 - S_1}{4}\right)^2\right) - \frac{1}{2}S_2$$

For $\|x\|_1 = 2$, we have $\frac{C_1}{2}, \frac{S_1}{2}, \frac{C_1 \pm S_1}{4} \in [-1, 1]$.

Let $\Phi_2$ be the piecewise-linear interpolant of $z^2$ on the uniform grid $z_k = -1 + \frac{2k}{7}$, $k = 0, \dots, 7$.

Using Lem. I.4 with $s = 2$, $N = 7$, $\|\Phi_2 - z^2\|_{L_\infty([-1,1])} \leq 1/49$, and $\Phi_2(z) = \sum_{i=1}^8 c_i \operatorname{ReLU}(a_i z - b_i)$, where

| $i$ | 1 | 2 | 3 | 4 | 5 | 6 | 7 | 8 |
|---|---|---|---|---|---|---|---|---|
| $(a_i, b_i)$ | $(1, -1)$ | $(-1, -1)$ | $(1, -\frac{5}{7})$ | $(1, -\frac{3}{7})$ | $(1, -\frac{1}{7})$ | $(1, \frac{1}{7})$ | $(1, \frac{3}{7})$ | $(1, \frac{5}{7})$ |
| $c_i$ | $-\frac{17}{14}$ | $\frac{1}{2}$ | $\frac{4}{7}$ | $\frac{4}{7}$ | $\frac{4}{7}$ | $\frac{4}{7}$ | $\frac{4}{7}$ | $\frac{4}{7}$ |

We now construct a two-layer ReLU MLP with total width $d = 36p$.

**First layer.** For $r \in [p]$ and $i = 1, \dots, 8$ define

$$w_r^{(\nu,1,i)} = \frac{a_i}{2} c_r^{\langle \nu \rangle} - \frac{b_i}{2}, \qquad w_r^{(\nu,2,i)} = \frac{a_i}{2} s_r^{\langle \nu \rangle} - \frac{b_i}{2},$$

$$w_r^{(\nu,3,i)} = \frac{a_i}{4}\left(c_r^{\langle \nu \rangle} + s_r^{\langle \nu \rangle}\right) - \frac{b_i}{2}, \quad w_r^{(\nu,4,i)} = \frac{a_i}{4}\left(c_r^{\langle \nu \rangle} - s_r^{\langle \nu \rangle}\right) - \frac{b_i}{2},$$

$$w_r^{(\nu, C_2^\pm)} = \pm \frac{1}{2} c_r^{\langle 2\nu \rangle}, \qquad w_r^{(\nu, S_2^\pm)} = \pm \frac{1}{2} s_r^{\langle 2\nu \rangle}.$$

Then $\sigma(\langle w^{(\nu,1,i)}, x \rangle) = \operatorname{ReLU}(a_i C_1/2 - b_i)$, etc. Since $|a_i| \leq 1$, $|b_i| \leq 1$, and $|c_r^{\langle \nu \rangle}|, |s_r^{\langle \nu \rangle}| \leq 1$, we have $\|W\|_\infty \leq 1$.

**Second layer.** For $q \in [p], \nu \in [p]$ set

$$\begin{aligned} V_{q,(\nu,1,i)} &= +2c_i \cos(2\pi\nu q/p), & V_{q,(\nu,2,i)} &= -2c_i \cos(2\pi\nu q/p), \\ V_{q,(\nu,3,i)} &= +4c_i \sin(2\pi\nu q/p), & V_{q,(\nu,4,i)} &= -4c_i \sin(2\pi\nu q/p), \end{aligned} \qquad i = 1, \dots, 8,$$

and

$$V_{q,(\nu,C_2^\pm)} = \mp \cos(2\pi\nu q/p), \qquad V_{q,(\nu,S_2^\pm)} = \mp \sin(2\pi\nu q/p).$$

We have $\|V\|_\infty \leq \max\{|4c_i|, 1\} = \frac{34}{7}$.

Let $T = [\, c^{\langle 0 \rangle} \; c^{\langle 1 \rangle} \; s^{\langle 1 \rangle} \; \cdots \; c^{\langle p-1 \rangle} \; s^{\langle p-1 \rangle} \,]$ and write $V = TB$. Then

$$TT^\top = c^{\langle 0 \rangle} c^{\langle 0 \rangle \top} + \sum_{\nu=1}^{p-1}\left(c^{\langle \nu \rangle} c^{\langle \nu \rangle \top} + s^{\langle \nu \rangle} s^{\langle \nu \rangle \top}\right) = p\, I_p,$$

so $\|T\|_2 = \sqrt{p}$.

Each hidden unit loads a single row in $B$, hence $BB^\top$ is diagonal. The largest row norm equals $\sqrt{2 \sum_{i=1}^8 (4c_i)^2 + 2} = \frac{\sqrt{5874}}{7}$, so

$$\|V\|_2 \leq \|T\|_2 \|B\|_2 \leq 11\sqrt{p}.$$

Finally, define

$$\widehat{C}_\nu(x) = 2\,\Phi_2\big(\tfrac{C_1}{2}\big) - 2\,\Phi_2\big(\tfrac{S_1}{2}\big) - \tfrac{1}{2}C_2, \quad \widehat{S}_\nu(x) = 4\,\Phi_2\big(\tfrac{C_1+S_1}{4}\big) - 4\,\Phi_2\big(\tfrac{C_1-S_1}{4}\big) - \tfrac{1}{2}S_2,$$

and logits $s_q^\theta(x) = \sum_{\nu=0}^{p-1} \left[ \cos(2\pi\nu q/p)\, \widehat{C}_\nu(x) + \sin(2\pi\nu q/p)\, \widehat{S}_\nu(x) \right]$. Since $\|\Phi_2 - z^2\|_\infty \leq 1/49$ and $\nu = 0$ contributes a class-independent offset, for $\nu \geq 1$,

$$|\widehat{C}_\nu - C_\nu| \leq 4/49 \quad \text{and} \quad |\widehat{S}_\nu - S_\nu| \leq 8/49.$$

Therefore,

$$\max_q |s_q^\theta(x) - s_q^\star(x)| \leq \tfrac{12}{49}(p-1) + \frac{2}{49},$$

where $s_q^\star(x) = \sum_{\nu=0}^{p-1} \cos\left(2\pi\nu(\sum_i s_i - q)/p\right)$ satisfies $s_{y(x)}^\star(x) = p$ and $s_q^\star(x) = 0$ if $q \neq y(x)$. The margin follows:

$$s_{y(x)}^\theta(x) - \max_{q \neq y(x)} s_q^\theta(x) \;\geq\; p - 2\left(\tfrac{12}{49}(p-1) + \frac{2}{49}\right) = \frac{25}{49}p + \frac{20}{49}.$$

$\square$

## J MARGIN BOUNDS VIA $\ell_\infty$ VECTOR CONTRACTION

### J.1 MARGIN SURROGATES AND EMPIRICAL $\gamma$-MARGIN ERROR

Given scores $s \in \mathbb{R}^p$ for an example with label $y \in [p]$, the *sample margin error*

$$\phi_y(s) = \max_{k \neq y}(s_k - s_y)$$

The $\gamma$-*ramp loss*

$$\psi_\gamma(u) = \min\{1,\ \max\{0,\ 1 + u/\gamma\}\} \in [0, 1].$$

The map $u \mapsto \psi_\gamma(u)$ is $1/\gamma$-Lipschitz on $\mathbb{R}$, and $\phi_y$ is 2-Lipschitz w.r.t. $\|\cdot\|_\infty$ (changing any coordinate of $s$ by at most $\varepsilon$ changes $\phi_y$ by at most $2\varepsilon$), hence

$$g_y := \psi_\gamma \circ \phi_y \quad \text{is} \quad \tfrac{2}{\gamma}\text{-Lipschitz w.r.t. } \|\cdot\|_\infty, \qquad g_y \in [0, 1].$$

**Definition J.1** (Empirical Margin Error). *For a score function $s^\theta$ and sample $S = \{(x^{(i)}, y^{(i)})\}_{i=1}^n$, the* empirical $\gamma$-margin error *is*

$$\widehat{\mathcal{R}}_\gamma(s^\theta; S) \;=\; \frac{1}{n}\sum_{i=1}^n \mathbf{1}\left\{ s^\theta\big(x^{(i)}\big)_{y^{(i)}} \leq \gamma + \max_{j \neq y^{(i)}} s^\theta\big(x^{(i)}\big)_j \right\}.$$

For an interpolating solution, it suffices to take $\gamma = \gamma_\theta(S)$, the minimum sample margin, in which case $\widehat{\mathcal{R}}_\gamma(s^\theta; S) = 0$.

**Definition J.2** (Empirical Rademacher complexity). *Let $S = \left\{ z_i = (x^{(i)}, y^{(i)}) \right\}_{i=1}^n$ be fixed, and let $\mathcal{G} \subset [0,1]^{\mathcal{Z}}$. Let $\epsilon = (\epsilon_1, \ldots, \epsilon_n)$ be i.i.d. Rademacher variables ($\mathbb{P}[\epsilon_i = 1] = \mathbb{P}[\epsilon_i = -1] = 1/2$). The empirical Rademacher complexity of $\mathcal{G}$ on $S$ is*

$$\mathfrak{R}_S(\mathcal{G}) \;=\; \frac{1}{n}\mathbb{E}_\epsilon\left[ \sup_{g \in \mathcal{G}} \sum_{i=1}^n \epsilon_i\, g(z_i) \right].$$

**Theorem J.3** (Rademacher Generalization Bounds, Thm. 3.3 of (Mohri et al., 2018)). *Let $\mathcal{D}$ be the true distribution, $\mathcal{G} \subset [0,1]^{\mathcal{Z}}$ and let $S = (z_1, \ldots, z_n) \sim \mathcal{D}^n$. With probability at least $1 - \delta$ over $S$, the following holds simultaneously for all $g \in \mathcal{G}$:*

$$\mathbb{E}_{z \sim \mathcal{D}}[g(z)] \;\leq\; \frac{1}{n}\sum_{i=1}^n g(z_i) \;+\; 2\,\mathfrak{R}_S(\mathcal{G}) \;+\; 3\sqrt{\frac{\ln(2/\delta)}{2n}},$$

*where $\mathfrak{R}_S(\mathcal{G})$ is the empirical Rademacher complexity of $\mathcal{G}$ on $S$.*

Apply Thm. J.3 with $\mathcal{G} = \mathcal{F}_\gamma := \{(x,y) \mapsto \psi_\gamma \circ \phi_y(f(x)) : f \in \mathcal{F}\}$, and note $\mathbf{1}\{\operatorname{uargmax}_i f_i(x) \neq y\} \leq \psi_\gamma \circ \phi_y\big(f(x)\big)$. That yields the following corollary:

**Corollary J.4** (Rademacher complexity and Multiclassification).

$$\mathbb{P}_{(x,y)\in\mathcal{D}}\big[f(x)\neq y\big] \leq \widehat{\mathcal{R}}_{\gamma}(f) + 2\,\mathfrak{R}_S(\mathcal{F}_\gamma) + 3\sqrt{\frac{\ln(2/\delta)}{2n}}. \tag{14}$$

Let $S = \{(x^{(i)}, y^{(i)})\}_{i=1}^n$ be the training sample generated from the true distribution and write

$$Q_2(S) \;=\; \Big(\tfrac{1}{n}\sum_{i=1}^n \|x^{(i)}\|_2^2\Big)^{1/2}.$$

### J.2 MARGIN BOUNDS FOR SINE MLP

**Definition J.5** (Covering Number for sets). Let $(X, d)$ be a metric space, $F \subseteq X$ a non-empty subset, and $r > 0$. The covering number of $F$, denoted $\mathcal{N}(F, d, r)$, is

$$\mathcal{N}(F, d, r) = \min\left\{k \in \mathbb{N} \mid \exists\{x_1, \ldots, x_k\} \subseteq X \text{ such that } F \subseteq \bigcup_{i=1}^k B_d(x_i, r)\right\},$$

where $B_d(x, r) = \{y \in X \mid d(x, y) \leq r\}$ is the closed ball of radius $r$ centered at $x$.

**Definition J.6** (Empirical $L_2$ covering number of a function class). Let $\mathcal{F} \subseteq \{f : \mathcal{X} \to \mathbb{R}\}$ be a class of real-valued functions and let $x_{1:n} = (x_1, \ldots, x_n) \in \mathcal{X}^n$. Define the empirical $L_2$ metric

$$d_{2,x_{1:n}}(f, g) \;:=\; \Big(\frac{1}{n}\sum_{i=1}^n \big(f(x_i) - g(x_i)\big)^2\Big)^{1/2}.$$

For $\varepsilon > 0$, the empirical $L_2$ covering number of $\mathcal{F}$ at scale $\varepsilon$ with respect to the sample $x_{1:n}$ is

$$\mathcal{N}_2(\varepsilon, \mathcal{F}, x_{1:n}) \;:=\; \min\Big\{k \in \mathbb{N} \;:\; \exists\, f_1, \ldots, f_k \text{ such that } \mathcal{F} \subseteq \bigcup_{j=1}^k B_{d_{2,x_{1:n}}}(f_j, \varepsilon)\Big\},$$

where $B_{d_{2,x_{1:n}}}(f, \varepsilon) = \{g : d_{2,x_{1:n}}(f, g) \leq \varepsilon\}$.

**Lemma J.7** (Covering the box $[-\pi, \pi)^p$ by Euclidean balls). *Fix $p \in \mathbb{N}$ and $r > 0$. Then*

$$\mathcal{N}\big([-\pi, \pi)^p, \|\cdot\|_2, r\big) \;\leq\; \Big\lceil\frac{\pi\sqrt{p}}{r}\Big\rceil^p.$$

*Proof.* Covering numbers are translation invariant: for any $a \in \mathbb{R}^p$, $\mathcal{N}(F, \|\cdot\|_2, r) = \mathcal{N}(F + a, \|\cdot\|_2, r)$. Hence it suffices to cover $[0, 2\pi)^p$.

Set the grid step $h := 2r/\sqrt{p}$ and the number of points per dimension $m := \lceil 2\pi/h \rceil = \lceil \pi\sqrt{p}/r \rceil$. Along each coordinate, place grid points with a half-step offset from the origin:

$$G_1 := \big\{(j + \tfrac{1}{2})h \;:\; j = 0, 1, \ldots, m-1\big\},$$

so $|G_1| = m$. Let the full grid be the Cartesian product $G := G_1^p$; then $|G| = m^p$.

Given any point $x \in [0, 2\pi)^p$, choose $g \in G$ by rounding each coordinate of $x$ to the nearest point in $G_1$ (breaking ties arbitrarily). By construction, the distance from any coordinate $x_i$ to its corresponding grid point $g_i$ is at most half the grid step, so $\|x - g\|_\infty \leq h/2 = r/\sqrt{p}$. We have

$$\|x - g\|_2 \leq \sqrt{p}\,\|x - g\|_\infty \leq \sqrt{p}\cdot\frac{r}{\sqrt{p}} = r.$$

Therefore, the set of closed $\ell_2$-balls $\{B_2(g, r) : g \in G\}$ covers the box $[0, 2\pi)^p$, and

$$\mathcal{N}\big([0, 2\pi)^p, \|\cdot\|_2, r\big) \leq |G| = m^p = \Big(\Big\lceil\frac{\pi\sqrt{p}}{r}\Big\rceil\Big)^p.$$

$\square$

**Lemma J.8** (Standard Dudley entropy integral ). *Assume that all $\mathcal{F}_{x_{1:n}} \subset \mathbb{R}^n$. Let $\mathfrak{R}_n(\mathcal{F})$ be the empirical Rademacher number of $\mathcal{F}$ on $x_{1:n}$. We have:*

$$\mathfrak{R}_n(\mathcal{F}) \leq \inf_{\alpha \geq 0} \left( 4\alpha + 12 \int_\alpha^\infty \sqrt{\frac{\log N_2(\epsilon, \mathcal{F}, x_{1:n})}{n}} d\epsilon \right)$$

**Theorem J.9** (Width-independent multiclass margin bound for the sine MLP). *Consider the two-layer sine network with parameters $\theta = (W, V) \in \mathbb{R}^{d \times p} \times \mathbb{R}^{p \times d}$, where the output matrix satisfies $\|V\|_\infty \leq S_1$. Then for any $\gamma > 0$ and $\delta \in (0, 1)$, with probability at least $1 - \delta$ over the random draw of the training samples $S$, the following holds simultaneously for all such $\theta$:*

$$\mathbb{P}_{(X,Y) \sim \mathcal{D}}\big[h_\theta(X) \neq Y\big] \leq \widehat{\mathcal{R}}_\gamma(s^\theta) + \widetilde{\mathcal{O}}\left( \frac{S_1}{\gamma} \cdot \frac{p}{\sqrt{n}} \right) + \widetilde{\mathcal{O}}\left( \frac{1}{\sqrt{n}} \right).$$

*Proof.* Because inputs are bag of words ($x \in \{0, 1, \ldots, m\}^p$ with $\|x\|_1 = m$), shifting any element of $W$ by $2\pi k$ ($k \in \mathbb{Z}$) does not change $s^\theta(x) = V \sin(Wx)$. Hence without loss of generality, each element of $W$ may be reduced to modulo $2\pi$ to $[-\pi, \pi)$ with no effect on the model output. This periodic reduction is the core argument in the sine analysis.

Notice that $g_y := \psi_\gamma \circ \phi_y$ is $\frac{2}{\gamma}$-Lipschitz w.r.t. $\|\cdot\|_\infty$ and $g_y \in [0, 1]$. Applying Thm. J.3 with $\mathcal{G} = \mathcal{F}_\gamma := (x, y) \mapsto g_y(s^\theta(x)) : \theta$ and recalling $\mathbf{1}\{\mathrm{uargmax} f \neq y\} \leq g_y$, we obtain

$$\mathbb{P}\big[h_\theta(X) \neq Y\big] \leq \widehat{\mathcal{R}}_\gamma(s^\theta) + 2\mathfrak{R}_S(\mathcal{F}_\gamma) + 3\sqrt{\frac{\ln(2/\delta)}{2n}}. \tag{15}$$

$\ell_\infty$ **vector contraction**. Let $\mathcal{S} := \{s^\theta : \theta = (W, V), \|V\|_\infty \leq S_1, W \in [-\pi, \pi)^{d \times p}\}$ and denote the coordinate classes

$$\mathcal{S}|_j := \big\{ x \mapsto v_j^\top \sin(Wx) : \|v_j\|_1 \leq S_1, W \in [-\pi, \pi)^{d \times p} \big\}.$$

For fixed $S = (x^{(1)}, \ldots, x^{(n)})$ and the Lipschitz maps $\varphi_i \equiv g_{y^{(i)}}$ (each $\frac{2}{\gamma}$-Lipschitz w.r.t. $\|\cdot\|_\infty$), the $\ell_\infty$ vector contraction inequality (Thm. 1 of (Foster & Rakhlin, 2019)) gives

$$\mathfrak{R}_S(\mathcal{F}_\gamma) \leq C \frac{2}{\gamma} \sqrt{p} \max_{j \in [p]} \mathfrak{R}_S(\mathcal{S}|_j) \log^{\frac{3}{2} + \delta_0} \left( \frac{\beta}{\max_j \mathfrak{R}_S(\mathcal{S}|_j)} \right), \tag{16}$$

for any fixed $\delta_0 > 0$ and some $C = C(\delta_0)$. Since $\sin(Wx) \in [-1, 1]^d$ and $\|v_j\|_1 \leq S_1$, we have

$$\|s^\theta(x)\|_\infty \leq S_1, \text{ and thus } \beta \leq 1 + S_1. \tag{17}$$

**Coordinate reduction via $\ell_1$-$\ell_\infty$ duality**. For any fixed $S = (x^{(1)}, \ldots, x^{(n)})$ and $j \in [p]$,

$$n\mathfrak{R}_S(\mathcal{S}|_j) = \mathbb{E}_\epsilon \sup_{\substack{\|v_j\|_1 \leq S_1 \\ W \in \mathbb{R}^{d \times p}}} \sum_{i=1}^n \epsilon_i v_j^\top \sin(Wx^{(i)})$$

$$= \mathbb{E}_\epsilon \sup_{\substack{\|v_j\|_1 \leq S_1 \\ W \in \mathbb{R}^{d \times p}}} v_j^\top \left( \sum_{i=1}^n \epsilon_i \sin(Wx^{(i)}) \right)$$

$$\leq S_1 \mathbb{E}_\epsilon \sup_{W \in \mathbb{R}^{d \times p}} \left\| \sum_{i=1}^n \epsilon_i \sin(Wx^{(i)}) \right\|_\infty$$

$$= S_1 \mathbb{E}_\epsilon \sup_{w \in \mathbb{R}^p} \left| \sum_{i=1}^n \epsilon_i \sin(w^\top x^{(i)}) \right|$$

$$= S_1 \mathbb{E}_\epsilon \sup_{w \in \mathbb{R}^p} \sum_{i=1}^n \epsilon_i \sin(w^\top x^{(i)})$$

$$= S_1 \mathbb{E}_\epsilon \sup_{w \in [-\pi, \pi)^p} \sum_{i=1}^n \epsilon_i \sin(w^\top x^{(i)})$$

$$= S_1 n \mathfrak{R}_S(\mathcal{F}_{\sin}). \tag{18}$$

Here we used $\sup_{\|a\|_1 \le S_1} \langle a, b \rangle = S_1 \|b\|_\infty$, and denoted the single-sine family

$$\mathcal{F}_{\sin} := \left\{ x \mapsto \sin(w^\top x) : w \in [-\pi, \pi)^p \right\}.$$

**Rademacher complexity of the single-sine family.**

Endow $\mathcal{F}_{\sin}$ with the empirical $L_2$ metric

$$d(w, w')^2 := \frac{1}{n} \sum_{i=1}^{n} \left( \sin(w^\top x^{(i)}) - \sin(w'^\top x^{(i)}) \right)^2.$$

Notice that $d(w, w') \le 2$ for all $w, w'$, so for any $\varepsilon \in (2, \infty)$, $\mathcal{N}_2(\varepsilon, \mathcal{F}_{\sin}, x_{1:n}) = 1$.

For any $i$,

$$\left| \sin(w^\top x^{(i)}) - \sin(w'^\top x^{(i)}) \right| \le \left| (w - w')^\top x^{(i)} \right| \le \|w - w'\|_2 \|x^{(i)}\|_2 \le m \|w - w'\|_2$$

so if $\|w - w'\|_2 \le \varepsilon/m$ then $d(w, w') \le \varepsilon$. Consequently, for any $\varepsilon \in (0, 2]$,

$$\mathcal{N}_2(\varepsilon, \mathcal{F}_{\sin}, x_{1:n}) \le \mathcal{N}\left( [-\pi, \pi)^p, \|\cdot\|_2, \varepsilon/m \right) \le \left\lceil \frac{\pi m \sqrt{p}}{\varepsilon} \right\rceil^p, \tag{19}$$

where we used Lem. J.7.

Applying the standard Dudley entropy integral with any $\alpha \in (0, 1]$ yields

$$\mathfrak{R}_S(\mathcal{F}_{\sin}) \le 4\alpha + 12 \int_\alpha^2 \sqrt{\frac{\log \mathcal{N}_2(\varepsilon, \mathcal{F}_{\sin}, x_{1:n})}{n}} \, d\varepsilon \tag{20}$$

Let $C := \pi m \sqrt{p} > 2$. Then $\left\lceil \frac{\pi m \sqrt{p}}{\varepsilon} \right\rceil \le \frac{\pi m \sqrt{p}}{\varepsilon} + 1 \le \frac{2\pi m \sqrt{p}}{\varepsilon}$, for all $\varepsilon \in (0, 2]$. Thus

$$\log \mathcal{N}_2(\varepsilon, \mathcal{F}_{\sin}, x_{1:n}) \le \log\left( \left\lceil \frac{\pi m \sqrt{p}}{\varepsilon} \right\rceil^p \right) \le p \log\left( \frac{2\pi m \sqrt{p}}{\varepsilon} \right)$$

Hence for any $\alpha \in (0, 1]$,

$$\int_\alpha^2 \sqrt{\frac{\log \mathcal{N}_2(\varepsilon, \mathcal{F}_{\sin}, x_{1:n})}{n}} \, d\varepsilon \le \int_\alpha^2 \sqrt{\frac{p}{n} \log\left( \frac{2\pi m \sqrt{p}}{\varepsilon} \right)} d\varepsilon \le (2 - \alpha)\sqrt{\frac{p}{n} \log\left( \frac{2\pi m \sqrt{p}}{\alpha} \right)}$$

Plugging this into equation 20 gives

$$\mathfrak{R}_S(\mathcal{F}_{\sin}) \le 4\alpha + 12(2 - \alpha)\sqrt{\frac{p}{n} \log\left( \frac{2\pi m \sqrt{p}}{\alpha} \right)}$$

Choosing $\alpha = \frac{1}{\pi m n \sqrt{p}} \in (0, 1]$. Then

$$\log\left( \frac{2\pi m \sqrt{p}}{\alpha} \right) = \log\left( 2\pi m \sqrt{p} \cdot \pi m n \sqrt{p} \right) = \log(2\pi^2 m^2 p n),$$

So

$$\mathfrak{R}_S(\mathcal{F}_{\sin}) \le \frac{4}{\pi m n \sqrt{p}} + 24\sqrt{\frac{p}{n}} \sqrt{\log\left( 2\pi^2 m^2 p n \right)} = \widetilde{\mathcal{O}}\left( \sqrt{\frac{p}{n}} \right). \tag{21}$$

Combining equation 18 and equation 21 we obtain, for every $S$,

$$\max_{j \in [p]} \mathfrak{R}_S(\mathcal{S}|_j) \le S_1 \mathfrak{R}_S(\mathcal{F}_{\sin}) = \widetilde{\mathcal{O}}\left( S_1 \sqrt{\frac{p}{n}} \right). \tag{22}$$

Fix $\delta_0 = \frac{1}{2}$, substituting equation 22 into equation 15 yields

$$\mathbb{P}\left[ h_\theta(X) \ne Y \right] \le \widehat{\mathcal{R}}_\gamma(s^\theta) + \widetilde{\mathcal{O}}\left( \frac{S_1}{\gamma} \cdot \frac{p}{\sqrt{n}} \right) + \widetilde{\mathcal{O}}\left( \frac{1}{\sqrt{n}} \right).$$

$\square$

### J.3 Margin bounds for ReLU MLP

**Lemma J.10.** *Let $Z \in \mathbb{R}^{p \times n}$ be the data matrix whose $i$-th column is $z_i = \sum_{k=1}^{m} e_{s_{i,k}} \in \{0, 1, \ldots, m\}^p$. Let $N_{j\ell} = \sum_{i=1}^{n} \mathbf{1}\{s_{i,j} = s_{i,\ell}\}$. Then $\|Z\|_F^2 = \sum_{j=1}^{m} \sum_{\ell=1}^{m} N_{j\ell}$.*

*Proof.* Write $Z = \sum_{j=1}^{m} Z_j$ where $Z_j := (e_{s_{1,j}}, \ldots, e_{s_{n,j}}) \in \mathbb{R}^{p \times n}$. Then

$$\|Z\|_F^2 = \Big\langle \sum_{j=1}^{m} Z_j, \sum_{\ell=1}^{m} Z_\ell \Big\rangle_F = \sum_{j=1}^{m} \sum_{\ell=1}^{m} \mathrm{tr}(Z_j^\top Z_\ell).$$

For $r, c$,

$$(Z_j^\top Z_\ell)_{rc} = \sum_{s=1}^{p} (Z_j)_{sr}(Z_\ell)_{sc} = (e_{s_{r,j}})^\top e_{s_{c,\ell}},$$

so $(Z_j^\top Z_\ell)_{ii} = (e_{s_{i,j}})^\top e_{s_{i,\ell}} = \mathbf{1}\{s_{i,j} = s_{i,\ell}\}$. Hence

$$\mathrm{tr}(Z_j^\top Z_\ell) = \sum_{i=1}^{n} \mathbf{1}\{s_{i,j} = s_{i,\ell}\} = N_{j\ell},$$

and substituting yields $\|Z\|_F^2 = \sum_{j=1}^{m} \sum_{\ell=1}^{m} N_{j\ell}$. $\qquad\square$

**Lemma J.11** (Hoeffding bound). *Assume that for each $i \in [n]$, the symbols $(s_{i,1}, \ldots, s_{i,m})$ are i.i.d. uniform on $[p] := \{1, \ldots, p\}$, and that they are independent across $i$. Let $z_i = \sum_{k=1}^{m} e_{s_{i,k}} \in \mathbb{R}^p$, $Z = (z_1, \ldots, z_n) \in \mathbb{R}^{p \times n}$, and $x^{(i)} := z_i$. Then for any $\delta' \in (0, 1)$, with probability at least $1 - \delta'$,*

$$\sum_{i=1}^{n} \|x^{(i)}\|_2^2 \ \le \ nm\left(1 + \frac{m-1}{p}\right) + m(m-1)\sqrt{\frac{n \log(1/\delta')}{2}},$$

*and therefore*

$$Q_2(S) := \Big(\tfrac{1}{n}\sum_{i=1}^{n} \|x^{(i)}\|_2^2\Big)^{1/2} \le \overline{Q}_2(m, p, n, \delta') := \left[m\left(1 + \frac{m-1}{p}\right) + m(m-1)\sqrt{\frac{\log(1/\delta')}{2n}}\right]^{1/2}.$$

*Proof.* For a fixed $i$, define

$$Y_i \ := \ \sum_{j,\ell=1}^{m} \mathbf{1}\{s_{i,j} = s_{i,\ell}\}.$$

Note that $z_i = \sum_{k=1}^{m} e_{s_{i,k}}$ has coordinates $z_i(c) = \sum_{k=1}^{m} \mathbf{1}\{s_{i,k} = c\}$, hence

$$\|z_i\|_2^2 = \sum_{c=1}^{p} z_i(c)^2 = \sum_{c=1}^{p} \Big(\sum_{j=1}^{m} \mathbf{1}\{s_{i,j} = c\}\Big)\Big(\sum_{\ell=1}^{m} \mathbf{1}\{s_{i,\ell} = c\}\Big) = \sum_{j,\ell=1}^{m} \mathbf{1}\{s_{i,j} = s_{i,\ell}\} = Y_i.$$

Therefore $\sum_{i=1}^{n} \|x^{(i)}\|_2^2 = \sum_{i=1}^{n} \|z_i\|_2^2 = \sum_{i=1}^{n} Y_i$. Observe that

$$\mathbb{E}[Y_i] = \sum_{j=1}^{m} \mathbb{E}\,\mathbf{1}\{s_{i,j} = s_{i,j}\} + \sum_{\substack{j,\ell=1 \\ j \ne \ell}}^{m} \mathbb{E}\,\mathbf{1}\{s_{i,j} = s_{i,\ell}\} = m + m(m-1) \cdot \mathbb{P}[s_{i,1} = s_{i,2}].$$

Since $s_{i,1}, s_{i,2}$ are independent uniform on $[p]$, $\mathbb{P}[s_{i,1} = s_{i,2}] = 1/p$, hence

$$\mathbb{E}[Y_i] = m\left(1 + \frac{m-1}{p}\right), \qquad \mathbb{E}\Big[\sum_{i=1}^{n} Y_i\Big] = n\,m\left(1 + \frac{m-1}{p}\right).$$

Also notice that $m \le Y_i \le m^2$ and $(Y_i)_{i=1}^n$ are independent, let $S_n := \sum_{i=1}^{n} Y_i$. Hoeffding's inequality for independent $Y_i \in [a_i, b_i]$ gives

$$\mathbb{P}[S_n - \mathbb{E}S_n \ge t] \ \le \ \exp\left(-\frac{2t^2}{\sum_{i=1}^{n}(b_i - a_i)^2}\right) = \exp\left(-\frac{2t^2}{n\,(m^2 - m)^2}\right).$$

Set the right-hand side to $\delta'$ and solve for $t$ to get

$$t = (m^2 - m)\sqrt{\frac{n\log(1/\delta')}{2}} = m(m-1)\sqrt{\frac{n\log(1/\delta')}{2}}.$$

Therefore, with probability at least $1 - \delta'$,

$$\sum_{i=1}^{n} \|x^{(i)}\|_2^2 = \sum_{i=1}^{n} Y_i \leq nm\left(1 + \frac{m-1}{p}\right) + m(m-1)\sqrt{\frac{n\log(1/\delta')}{2}}.$$

Dividing by $n$ and taking square roots yields the stated bound on $Q_2(S)$. $\qquad\square$

We now state and prove the width-independent multiclass margin bound for homogeneous activation. The main idea is to use $\ell_\infty$ contraction to reduce the problem to the real output, and then utilize a technical lemma from (Golowich et al., 2017). The core part of the proof is almost identical, and is included only for completeness.

**Lemma J.12** (Lem. 1 of (Golowich et al., 2017)). *Let $\sigma$ be a 1-Lipschitz, positive-homogeneous activation function which is applied element-wise (such as the ReLU). Then for any class of vector-valued functions $\mathcal{F}$, and any convex and monotonically increasing function $g : \mathbb{R} \to [0, \infty)$,*

$$\mathbb{E}_\epsilon \sup_{f\in\mathcal{F},\, W:\|W\|_F\leq R} g\left(\left\|\sum_{i=1}^{m} \epsilon_i \sigma(Wf(x_i))\right\|_2\right) \leq 2\cdot\mathbb{E}_\epsilon \sup_{f\in\mathcal{F}} g\left(R\cdot\left\|\sum_{i=1}^{m}\epsilon_i f(x_i)\right\|_2\right).$$

**Theorem J.13** (Width-independent multiclass margin bound for homogeneous activation). *Assume $p > m$ and $n > m^2$, $n \geq 17$, and $\sigma$ is a 1-Lipschitz, positive-homogeneous activation function. For any $\gamma > 0$ and $\delta \in (0, 1)$, with probability at least $1 - \delta$ over the random draw of the training samples $S$, the following holds simultaneously for all $\theta = (W, V)$ with $\|V\|_2 \leq S_2$ and $\|W\|_F \leq B$,*

$$\mathbb{P}_{(X,Y)\in\mathcal{D}}\big[h_\theta(X) \neq Y\big] \leq \widehat{\mathcal{R}}_\gamma(s^\theta) + \widetilde{\mathcal{O}}\left(\frac{S_2 B}{\gamma}\sqrt{\frac{pm}{n}}\right) + \widetilde{\mathcal{O}}\left(\frac{1}{\sqrt{n}}\right).$$

*Here $\widetilde{\mathcal{O}}(\cdot)$ hides factors polylogarithmic in $n$ and $\delta^{-1}$.*

*Proof of Thm. J.13.* The multiclass margin satisfies $|\phi_y(s) - \phi_y(s')| \leq 2\|s - s'\|_\infty$ for all $s, s'$, hence $g_y := \psi_\gamma \circ \phi_y$ is $\frac{2}{\gamma}$-Lipschitz w.r.t. $\|\cdot\|_\infty$ and $|g_y| \leq 1$.

$\ell_\infty$**-vector contraction**. For a vector class $\mathcal{S} \subset \{x \mapsto s(x) \in \mathbb{R}^p\}$ and $L$-Lipschitz maps $\{\varphi_i\}_{i=1}^n$ w.r.t. $\|\cdot\|_\infty$, a standard $\ell_\infty$ vector contraction inequality (see, e.g., Thm. 1 in (Foster & Rakhlin, 2019)) implies that for the fixed sample $S = (x^{(1)}, \ldots, x^{(n)})$,

$$\mathfrak{R}_S(\varphi \circ \mathcal{S}) := \frac{1}{n}\mathbb{E}_\varepsilon\left[\sup_{s\in\mathcal{S}}\sum_{i=1}^{n}\varepsilon_i\,\varphi_i(s(x^{(i)}))\right] \leq CL\sqrt{p}\max_{j\in[p]}\mathfrak{R}_S(\mathcal{S}|_j)\log^{\frac{3}{2}+\delta_0}\left(\frac{\beta}{\max_j\mathfrak{R}_S(\mathcal{S}|_j)}\right),$$

$$\tag{23}$$

for any fixed $\delta_0 > 0$, with $C = C_{\delta_0} < \infty$. Here

$$\mathfrak{R}_S(\mathcal{S}|_j) := \frac{1}{n}\mathbb{E}_\varepsilon\left[\sup_{s\in\mathcal{S}}\sum_{i=1}^{n}\varepsilon_i\,s_j(x^{(i)})\right], \qquad \beta \geq \sup_\theta\max_i\big\{|\varphi_i(s^\theta(x^{(i)}))|,\, \|s^\theta(x^{(i)})\|_\infty\big\}.$$

Let $\mathcal{S} = \{s^\theta : \|V\|_2 \leq S_2,\, \|W\|_F \leq B\}$ and $\mathcal{S}|_j = \{x \mapsto v_j^\top\sigma(Wx) : \|V\|_2 \leq S_2,\, \|W\|_F \leq B\}$, where $v_j \in \mathbb{R}^d$ is the $j$-th row of $V$. Fix $\lambda > 0$, to be chosen later. For any fixed $x_{1:n}$, the

Rademacher complexity can be upper bounded as

$$n\,\mathfrak{R}_S(\mathcal{S}|_j) = \mathbb{E}_\epsilon \sup_{\substack{\|V\|_2 \leq S_2 \\ \|W\|_F \leq B}} \sum_{i=1}^n \epsilon_i\, v_j^\top \sigma\Big(W x^{(i)}\Big)$$

$$\leq \mathbb{E}_\epsilon \sup_{\substack{\|v_j\|_2 \leq S_2 \\ \|W\|_F \leq B}} \sum_{i=1}^n \epsilon_i\, v_j^\top \sigma\Big(W x^{(i)}\Big) \qquad \text{(Cauchy–Schwarz)}$$

$$\leq \frac{1}{\lambda} \log \mathbb{E}_\epsilon \sup_{\substack{\|v_j\|_2 \leq S_2 \\ \|W\|_F \leq B}} \exp\left(\lambda \sum_{i=1}^n \epsilon_i\, v_j^\top \sigma\Big(W x^{(i)}\Big)\right)$$

$$\leq \frac{1}{\lambda} \log \mathbb{E}_\epsilon \sup_{\substack{\|v_j\|_2 \leq S_2 \\ \|W\|_F \leq B}} \exp\left(\|v_j\|_2 \cdot \lambda \left\|\sum_{i=1}^n \epsilon_i\, \sigma\Big(W x^{(i)}\Big)\right\|_2\right)$$

$$\leq \frac{1}{\lambda} \log \mathbb{E}_\epsilon \sup_{\|W\|_F \leq B} \exp\left(S_2 \cdot \lambda \left\|\sum_{i=1}^n \epsilon_i\, \sigma\Big(W x^{(i)}\Big)\right\|_2\right).$$

Applying Lem. J.12 with the given 1-Lipschitz, positive-homogeneous $\sigma$, $\mathcal{F} = \{f : f(x) = x\}$ (identity class), and $g(t) = \exp(S_2 \lambda t)$, we obtain

$$\frac{1}{\lambda} \log \mathbb{E}_\epsilon \sup_{\|W\|_F \leq B} \exp\left(S_2 \cdot \lambda \left\|\sum_{i=1}^n \epsilon_i \sigma\Big(W x^{(i)}\Big)\right\|_2\right) \leq \frac{1}{\lambda} \log\left(2\,\mathbb{E}_\epsilon \exp\left(S_2 \cdot \lambda B \left\|\sum_{i=1}^n \epsilon_i x^{(i)}\right\|_2\right)\right).$$

Denote $M = S_2 B$, and define the random variable (as a function of $\epsilon = (\epsilon_1, \dots, \epsilon_n)$):

$$Z = M \cdot \left\|\sum_{i=1}^n \epsilon_i x^{(i)}\right\|_2.$$

Then

$$\frac{1}{\lambda} \log\left(2\,\mathbb{E}_\epsilon \exp(\lambda Z)\right) = \frac{\log 2}{\lambda} + \frac{1}{\lambda} \log\left(\mathbb{E}_\epsilon \exp\left(\lambda(Z - \mathbb{E}Z)\right)\right) + \mathbb{E}Z.$$

By Jensen's inequality,

$$\mathbb{E}Z \leq M \sqrt{\mathbb{E}_\epsilon\left[\left\|\sum_{i=1}^n \epsilon_i x^{(i)}\right\|_2^2\right]} = M \sqrt{\mathbb{E}_\epsilon\left[\sum_{i,i'=1}^m \epsilon_i \epsilon_{i'} x_i^\top x_{i'}\right]} = M \sqrt{\sum_{i=1}^n \|x^{(i)}\|_2^2}.$$

Moreover, $Z$ satisfies a bounded-difference condition

$$Z(\epsilon_1, \dots, \epsilon_i, \dots, \epsilon_n) - Z(\epsilon_1, \dots, -\epsilon_i, \dots, \epsilon_n) \leq 2M \|x^{(i)}\|_2,$$

and hence is sub-Gaussian with variance factor $v = M^2 \sum_{i=1}^n \|x^{(i)}\|_2^2$, yielding

$$\frac{1}{\lambda} \log\left(\mathbb{E}_\epsilon \exp \lambda(Z - \mathbb{E}Z)\right) \leq \frac{\lambda M^2}{2} \sum_{i=1}^n \|x^{(i)}\|_2^2.$$

Choosing $\lambda = \frac{\sqrt{2 \log 2}}{M \sqrt{\sum_{i=1}^n \|x^{(i)}\|_2^2}}$ gives

$$\frac{1}{\lambda} \log\left(2 \cdot \mathbb{E}_\epsilon \exp(\lambda Z)\right) \leq M\left(\sqrt{2 \log 2} + 1\right) \sqrt{\sum_{i=1}^n \|x^{(i)}\|_2^2}.$$

Therefore,

$$\mathfrak{R}_S(\mathcal{S}|_j) \leq S_2 B\left(\sqrt{2 \log 2} + 1\right) \frac{1}{\sqrt{n}} \sqrt{\frac{1}{n} \sum_{i=1}^n \|x^{(i)}\|_2^2}. \tag{24}$$

**Controlling $\max_j \mathfrak{R}_S(\mathcal{S}|_j)$ and the log term.** Define the "good" subset

$$\mathcal{X}^n_{\text{good}}(\delta') := \Big\{ x_{1:n} \in \mathcal{X}^n : \; \tfrac{1}{n} \sum_{i=1}^n \|x^{(i)}\|_2^2 \leq \overline{Q}_2(m,p,n,\delta')^2 \Big\}.$$

By Lem. J.11, with probability $\geq 1 - \delta'$ the realized sample satisfies $x_{1:n} \in \mathcal{X}^n_{\text{good}}(\delta')$. On this event, equation 24 yields

$$0 \leq \max_{j \in [p]} \mathfrak{R}_S(\mathcal{S}|_j) \; \leq \; S_2 B \left( \sqrt{2 \log 2} + 1 \right) \frac{1}{\sqrt{n}} \overline{Q}_2(m,p,n,\delta'). \tag{25}$$

Furthermore, for any $\theta$ and $x$, $\|s^\theta(x)\|_\infty \leq \|V\|_2 \|\sigma(Wx)\|_2 \leq S_2 B \|x\|_2$, and since here $x \in \{0,1,\ldots,m\}^p$ with $\|x\|_1 = m$, we have $\|x\|_2 \leq m$. Thus we may take the simple, deterministic bound

$$\beta \; \leq \; 1 + S_2 B\, m.$$

To upper bound the logarithm in equation 23 more conveniently, also define

$$b \; := \; 1 + S_2 B \sqrt{n}\, \overline{Q}_2(m,p,n,\delta')\,,$$

so that $\beta \leq b$ and hence $\log(\beta/t) \leq \log(b/t)$ for all $t > 0$.

Applying equation 23 with $L = 2/\gamma$ and using equation 25, we obtain on the event of Lem. J.11

$$\mathfrak{R}_S(\mathcal{F}_\gamma) \leq C \frac{2}{\gamma} \sqrt{p} \, \max_{j \in [p]} \mathfrak{R}_S(\mathcal{S}|_j) \, \log^{\frac{3}{2}+\delta_0} \Big( \frac{\beta}{\max_j \mathfrak{R}_S(\mathcal{S}|_j)} \Big)$$

$$\leq C \frac{2}{\gamma} \sqrt{p} \, \max_{j \in [p]} \mathfrak{R}_S(\mathcal{S}|_j) \, \log^{\frac{3}{2}+\delta_0} \Big( \frac{b}{\max_j \mathfrak{R}_S(\mathcal{S}|_j)} \Big).$$

Let

$$h(t) \; = \; t \, \log^a \Big( \frac{b}{t} \Big), \qquad a := \tfrac{3}{2} + \delta_0 > \tfrac{3}{2}\,.$$

Substituting $\delta_0 = 0.5$ gives $a = 2$. From equation 25, with $t := \max_j \mathfrak{R}_S(\mathcal{S}|_j)$ we have

$$t \; \leq \; \frac{\sqrt{2 \log 2} + 1}{\sqrt{n}} \, S_2 B\, \overline{Q}_2(m,p,n,\delta') \; = \; \frac{\sqrt{2 \log 2} + 1}{n} \, (b-1) \; \leq \; \frac{\sqrt{2 \log 2} + 1}{n} \, b.$$

Since $n \geq 17 \geq e^2(\sqrt{2 \log 2} + 1)$, we have $t \leq b\, e^{-2}$; on $[0, b\, e^{-2}]$ the function $h$ is increasing, hence

$$h(t) \; \leq \; h\Big( \tfrac{\sqrt{2 \log 2}+1}{n}\, b \Big) \; = \; \frac{\sqrt{2 \log 2} + 1}{n} \, b \, \log^2 \Big( \frac{b}{b(\sqrt{2 \log 2} + 1)/n} \Big) \; = \; \frac{\sqrt{2 \log 2} + 1}{n} \, b \, \log^2 \Big( \frac{n}{\sqrt{2 \log 2} + 1} \Big).$$

Therefore, for some absolute $C' > 0$,

$$\mathfrak{R}_S(\mathcal{F}_\gamma) \; \leq \; C' \frac{1}{\gamma} \sqrt{\frac{p}{n}} \, S_2 B\, \overline{Q}_2(m,p,n,\delta') \, \log^2 \Big( \frac{n}{\sqrt{2 \log 2} + 1} \Big). \tag{26}$$

**Final bound.** By Lem. J.11, with probability at least $1 - \delta'$,

$$\overline{Q}_2(m,p,n,\delta')^2 = m\Big( 1 + \frac{m-1}{p} \Big) + m(m-1) \sqrt{\frac{\log(1/\delta')}{2n}} \; \leq \; 2m + m\sqrt{\log(1/\delta')},$$

where we used $p > m$ and $n > m^2$. Hence $\overline{Q}_2(m,p,n,\delta') = \widetilde{\mathcal{O}}(\sqrt{m})$. Since $\frac{1}{n} \sum_{i=1}^n \psi_\gamma(\phi_{y^{(i)}}(s^\theta(x^{(i)}))) \leq \widehat{\mathcal{R}}_\gamma(s^\theta)$, combining equation 14 and equation 26, and taking a union bound with the choice $\delta' = \delta/2$ while applying Cor. J.4 with confidence parameter $\delta/2$, yields the stated result with overall probability at least $1 - \delta$. $\qquad \square$

*Remark* J.14 (Data-dependent specialization). The bound is width-independent and depends on the sample only through $Q_2(S)$. In our setup, $x \in \{0,1,\ldots,m\}^p$ with $\|x\|_1 = m$; thus $\|x\|_2 \leq m$, so $\beta \leq 1 + S_2 Bm$ deterministically. We further used distributional assumptions on $s_{1:m}$ (e.g., i.i.d. uniform over $[p]$) only to obtain sharper high-probability bounds on $Q_2(S)$.

We are now able to prove theorems in Sec. 6:

*Proof of Thm. 6.2.* The proof consists of showing all networks with small training error and small normalized margin generalize, and at least one such network exist.

In Thm. J.9, set $\gamma = \gamma_\theta(\mathcal{D}_{\text{train}})$, then the empirical $\gamma$-margin error is

$$\widehat{\mathcal{R}}_\gamma(s^\theta) = \frac{1}{n}\sum_{i=1}^n \mathbf{1}\left\{ f(x_i)_{y_i} \leq \gamma + \max_{j \neq y_i} f(x_i)_j \right\} = 0.$$

Notice that $\overline{\gamma}_\theta = \frac{\gamma_\theta(\mathcal{D}_{\text{train}})}{\|V\|_{1,\infty}}$, by Thm. J.9,

$$\mathbb{P}_{(X,Y)\in\mathcal{D}}\big[h_\theta(X) \neq Y\big] \leq \widetilde{\mathcal{O}}\left( \frac{1}{\overline{\gamma}_\theta}\, p\sqrt{\frac{1}{n}} \right) + \widetilde{\mathcal{O}}\left( \frac{1}{\sqrt{n}} \right) \leq \widetilde{\mathcal{O}}\left( p\sqrt{\frac{1}{n}} \right) + \widetilde{\mathcal{O}}\left( \frac{1}{\sqrt{n}} \right) = \widetilde{\mathcal{O}}\left( p\sqrt{\frac{1}{n}} \right).$$

When $2p \leq d$, Sec. I.1 gives a network whose normalized margin is

$$\overline{\gamma}_\theta = \frac{\gamma_\theta(\mathcal{D}_{\text{train}})}{\|V\|_{1,\infty}} \geq \frac{p}{2p} = \frac{1}{2} = \Omega(1).$$

$\square$

*Proof of Thm. 6.3.* In Thm. J.13, set $\gamma = \gamma_\theta(\mathcal{D}_{\text{train}})$. Then the empirical $\gamma$-margin error is zero,

$$\widehat{\mathcal{R}}_\gamma(s^\theta) = \frac{1}{n}\sum_{i=1}^n \mathbf{1}\left\{ f(x_i)_{y_i} \leq \gamma + \max_{j \neq y_i} f(x_i)_j \right\} = 0,$$

and Thm. J.13 gives

$$\mathbb{P}_{(X,Y)\in\mathcal{D}}\big[h_\theta(X) \neq Y\big] \leq \widetilde{\mathcal{O}}\left( \frac{1}{\overline{\gamma}_\theta}\sqrt{\frac{p\,m}{n}} \right) + \widetilde{\mathcal{O}}\left( \frac{1}{\sqrt{n}} \right).$$

Apply Thm. I.6 with $\tau = 0.1$, which yields a margin $\gamma(x) \geq 0.6p$ on $\mathcal{X}_m$ and width $d \leq p\,C_m$, where

$$C_m = 13\,m\,2^m\left( m\sqrt{10em}\,(1+2em)^{\frac{m-1}{2}} + 2 \right).$$

Using $(1+2em)^{(m-1)/2} \leq (2em)^{(m-1)/2}e^{1/(4e)}$, we obtain

$$C_m \leq 26\sqrt{5}\,e^{\frac{1}{4e}}\,m^{\frac{m}{2}+2}\,(\sqrt{8e})^m \leq 64\,m^{\frac{m}{2}+2}\,(4.67)^m.$$

Thus the width condition in the statement $d \geq 64\,p\,m^{\frac{m}{2}+2}(4.67)^m$ is sufficient for $d \geq p\,C_m$.

From equation 9–equation 10 in Thm. I.6,

$$\|W\|_F \leq \frac{2}{m}p\sqrt{C_m}, \qquad \|V\|_2 \leq \sqrt{2p}\,\sqrt{C_m}\,\frac{(m+\frac{1}{2})m^{2m}}{m!\,2^m}.$$

Write

$$K_m := C_m\,\frac{(m+\frac{1}{2})m^{2m}}{m!\,2^m}.$$

Using Stirling's lower bound $m! \geq \sqrt{2\pi m}\,(m/e)^m$ and the same $(1+2em)$ bound as above gives the clean upper bound

$$K_m \leq \frac{39\sqrt{5}\,e^{1/(4e)}}{\sqrt{4\pi e}}\,m^{1.5m+2.5}\,(\sqrt{2}\,e^{3/2})^m \leq 17\,m^{1.5m+2.5}\,(6.34)^m.$$

Consequently,

$$\|V\|_2\|W\|_F \leq 2\sqrt{2}\,p\sqrt{p}\,K_m,$$

and

$$\overline{\gamma}_\theta = \frac{\gamma_\theta(\mathcal{D}_{\text{train}})}{\|V\|_2\|W\|_F} \geq \frac{0.6\,p}{2\sqrt{2}\,p\sqrt{p}\,K_m} = \frac{0.3}{\sqrt{2}}\cdot\frac{1}{K_m\sqrt{p}} = \Omega\left( \frac{1}{\sqrt{p}}\cdot\frac{1}{m^{1.5m+2.5}\,(6.34)^m} \right).$$

$\square$

