# OpenReview forum: "Provable Benefits of Sinusoidal Activation for Modular Addition"
_ICLR.cc/2026/Conference — Submitted to ICLR 2026_

### Official Review · Reviewer_EK3F · 2025-10-24

**Soundness:** 3
**Presentation:** 2
**Contribution:** 3
**Rating:** 6
**Confidence:** 3

**Summary:**

This paper investigates the role of activation functions in learning modular addition with two-layer neural networks, comparing sinusoidal (sine) activations against standard ReLU. The authors demonstrate that sine networks achieve superior expressiveness - requiring only width 2 to interpolate modular addition, while ReLU networks need width scaling linearly with the number of summands m. They provide Natarajan-dimension-based generalization bounds showing $\tilde{O}(p)$ sample complexity for constant-width sine networks and margin-based generalization guarantees in the overparameterized regime. Experiments validate that sine networks generalize better than ReLU on modular addition tasks in both underparameterized and overparameterized settings.

**Strengths:**

- Addresses an interesting and important research question. The paper tackles a fundamental question about inductive bias in neural networks - whether periodic structure in the activation can help learn periodic tasks. The choice of modular addition as a testbed is well-motivated since it's a canonical algorithmic reasoning task that exhibits grokking and has been extensively studied mechanistically. The theoretical formalization of this intuition is novel and valuable.
- Impressive and comprehensive theoretical results. The paper provides a complete theoretical picture spanning multiple regimes. The expressivity separation (Theorem 4.1 vs 4.2) cleanly shows sine activation needs only $d=2$ while ReLU requires $d \geq m/p - 1$, which is striking. The Natarajan-dimension analysis (Theorem 5.9) is technically sophisticated, covering piecewise-polynomial, trigonometric-polynomial, and rational-exponential activations in a unified framework. The margin-based bounds (Theorems 6.2-6.3) properly account for optimizer-induced geometry, and the explicit constructions in Appendix F demonstrate that the bounds are tight.
- Well-designed experiments that directly validate the theory. The experimental setup carefully controls for confounds by matching architectures, datasets, and training budgets between sine and ReLU.

**Weaknesses:**

- The paper slightly overclaims its findings. The introduction states, "On periodic tasks, periodic bias increases expressivity and makes learning provably easier" as a general hypothesis, but this is only validated for one highly specific task - modular addition. There's no evidence this extends to other periodic problems like learning trigonometric functions or circular convolutions. The paper doesn't even establish whether 2-layer sine MLPs are universal approximators on standard domains, which limits the generality of the claims.
- Insufficient discussion of practical limitations that limit broader applicability. The paper doesn't adequately address why sinusoidal activations are rarely used in practice despite being proposed decades ago. Almost all successful modern activations (ReLU, GELU, SiLU, Swish, Leaky ReLU) are monotonic or nearly monotonic, which appears important for gradient-based optimization. The sine construction in Theorem 4.1 requires precise weight initialization that may be fragile - small perturbations could destroy the solution. The paper mentions initialization briefly in Appendix B but doesn't analyze sensitivity or provide guidance for practitioners. There's no analysis of optimization dynamics, gradient flow, or trainability that would explain when/why sine networks can actually be learned in practice.
- Some technical gaps and unclear statements. The paper claims "nearly optimal" sample complexity in Corollary 5.11 based on the PAC lower bound in Theorem D.6, but that lower bound assumes a uniformly random label permutation that the learner doesn't know - this is a much harder problem than the standard realizable setting. The connection between this hardness and the ERM upper bound is not clearly established.

**Questions:**

Why do you think monotonic activations dominate in practice if periodic activations have such strong theoretical advantages on periodic tasks? Is there a fundamental tradeoff between expressiveness and trainability that your analysis misses?

---

> ### Author Response · Authors · 2025-11-22
>
> We thank Reviewer EK3F for acknowledging the well-motivated question, comprehensive theory, and controlled experiments. We address the points below.
>
> **Slight overclaiming of the findings.**
>
> We acknowledge that our formal theoretical results apply only to modular addition, a canonical and analyzable periodic task. We explicitly restrict our theoretical validation to this testbed in Section 1 (Introduction). However, there is empirical evidence supporting the general hypothesis from other tasks, many of which are periodic:
>
> > Provable periodic extrapolation. Standard activations (ReLU/tanh/sigmoid) provably fail to extrapolate periodic functions, whereas periodicity-biased or periodic activations (e.g., $\sin/\cos$ or $x+\sin^2(\beta x)$) admit theorems guaranteeing accurate periodic extrapolation beyond the training range—establishing a provable advantage beyond modular addition [5].
> >
>
> > Classification benchmarks. With suitable initialization, sine activations can train faster and sometimes generalize better than monotonic activations across Two Spirals, Separation, and Negation-N [1].
> >
>
> > Image/3D/inverse rendering. Fourier features/positional encodings help standard ReLU MLPs overcome spectral bias on high-frequency/periodic structure [2].
> >
>
> > Images, audio, video, PDEs (Poisson/Helmholtz/wave). Periodic activations with principled initialization (SIREN) effectively represent oscillatory structure and, in reported settings, outperform non-periodic baselines; for Helmholtz, plane-wave (periodic) activations further strengthen this trend [3].
> >
>
> Separately, two-layer sine MLPs are universal approximators on compact domains, as established by universal approximation theorems for continuous, non-polynomial activations [4,6]; our work does not rely on this, so we do not claim broader expressivity than we prove.
>
> **Insufficient discussion of practical limitations that limit broader applicability.**
>
> We focus on statistical learning guarantees; tracking explicit optimization dynamics or gradient flow is beyond the scope of this work. Theorem 4.1 is an expressivity construction, not a training recipe, and can be initialization-sensitive. In practice, periodic activations (e.g., SIREN) often require tailored initialization to preserve gradient flow, and sine nets can be harder to optimize than monotone or near-monotone nets. By contrast, our high-margin interpolation solutions (App. I) are initialization-insensitive, and solutions favored by regularization aligned with the $||V||_{1,\infty}$ geometry are stable to small perturbations. App. B (“Implicit bias of optimizers”) discusses how AdamW and Muon encourage optimizer-induced geometries, offering practical guidance beyond fragile initializations.
>
> **Some technical gaps and unclear statements.**
>
> We modified Theorem F.11 to clarify the near-optimality claim in Remark 5.9: optimality is stated for label-permutation-equivariant learners, with notable examples including AdaGrad, Adam, and GD/SGD with momentum under i.i.d. final-layer initialization.
>
> **Q1:**
>
> (i) **Lack of periodicity in tasks.** In practice, most large-scale vision/NLP targets are non-periodic, and monotone/near-monotone activations (ReLU, GELU, SiLU) offer robust, scalable optimization. Our work analyzes statistical expressivity/generalization for periodic targets and does not model optimization, so we make no training-stability claims.
>
> (ii) **Sensitivity to tuning.** Periodic activations/encodings inject a strong oscillatory bias: on periodic/high-frequency targets they improve approximation and learning efficiency [2,3]. The same specialization raises practical concerns in typical high-dimensional classification: SIREN requires principled initialization and is sensitive to hyperparameters [3]; Fourier bandwidth can push models toward high-frequency overfitting unless regularized, whereas standard ReLU MLPs exhibit a low-frequency spectral bias that often makes optimization more forgiving [7].
>
> Therefore, the gap is a practical robustness–expressivity trade-off, not a fundamental impossibility. Formalizing that tradeoff is outside our scope but is consistent with prior evidence on the spectral bias of ReLU and on recovering high frequencies via Fourier features or SIREN with specialized initialization [3,7].

---

> > ### Author Response · Authors · 2025-11-22
> >
> > **References.**
> >
> > [1] J. M. Sopena, E. Romero, and R. Alquezar, “Neural networks with periodic and monotonic activation functions: a comparative study in classification problems,” Proc. ICANN ’99, pp. 323–328, 1999.
> >
> > [2] M. Tancik et al., “Fourier features let networks learn high frequency functions in low dimensional domains,” NeurIPS, 2020.
> >
> > [3] V. Sitzmann et al., “Implicit neural representations with periodic activation functions,” NeurIPS, 2020.
> >
> > [4] M. Leshno, V. Y. Lin, A. Pinkus, and S. Schocken, “Multilayer feedforward networks with a nonpolynomial activation function can approximate any function,” Neural Networks, 1993.
> >
> > [5] Z. Liu, T. Hartwig, and M. Ueda, “Neural Networks Fail to Learn Periodic Functions and How to Fix It,” NeurIPS, 2020.
> >
> > [6] K. Hornik, “Approximation capabilities of multilayer feedforward networks,” *Neural Networks*, 4(2):251–257, 1991.
> >
> > [7] N. Rahaman et al., “On the spectral bias of neural networks,” ICML, PMLR 97:5301–5310, 2019.

---

### Official Review · Reviewer_fCwo · 2025-10-27

**Soundness:** 3
**Presentation:** 4
**Contribution:** 2
**Rating:** 6
**Confidence:** 2

**Summary:**

This paper provides a rigorous theoretical and empirical investigation of why sinusoidal activations outperform standard nonlinearities such as ReLU on periodic algorithmic tasks, using modular addition as a central test case. The authors first construct a width-2 sine MLP that exactly computes modular addition under a shared input embedding, and they prove that any ReLU MLP realizing the same function must have width growing at least linearly with the number of summands $m$. They then extend this analysis to generalization, deriving Natarajan-dimension bounds for a wide family of activation functions—including piecewise-polynomial, trigonometric, and rational-exponential classes—and showing that for constant-width sine networks, the sample complexity can be as low as$\tilde O(p)$. In the overparameterized setting, they establish width-independent generalization guarantees, proving that sine networks achieve population error $\tilde O(p/\sqrt{n})$ when the normalized margin remains constant. Finally, empirical results confirm that sine networks train faster and generalize better than ReLU networks under matched setups, with normalized margins closely tracking test accuracy.


Overall, the paper offers a clear theoretical link between periodic inductive bias and learnability, demonstrating that sinusoidal activations make modular arithmetic both expressively simpler and statistically more efficient.

**Strengths:**

The paper presents clear and elegant theoretical results—the constructive proof that a width-2 sine MLP can exactly compute modular addition is both strong and intuitive. The generalization analysis is technically solid: the Natarajan-dimension and margin results are well-executed and connect naturally to established learning-theory tools. The empirical findings are also consistent with the theory, as the experiments replicate the same assumptions (shared embeddings, identical optimizers and data) and demonstrate a close match between margin growth and generalization behavior. Finally, the work is highly relevant to mechanistic studies, as it formalizes the periodic patterns often observed in neural networks trained on modular arithmetic and ties them to an explicit architectural bias toward periodic structure.

**Weaknesses:**

- The experimental and modeling setup is fairly restricted. The model operates only on bag-of-tokens count vectors without sequence order and is limited to a two-layer MLP. This design isolates periodicity effectively but omits much of the structure that real-world models, such as Transformers, rely on. As a result, it remains unclear how the theoretical advantages would carry over when tokens are contextualized or order-dependent.

- The study also focuses exclusively on modular addition. While this is a clean and interpretable benchmark, the conclusions would be more compelling if validated on other periodic or quasi-periodic tasks such as parity, sine-wave regression, or modular multiplication, to assess the robustness of the theoretical claims.

- The interpretation of the “constant-width” result could be clarified further. The $\tilde O(p)$ sample complexity bound depends on the assumptions of realizability and a discrete bounded input space. It would be useful to discuss how these guarantees scale under more realistic conditions, such as continuous-valued or noisy inputs.

- Finally, the paper stops short of discussing broader implications. Although it convincingly shows that periodic activations are beneficial for periodic data, it does not explore whether or how such activations might be incorporated into modern architectures, or whether their advantages persist when combined with non-periodic components.

**Questions:**

### Q1
Your analysis assumes a shared, position-independent embedding so the network only observes token counts.
How sensitive do you think your results are to this “bag-of-tokens” assumption?
If the model had positional or additive embeddings instead, would the same expressivity and generalization arguments still hold?


### Q2
The proofs rely on the discrete, periodic nature of modular arithmetic.
Could the same reasoning extend to continuous periodic functions, like $y=\sin(x_1+\cdots+x_m)$?
And if the outputs were real-valued rather than categorical residues, would your margin-based analysis still apply?


### Q3
Have you considered testing sine activations in larger or more realistic models—for instance, deeper MLPs or even Transformers?
It would be interesting to know whether the same periodic bias improves performance on algorithmic or periodic reasoning tasks.


### Q4
For the high-margin variant (width \(2p\)), what’s the trade-off between the minimal width construction and this larger model?
Do you observe margin growth matching the theoretical prediction, or does it plateau empirically?


### Q5
Your experiments use AdamW and Muon optimizers.
Do you expect the main conclusions—particularly the margin and accuracy advantages—to hold if you switched to a simpler optimizer like vanilla SGD?


### Q6
The paper https://arxiv.org/abs/2406.03445 shows that giving LLMs Fourier-style number embeddings helps them learn modular addition quickly, avoiding the grokking phase.
Since your work explores periodic bias through sine activations, can you connect these two ideas?
Could using sine activations inside a Transformer’s feed-forward blocks similarly speed up learning or reduce grokking?
It might be worth a small experiment: train a 1-layer Transformer on modular addition, compare GELU vs. sine vs. a hybrid activation, and track how quickly each reaches high test accuracy and margin growth.

---

> ### Author Response · Authors · 2025-11-22
>
> We thank Reviewer fCwo for recognizing the clear construction, rigorous analysis, the margin–generalization link, and the work’s relevance to mechanistic interpretability. We address the concerns below.
>
> **Restricted setup.**
>
> Our minimal bag-of-tokens, two-layer MLP setup intentionally isolates periodicity for clean analysis. To probe generality, we trained a 1-layer, 1-head decoder-only Transformer on modular addition. With sine FFN activations, it attains high test accuracy with far fewer samples, whereas ReLU/GELU need more data and underperform at moderate budgets (Fig. 6). Extending the theory to contextual, order-dependent models is promising but nontrivial because the normalized margin is not straightforward to define for Transformers; we leave this to future work. For Natarajan bounds, a general form following the template of Theorems 8.4 and 8.14 in [9]—which includes Transformers via Theorem 8.14—can be derived, but proving a concrete advantage in Transformers remains challenging.
>
> **Other periodic or quasi-periodic tasks.**
>
> Our proof focuses on modular addition, the minimal analyzable periodic testbed. The mechanism—periodic extrapolation enabled by periodic/periodicity-biased activations—extends to other periodic or quasi-periodic tasks:
> (i) parity is addition mod 2;
> (ii) sine-wave regression is periodic, where standard activations provably fail to extrapolate while periodic ones succeed [5];
> (iii) modular multiplication over a prime modulus is isomorphic to modular addition via a primitive root/discrete logarithm, so the same analysis applies up to relabeling.
>
> We limit our claims to what we can prove, while noting broad empirical support for the benefits of sinusoidal activation functions for periodic tasks in vision, audio, and PDEs [2, 3, 7].
>
> **Near-optimality clarification.**
>
> We revised Remark 5.9 and Theorem. F.11 to state that near-optimality holds for any label-permutation–equivariant learner, including AdaGrad, Adam, and GD/SGD with momentum under i.i.d.\ final-layer initialization.
>
> Our lower bound is driven by label symmetry, not by whether features are discrete, continuous, or noisy. The argument uses only (i) the learner is label-permutation–equivariant; and (ii) the training labels $Y_i=f(X_i)$ are i.i.d. and uniformly distributed on $[p]$.
>
> Thus, in continuous-valued or noisy settings that preserve these symmetries (i.e., no leakage of label identity through $X$ or noise), the information-theoretic barrier underlying the $\tilde{\mathcal{O}}(p)$ scaling remains. The realizability and bounded, discrete $X$ assumptions were used only to obtain a clean finite-sample PAC bound with explicit exponents; extending those exponents to more general continuous or noisy cases is technically involved and beyond our scope.
>
> **Broader implications.**
>
> Periodic activations can be integrated cleanly into modern architectures. In modular addition, a 1-layer, 1-head decoder-only Transformer with a sine FFN attains high test accuracy with substantially fewer samples; ReLU/GELU need more data and remain weaker at moderate budgets (Fig. 6). This shows the advantages persist even alongside non-periodic components (self-attention).

---

> > ### Author Response · Authors · 2025-11-22
> >
> > **Q1:**
> >
> > Generalization bounds are robust to the “bag-of-tokens” assumption; our expressivity constructions are not.
> >
> > Natarajan bounds: Not sensitive. The proofs for Theorem 5.6 (and the general form, Theorem H.6) extend to positional or additive embeddings with minor edits. Actually, it’s possible to derive a most general form that follows the template of [9], Theorems 8.4 and 8.14.
> >
> > Margin bounds: Not sensitive, since the technique (contraction inequality+ covering number/peeling argument) is used. Constants, normalizations, and the specific high-margin interpolating solution depend on the embedding (positional/additive), so the statements rescale but go through.
> >
> > Expressivity: Sensitive. Our constructions rely on the bag-of-tokens assumption. With positional/additive embeddings, new constructions are required. We view this as an interesting direction rather than a limitation of the bounds.
> >
> > **Q2:**
> >
> > No. Natarajan dimension and multiclass margin bounds address multiclass $0-1$ loss, not regression. Without discretizing outputs, our expressivity results and bounds do not carry over. For real-valued targets (e.g., $y=\sin(x_1+\cdots+x_m)$), the appropriate tools are pseudo-dimension/fat-shattering or Barron-type analyses. For background, see [9], Part three (“Learning Real-Valued Functions”), especially Ch. 16.
> >
> > **Q3:**
> >
> > Yes. On modular addition, a 1-layer, 1-head decoder-only Transformer with a sine-activated FFN reaches high test accuracy with substantially fewer samples than ReLU/GELU baselines (Fig. 6), indicating the periodic bias aids Algorithmic/periodic reasoning and persists alongside non-periodic components (self-attention).
> >
> > **Q4:**
> >
> > Trade-off: width vs. margin. The $d=2p$ high-margin variant yields $\Omega(p)$ raw margins (normalized margins stay $\Theta(1)$), improving perturbation robustness. By contrast, the $d=2$ minimal-width network has a margin of $\Theta(p^{-2})$ (Section E.1). Empirically, normalized margins tend to increase with generalization but may plateau, depending on the optimizer and weight decay.
> >
> > **Q5:**
> >
> > Yes, our accuracy conclusions are unchanged under vanilla SGD; our analysis is optimizer-agnostic.
> >
> > We include vanilla SGD results for a 2-layer MLP in the underparameterized regime (Fig. 9). After interpolation, SGD generalizes more slowly and is more learning-rate sensitive; AdamW reaches higher test accuracy faster, and SGD never catches up. Additionally, sine generalizes better than ReLU at matched width and training budget, consistent with AdamW and Muon.
> >
> > Normalized margins for sine vs. ReLU are not directly comparable because for sine models we apply weight decay only to the second layer, whereas for ReLU models we decay both layers. So we do not assert a margin advantage. Moreover, vanilla SGD’s L2 regularization does not directly control the weight norms of interest, unlike decoupled weight decay.
> >
> > **Q6:**
> >
> > Yes. In our modular addition setup (1-layer, 1-head Transformer), replacing GELU/ReLU with sine in the FFN substantially shortens grokking time (i.e., a faster rise to high test accuracy). Both Fourier-style number embeddings and sine activations inject a periodic inductive bias, reducing grokking. For completeness, we cited [arXiv:2406.03445] and note this connection in Section 2 (Related Work). However, defining a normalized margin for Transformers is nontrivial, quantifying margin growth is valuable, and we leave it to future work.
> >
> > **References.**
> >
> > [1] J. M. Sopena, E. Romero, and R. Alquezar, "Neural networks with periodic and monotonic activation functions: a comparative study in classification problems," Proc. ICANN ’99, pp. 323–328, 1999.
> >
> > [2] M. Tancik et al., "Fourier features let networks learn high frequency functions in low dimensional domains," NeurIPS, 2020.
> >
> > [3] V. Sitzmann et al., "Implicit neural representations with periodic activation functions," NeurIPS, 2020.
> >
> > [4] M. Leshno, V. Y. Lin, A. Pinkus, and S. Schocken, "Multilayer feedforward networks with a nonpolynomial activation function can approximate any function," Neural Networks, 1993.
> >
> > [5] Z. Liu, T. Hartwig, and M. Ueda, "Neural Networks Fail to Learn Periodic Functions and How to Fix It," NeurIPS, 2020.
> >
> > [6] L. Meronen, M. Trapp, and A. Solin, "Periodic Activation Functions Induce Stationarity," NeurIPS, 2021.
> >
> > [7] Z. Wang, T. Cui, and X. Xiang, "A Neural Network with Plane Wave Activation for Helmholtz Equation," arXiv:2012.13870 / Applied Numerical Mathematics, 2022.
> >
> > [8] K. Hornik, "Approximation capabilities of multilayer feedforward networks," *Neural Networks*, vol. 4, no. 2, pp. 251–257, 1991.
> >
> > [9] M. Anthony and P. L. Bartlett, Neural Network Learning: Theoretical Foundations. Cambridge: Cambridge University Press, 1999.

---

### Official Review · Reviewer_ZF3N · 2025-10-28

**Soundness:** 3
**Presentation:** 2
**Contribution:** 2
**Rating:** 4
**Confidence:** 2

**Summary:**

This paper analyzes the approximation power of two-layer neural networks with sine activation on modular addition data. It also presents an ERM bound derived through an analysis of the Natarajan dimension. Furthermore, it performs margin-based generalization in the overparameterized regime.

**Strengths:**

I believe that analyzing activation functions with periodic properties for periodic inputs is fair, natural, and meaningful research. The results presented in the paper appear to be novel in terms of originality, and the rigor of the proofs seems sufficient based on what I have seen.

**Weaknesses:**

1. There are several issues with the presentation of this paper. The meanings of the theorems and the logical flow of the proofs in both the main text and the appendix are not adequately explained. For example, Theorem 5.9 is obtained by appropriately bounding the presented Natarajan dimension and substituting it into an existing theorem. However, readers cannot find this logical flow in the main text before reading the proof. In addition, the appendix provides insufficient high-level explanations of the proofs. It is difficult to tell, before reading them in detail, whether the proofs are novel or simply follow existing proof techniques. A proper outline of the proofs should be added.

2. From an approximation perspective, the fact that the sine activation function approximates modular addition better than ReLU appears evident both in terms of the theorem’s content and its proof, and therefore does not seem to present significant novelty.

3. Even considering that this paper represents a basic attempt to study discrete periodic functions, the fact that its analysis focuses only on modular addition weakens its novelty. In fact, it is unclear why the paper limits its analysis to modular addition. The Natarajan-dimension-based analysis could seemingly be applied to many other types of periodic functions as well.
The reason I think so is that the only factor that seems to prevent Theorem 5.9 from holding for other general discrete data is Lemma E.3, and it appears that a slight modification could make the result generalizable.
If not, the authors should clarify why it is difficult to do so.
In the same context, I think describing the inputs in Table 1 as discrete and bounded is an overstatement, since the theorem applies only to modular addition. If this is not the case, the authors should clarify the issue.
Also, it is a little bit strange that the theorems providing bounds on the Natarajan dimension appear only in the appendix, while Table 1 in the main text presents their results.

Minor
When referring to the appendix, make sure to use a consistent format. For example, in line 195 (and many other places), it is referred to as Appendix B, whereas in line 202 it appears as Section F.1.1 or Section G.

**Questions:**

1. Could the authors clarify whether this result can be generalized to more general forms of discrete data?

**Details Of Ethics Concerns:**

NO or VERY MINOR ethics concerns only.

---

> ### Author Response · Authors · 2025-11-22
>
> We thank Reviewer ZF3N for the positive assessment of our paper’s rigor and novelty. We address the points below.
>
> **Issues with the presentation.**
>
> We now add high-level outlines at the start of Sections 5–6 that clarify the proof flow. We also include Appendix D (Proof Outlines),  which outlines the logical structure of our theoretical results and clarifies the connections between the main-text theorems and the detailed proofs.
>
> **Lack of novelty.**
>
> We agree that periodic activations are a natural fit for periodic tasks. Our contribution is to turn this intuition into rigorous mathematical statements. Concretely, we provide those results listed in the contribution: (i) a sharp expressivity separation between sine and ReLU MLPs; (ii) unified Natarajan-dimension bounds; (iii) width-independent margin guarantees; and (iv) near-optimal ERM sample complexity for constant-width sine networks. Empirically, sine networks extrapolate to lengths far beyond training, while ReLU baselines fall to chance.
>
> **Only focus on modular addition.**
>
> Our focus on modular addition is expository, not a limitation. The core Natarajan-dimension analysis is task-agnostic: Theorem 5.6 (formerly 5.9) extends beyond modular addition, and the more general two-layer MLP bounds appear as Theorem H.6, with setup in Sec. H.1. Lemma G.5 (formerly E.3) is a generic, task-agnostic tool and not the bottleneck.
>
> Extending to arbitrary periodic activations is nontrivial. In the proof, we apply classical bounds of polynomial function classes (Theorem. G.11). The key step, Lemma G.14 (formerly E.11), exploits a trigonometric-to-polynomial connection that enables our bounds, we do not know an analogous reduction for other periodic activations.
>
> Table 1 summarizes the results for a generalized, task-agnostic bound. Full statements and proofs are provided in Appendix H due to space constraints. To enhance clarity, we refer to inputs as integer-valued and bounded rather than discrete and bounded, and we formally define the input regimes in Appendix H.
>
> Bottom line: For two-layer MLPs, we provide task-agnostic Natarajan-dimension bounds for piecewise-polynomial, trigonometric-polynomial, and polynomial–rational–exponential activations (Theorem. H.6).
>
> **Minor.**
>
> We have added explicit cross-references to the main text. For details on notation changes, please see the general response.
>
> **Q1:**
>
> Yes, our results extend beyond modular addition. See our response to “Only focus on modular addition.” and Theorem H.6 (setup in Sec. H.1) in the revised manuscript for task-agnostic bounds for two-layer MLPs over general bounded integer-value inputs. We focus on modular addition in the main paper for clarity.

---

### Official Review · Reviewer_iHML · 2025-10-31

**Soundness:** 2
**Presentation:** 2
**Contribution:** 3
**Rating:** 4
**Confidence:** 4

**Summary:**

This paper investigates the role of activation functions in learning modular addition using two-layer neural networks, comparing sinusoidal (sine) and ReLU activations under a shared, position-independent input embedding setup. The authors demonstrate that sine activations achieve superior expressiveness, requiring only two neurons to exactly implement modular addition versus $\Omega(m/p)$ width for ReLU networks. They provide novel Natarajan-dimension generalization bounds yielding $\widetilde{\mathcal{O}}(\sqrt{dp})$ sample complexity for both activation families in the underparameterized regime, and establish that constant-width sine networks achieve near-optimal $\widetilde{\mathcal{O}}(p)$ sample complexity under ERM. For the overparameterized regime, they derive width-independent margin-based generalization bounds, showing that sine networks naturally attain large normalized margins ($\Omega(1)$) leading to $\widetilde{\mathcal{O}}(p/\sqrt{n})$ population error, while ReLU networks suffer exponentially decaying margins with the number of summands m. Experimental results validate these theoretical findings across both regimes, demonstrating consistent generalization advantages for sine over ReLU networks.

**Strengths:**

1. Strong theoretical framework with novel contributions. The paper makes rigorous theoretical contributions across multiple fronts. The Natarajan-dimension analysis (Theorem 5.9) is novel and broadly applicable, covering piecewise-polynomial, trigonometric-polynomial, and rational-exponential activations in a unified framework through elegant pairwise reduction techniques. The margin-based analysis for sine networks (Theorem 6.2) elegantly exploits the natural geometric properties of sinusoidal activations, establishing width-independent bounds under the $\lVert V\rVert_{1,\infty}$ norm constraint that align well with optimizer-induced geometries. The near-optimal sample complexity result (Theorem 5.11) for constant-width sine networks with matching lower bounds (Theorem D.6) demonstrates completeness of the theoretical picture.


2. Clear motivation and well-designed experimental validation. The paper motivates the study through connections to mechanistic interpretability results showing Fourier-like circuits in networks trained on modular addition, and broader applications of periodic representations across machine learning domains. The experimental design directly tests theoretical predictions with matched architectures, identical datasets, and controlled hyperparameters. The experiments cleanly separate underparameterized and overparameterized regimes, validating both uniform convergence predictions and margin-based generalization bounds. The use of $0.5%$-quantile margins rather than minimum margins to avoid outlier effects shows careful experimental design.

**Weaknesses:**

1. Incomplete comparison with ReLU network. The ReLU margin bound in Theorem 6.3 requires extraordinarily stringent conditions that may not be achievable in practice. Specifically, the theorem requires width $d \ge \frac{64,p^{m}}{m^{2}+2},4.67^{m}$ and normalized margin $\gamma_{\theta,\mathrm{ReLU}}=\Omega!\left(\frac{1}{\sqrt{p}}\cdot\frac{1}{m^{1.5m+2.5},6.34^{m}}\right)$, which involves exponential dependence on $m$ that becomes prohibitively large even for moderate values. For example, with $m=5$, the required margin scales as $\Omega(10^{-9})$, making the comparison potentially unfair. The paper does not discuss whether these conditions are artifacts of the proof technique or inherent limitations of ReLU networks.

2. Weak connection between theory and experiments. While the paper claims the experiments "validate" the theory, there are significant gaps between theoretical predictions and experimental observations that are not adequately addressed. The uniform convergence bound predicts $\widetilde{\mathcal{O}}(\sqrt{dp})$ sample complexity (Theorem 5.9), yet Figure 1 shows accuracy curves as functions of width $d$ for what appears to be fixed training set sizes, without explicitly demonstrating the $\sqrt{dp}$ scaling prediction. The margin-based theory predicts test error $\widetilde{\mathcal{O}}(p/\sqrt{n})$ when normalized margins are $\Omega(1)$ (Theorem 6.2), but Figures 2,3 plot test accuracy against weight decay rates rather than directly validating the predicted scaling with $n$ or $p$.

3. Missing ablations on key parameters. The experiments in Section 7 do not systematically vary the number of summands m or modulus p, which are the primary parameters in the theoretical analysis. The paper claims "we conduct experiments with two-layer sine and ReLU MLPs on modular addition under various settings" (Section 7, line 378), but the figures do not indicate what values of m and p were used.

**Questions:**

Theorem 6.3 requires ReLU networks to have width $d \ge \frac{64,p^{m}}{m^{2}+2},4.67^{m}$ and achieve normalized margins $\gamma_{\theta,\mathrm{ReLU}}=\Omega!\left(\frac{1}{\sqrt{p}}\cdot\frac{1}{m^{1.5m+2.5},6.34^{m}}\right)$. For $m=5$, this margin requirement becomes approximately $\Omega(10^{-9})$, which seems extraordinarily small.
1. Can you clarify whether these exponential dependencies are fundamental limitations of ReLU networks for modular addition, or are they artifacts of your proof technique?
2. Did any of your trained ReLU networks actually satisfy these width requirements and achieve these margin values?

Your theory suggests exponential separation between sine and ReLU (comparing Theorems 6.2 and 6.3), yet your experiments show sine "consistently outperforms" ReLU without quantifying the magnitude of improvement.
3. Can you provide quantitative measurements of the performance gap, such as the percentage by which sine exceeds ReLU test accuracy at specific width d and sample size n?

---

> ### Author Response · Authors · 2025-11-22
>
> We thank Reviewer iHML for the positive assessment and recognition of our theoretical rigor; we clarify the ReLU comparison and the theory–experiment link below.
>
> **Incomplete comparison with ReLU.**
>
> Yes, the conditions are proof artifacts, not inherent limits of ReLU on modular addition. Theorem. 6.3 is a deliberately conservative, weak-positive guarantee: it certifies generalization for small-margin ReLU interpolants under spectral/Frobenius control. The sufficient width $d \ge 64pm^{m/2+2}4.67^m$ and margin $\gamma_{\theta,\mathrm{ReLU}}=\Omega\big(p^{-1/2} m^{-(1.5m+2.5)} 6.34^{-m}\big)$ stem from the constructions in Section I. Empirically (Figs. 3, 11, 13), trained ReLU models achieve margins far above this minute threshold at widths well below the sufficient $d$. We now state this explicitly in Remark 6.4.
>
> **Theory and exps gap.**
>
> Our bounds are upper bounds, not tight predictions, so experiments need not display exact $\tilde O(\sqrt{dp/n})$ or $\tilde O(p/\sqrt{n})$ scalings. With $n$ fixed, Fig. 1 varies width $d$; as predicted by Theorem 5.6 (formerly 5.9), reducing $d$ (while keeping it large enough for optimization) improves test accuracy for both activations, consistent with the bound’s monotonicity in $d$. In the overparameterized regime, normalized margins increase and track test accuracy, aligning with the margin-based guarantee in Theorem 6.2.
>
> **Missing/incomplete ablations.**
>
> We agree $m$ and $p$ are key in our theory. However, our results are upper bounds, so they do not require exact scaling matches and do not preclude generalization with fewer samples.
>
> **Q1:**
>
> These are artifacts of our proof constructions, not fundamental limits. Theorem. 6.3 provides only a sufficient condition. The exponential $m$-dependence arises from the construction in Section I; it is not necessary. By contrast, our ReLU lower bound (Theorem. 4.2) scales linearly with $m/p$, not exponentially.
>
> Mechanistically, our explicit ReLU embedding construction follows the “pizza-algorithm” pattern [1]: fixed attention reduces to MLPs computing vector means and learning periodic embeddings. “Clock” and “pizza” circuits are two instantiations of the same aCRT template [2], so the construction has interpretability backing.
>
> Empirically (Figs. 3, 11, 13), with weight decay, ReLU networks learn modular addition with large normalized margins, indicating the exponential terms in Theorem. 6.3 are not intrinsic.
>
> Bottom line: learning periodic embeddings with ReLU is less parameter-efficient than with sinusoidal activations, but our result is weaker positive (not tight).
>
> **Q2:**
>
> Width: no. Normalized margin: yes. Empirically (Figs. 3, 11, 13), normalized margins exceeded Theorem. 6.3’s tiny threshold. The theorem’s width bound is only for certification in the analysis, trained models were far narrower.
>
> **Q3:**
>
> We neither claim an exponential separation nor make any quantitative guarantees about the performance gap between sine and ReLU networks. The reported accuracies and margins in Theorems 6.2 and 6.3 are not directly comparable, because for sine models we apply weight decay only to the second layer, whereas for ReLU models we decay both layers. In the overparameterized regime, our experiments are intended to demonstrate the relevance of margin and weight decay. In the underparameterized regime, by “consistently outperforms” we mean that the sine activation achieves strictly higher test accuracy than ReLU for every evaluated $(d,n)$. A single gap metric $Acc(\sin) - Acc(\text{ReLU})$ can be misleading near chance or saturation, so we always report test accuracies for both activations.
>
> **References:**
>
> [1] Z. Zhong, Z. Liu, M. Tegmark, and J. Andreas, "The Clock and the Pizza: Two Stories in Mechanistic Explanation of Neural Networks," in Advances in Neural Information Processing Systems (NeurIPS), 2023.
>
> [2] G. McCracken, G. Moisescu-Pareja, V. Létourneau, D. Precup, and J. Love, "Uncovering a Universal Abstract algorithm for Modular Addition in Neural Networks," arXiv:2505.18266, 2025.

---

### Author Response · Authors · 2025-11-22
**General Response**

We thank all reviewers for their thoughtful feedback. We have revised the manuscript. Major updates are highlighted in red.

**Summary of updates:**

1. **Benefits of sine activations in modular addition:** We added out-of-domain expressivity results (Thms. 4.2, 4.4) and length-generalization experiments for a two-layer MLP (Figs. 4–5). We also added a single-head, single-layer, decoder-only Transformer experiment (Fig. 6).
2. **Scope of claims (EK3F):** We clarify in the introduction that modular addition serves as the sole theoretical testbed for periodic tasks.
3. **Optimization (fCwo):** We added vanilla SGD results for a two-layer MLP in the underparameterized regime (Fig. 9).
4. **Presentation (ZF3N):** We unified appendix cross-references and notation, added high-level outlines at the start of each appendix section, and introduced Appendix D (Proof Outlines).
5. **ReLU bounds (iHML):** We added Remark 6.4 clarifying that the exponential bounds arise from the interpolation-based construction of the two-layer ReLU MLP in Section I (an artifact of this setting).
6. **Near-optimality clarity (fCwo, EK3F):** We modified Theorem F.11 to clarify the near-optimality claim in Remark 5.9 by stating that optimality holds for label-permutation–equivariant learners (e.g., AdaGrad, Adam, GD/SGD with momentum) under i.i.d. final-layer initialization.
7. **General Natarajan bound (ZF3N):** We added Appendix H explaining the capacity bounds in Table 1 and stating task-agnostic theorems for two-layer MLPs in the underparameterized regime (Theorem H.6).

---

### Meta-Review · Area_Chair_e7qj · 2026-01-07

**Summary:**

The paper presents a theoretical comparison between sinusoidal and ReLU activations in the specific context of modular addition. The authors establish a sharp expressivity gap, proving that Sine networks require significantly less width than ReLU networks for this task, and provide corresponding Natarajan-dimension and margin-based generalization bounds. While the mathematical rigor and the clarity of the proofs are highly acknowledged by all reviewers, the consensus highlights a critical debate regarding the work's overall scientific impact, given its narrow task-specific focus and the degree to which it confirms existing architectural intuitions.

**Reviewer Concerns:**

Addressed:
- The authors addressed concerns regarding the clarity of the proofs and unified the notation.They clarified that the Natarajan-dimension analysis can be extended to task-agnostic settings for two-layer MLPs and added a Transformer-based experiment to demonstrate potential scalability beyond simple MLPs.

Outstanding:
- One concern is that the core insight—periodic activations are suitable for periodic data—is a well-established inductive bias. Reviewers questioned if the rigorous formalization in this specific toy setting provides sufficient new insight for the broader machine learning community.
- Reviewer mentioned the provable benefit may be overstated relative to how ReLU networks actually behave in practice.
- The study remains largely confined to modular addition. The lack of evidence on more diverse periodic or quasi-periodic tasks limits the generalizability of the proposed theoretical framework.

**Reviewer Scores:**

Reviewer iHML & ZF3N: Maintained their scores, citing that the paper represents an elegant but incremental formalization without addressing broader optimization or scaling challenges.

Reviewer fCwo & EK3F: While recognizing the technical correctness and cleanliness of the mathematical construction, their original evaluations specifically noted significant reservations regarding the paper's practical influence beyond synthetic benchmarks.

---

### Decision · Program_Chairs · 2026-01-26

Reject